# Classification of Convergent OPE Channels for Lorentzian CFT Four-Point Functions

**Jiaxin Qiao**[a,b]

[a] Laboratoire de Physique de l'Ecole normale supérieure, ENS,
Université PSL, CNRS, Sorbonne Université, Université de Paris, F-75005 Paris, France
[b] Institut des Hautes Études Scientifiques, 91440 Bures-sur-Yvette, France

### Abstract

We analyze the convergence properties of operator product expansions (OPE) for Lorentzian CFT four-point functions of scalar operators. We give a complete classification of Lorentzian four-point configurations. All configurations in each class have the same OPE convergence properties in s-, t- and u-channels. We give tables including the information of OPE convergence for all classes. Our work justifies that in a subset of the configuration space, Lorentzian CFT four-point functions are genuine analytic functions. Our results are valid for unitary CFTs in $d \geq 2$. Our work also provides some Lorentzian regions where one can do bootstrap analysis in the sense of functions.

June 2021

# 1  Introduction

In this paper we study the convergence properties of operator product expansion (OPE) for Lorentzian four-point functions in conformal field theories (CFT).

Historically, analyticity of correlation functions is an important bridge connecting Lorentzian quantum field theories (QFT) and Euclidean QFTs. Starting from a Lorentzian correlator, we can get a Euclidean correlator by analytically continuing the time variables onto the imaginary axis [1]. Under certain conditions we can also do the reverse [2, 3, 4]. This procedure of analytic continuation, called Wick rotation, allows us to explore the Lorentzian nature of QFTs which may originate from statistical models in the Euclidean signature.

The Lorentzian correlators are not always genuine functions, instead they belong to a class of tempered distributions which are called Wightman distributions [5]. It is interesting to know at which Lorentzian configurations $(x_1, \ldots, x_n)$ the correlators $G_n(x_1, \ldots, x_n)$ are indeed functions. The Wightman distributions are known to be analytic functions in some regions $\mathcal{J}_n$ which are the sets of "Jost points" [6] ($G_n$ are called Wightman functions in their domains of analyticity).[1] $\mathcal{J}_n$ corresponds to some (not all) Lorentzian

---

[1]It does not mean that the Lorentzian correlators cannot be functions in other regions. For example, the correlators of generalized free fields are functions aside from light-cone singularities. Here we are talking about the minimal domain of

configurations with totally space-like separations. By using the microscopic causality constraints, one can extend $G_n$ to a larger domain, including all configurations with totally space-like separations [7]. In Minkowski space $\mathbb{R}^{d-1,1}$,[2] two points can also have time-like or light-like separation. The Lorentzian correlators usually diverge at configurations with light-like separations, and these configurations are called light-cone singularities [8]. Except for some exactly solvable models, the Lorentzian correlators at configurations which contain time-like separations are not fully studied.

There are more constraints in CFTs. In general QFTs, the domains of Wightman functions are Poincaré invariant, while in CFTs this Poincaré invariant domain can be further extended by using conformal symmetry. Furthermore, in CFTs we have better control on correlators with the help of OPE [9]. A successful example is the four-point functions in 2d local unitary CFTs, where the conformal algebra is infinite dimensional [10]. In this case, by using Al. Zamolodchikov's uniformizing variables $q, \bar{q}$ [11], one can show that the four-point function is regular analytic at all possible Lorentzian configurations aside from light-cone singularities [12]. We are going to study a similar problem in $d \geq 3$, for which the conformal group is finite dimensional and the radial coordinates $\rho, \bar{\rho}$ [13] are used in our analysis.[3] In addition, in 2d there exists non-local unitary CFTs, which have only the global conformal symmetry. The analysis in this work also applies to 2d non-local unitary CFTs.

Recently the conformal bootstrap approach has become a powerful tool in the study of strongly coupled systems [15]. On the numerical side, it gives precise predictions of experimentally measurable quantities, such as the critical exponents of the 3d Ising model [16, 17, 18, 19, 20], $O(N)$ model [20, 21, 22, 23] and other critical systems. The functional methods, which are used in the numerical approach, can be realized analytically in low dimensions , and lead to insights into low dimensional CFTs and S-matrices [24, 25, 26, 27]. While the basic CFT assumptions are made in the Euclidean signature, many attempts have been made to study the bootstrap equations in the Lorentzian signature [28, 29, 30, 31, 32, 33, 34]. In the conformal bootstrap approach, for crossing equations to be valid in the sense of functions, there should be at least two convergent OPE channels. To play the bootstrap game for four-point functions in the Lorentzian signature, it is important to know the convergent domains of various OPE channels. This provides an additional motivation for our work.

The main goal of this work is to give complete tables of Lorentzian four-point configurations with the information about convergence in the sense of analytic functions in various OPE channels. In this paper we will mostly focus on four-point functions of identical scalar operators. Our techniques can be immediately generalized to the case of non-identical scalar operators (see section 7). The four-point funcitons of spinning operators require extra work because of tensor structures. In this paper, we will only make some comments on the case of spinning operators. One may also be interested in the convergence of OPE in the sense of distributions [35]. We leave the discussions of distributional properties to the series of papers [36, 37, 38].

The outline of this paper is as follows. In section 2 we introduce the main problem and provide a quick summary of the main results in this paper. In section 3 we justify the analytic continuation of the CFT four-point function to the domain $\mathcal{D}$ (which will be defined in section 2.1), and the Lorentzian configurations live on the boundary of $\overline{\mathcal{D}}$. In section 4 we give criteria of OPE convergence in s-, u- and t-channels. In section 5, we make a classification of the Lorentzian four-point configurations. All configurations in the same class have the same convergent OPE channels. All information on the OPE

---

Lorentzian correlators which can be derived from general principles of QFT.

   [2]Often one uses Minkowski space to denote $\mathbb{R}^{3,1}$ only. While in this paper, we use this terminology for $\mathbb{R}^{d-1,1}$ and general $d$.

   [3]The set of four-point configurations $(x_1, x_2, x_3, x_4)$ with $|\rho|, |\bar{\rho}| < 1$ is a subset of $(x_1, x_2, x_3, x_4)$ with $|q|, |\bar{q}| < 1$. Since the $q$-variable argument is based on the Virasoro symmetry which is only true in 2d [14], we cannot apply it to the case of $d \geq 3$.

convergence properties can be looked up in appendix C. In section 6 we review some classical results from Wightman QFT, and compare them with CFT four-point functions. In section 7 we generalize our results to the case of non-identical scalar operators and make some comments on the case of spinning operators. In section 8 we make conclusions and point out some open questions related to this work.

## 2 Main problem and summary of results

### 2.1 Main problem

We start from CFT in the Euclidean signature. Let $x_k = (\tau_k, \mathbf{x}_k)$ denote the $k$-th point in the Euclidean space ($k = 1, 2, 3, 4$), where $\tau_k = x_k^0$ is the temporal variable and $\mathbf{x}_k = (x_k^1, x_k^2, \ldots, x_k^{d-1}) \in \mathbb{R}^{d-1}$ represents the vector of spatial variables. Lorentzian points are given by Wick rotating the temporal variables: $\tau = it$ where $t$ is a real number. To get Lorentzian four-point functions we need to analytically continue the Euclidean four-point functions with respect to temporal variables. We define the Wick rotation of the four-point function as follows:

**Step 1.**

We construct a function $G_4(x_1, x_2, x_3, x_4)$ such that:

- $G_4$ has domain $\mathcal{D}$ of complex $\tau_k$ and real $\mathbf{x}_k$. The temporal variables $(\tau_1, \tau_2, \tau_3, \tau_4)$ are in the set

$$\mathbb{C}_>^4 := \left\{ (\tau_1, \tau_2, \tau_3, \tau_4) \in \mathbb{C}^4 \middle| \mathrm{Re}(\tau_1) > \mathrm{Re}(\tau_2) > \mathrm{Re}(\tau_3) > \mathrm{Re}(\tau_4) \right\}. \tag{1}$$

  In other words, $\mathcal{D} = \mathbb{C}_>^4 \times \mathbb{R}^{4(d-1)}$.

- $G_4$ is analytic in the temporal variables $\tau_k$ and continuous in the spatial variables $\mathbf{x}_k$.

- $G_4$ agrees with the Euclidean four-point function $G_4^E$ when all the temporal variables are real.

**Step 2.**

The Wick rotation to Lorentzian CFT four-point function is defined by

$$G_4^L(t_k, \mathbf{x}_k) := \lim_{\substack{\epsilon_k \to 0 \\ \epsilon_1 > \epsilon_2 > \epsilon_3 > \epsilon_4}} G_4(\epsilon_k + it_k, \mathbf{x}_k) \tag{2}$$

when such a limit exists.

The reason we define Wick rotation in the above way is that we expect the Lorentzian CFT four-point function to be a Wightman four-point distribution, which is the boundary value of the Wightman four-point function from its domain of complex coordinates [1]. The domain of the four-point Wightman function includes $\mathcal{D}$, so the limit (2) gives the Wightman four-point distribution when such a limit exists. We will discuss this in section 6.

The main problem we want to discuss in this paper is:

- In which Lorentzian regions does the Lorentzian CFT four-point function, defined by (2), have a convergent operator product expansion in the sense of functions?

The goal of this paper is to give tables which contain OPE convergence properties of four-point functions at all possible Lorentzian configurations.

## 2.2 Summary of results

In this subsection, we provide a quick summary of the main results for readers who wish to know the general ideas of this paper before going into the technical details. Readers will find here:

- The proof of that the Euclidean CFT four-point function $G_4^E$ has analytic continuation to the domain $\mathcal{D}$ (i.e. $\mathrm{Re}(\tau_1) > \mathrm{Re}(\tau_2) > \mathrm{Re}(\tau_3) > \mathrm{Re}(\tau_4)$) (section 3). The analytic continuation will be performed by using the s-channel OPE, which means taking the OPE $\phi(x_1) \times \phi(x_2)$ in the four-point function

$$G_4^E(x_1, x_2, x_3, x_4) = \langle \phi(x_1)\phi(x_2)\phi(x_3)\phi(x_4)\rangle. \tag{3}$$

  The key observation is that for any four-point configuration in $\mathcal{D}$, the radial variables $\rho$ and $\bar{\rho}$ belong to the open unit disk, i.e. $|\rho|, |\bar{\rho}| < 1$. This observation, together with the series expansion of $G_4^E$ in $\rho$ and $\bar{\rho}$, implies that the s-channel OPE is convergent for any configuration in $\mathcal{D}$.

  One technical subtlety in $d \geqslant 3$ is that the radial variables $\rho$ and $\bar{\rho}$ are not individually globally well-defined analytic functions (appendix A). We treat this subtlety carefully when performing the analytic continuation, and show that $G_4$ (constructed in terms of $\rho$ and $\bar{\rho}$) is single valued and analytic everywhere in $\mathcal{D}$ (section 3.4).

- The criteria of OPE convergence of the Lorentzian CFT four-point function $G_4^L$ in s-, t- and u-channels (section 4). $G_4^L$ is defined to be the boundary value of analytically continued Euclidean four-point function (see eq. (2)). One can imagine that the OPE convergence properties of $G_4^L$ rely on the behavior of cross-ratio variables along the analytic continuation path. In the end we will see that for any fixed Lorentzian four-point configuration, one can check the criteria using any analytic continuation path in $\mathcal{D}$ (starting from a Euclidean four-point configuration), and the conclusion does not depend on the choice of the path.

- A classification of the Lorentzian four-point configurations (section 5). The Lorentzian configurations are classified into a finite number of groups according to the range of cross-ratio variables $(z, \bar{z})$ (section 5.1) and the causal orderings (section 5.2). The point is that in each group, all configurations have the same OPE convergence properties (section 5.3).

  Then the problem is reduced to checking convergence properties in a finite number of cases. The conclusion of OPE convergence properties is lengthy because in our classification there are many groups (although finite) to check, so we leave this part to appendix C. We share the Mathematica code for readers who wish to check our results (see the ancillary file "/anc/OPE_check.nb").

- A review of the domain of analyticity of correlation functions in Wightman QFT (section 6). It was showed in the classic literature that the Wightman function is regular analytic in the totally space-like kinematic region. The domain of the CFT four-point function contains this region, and furthermore it contains much more regions including time-like separation of points.

- Generalization to non-identical scalar four-point functions (section 7). We show that the OPE convergence properties of non-identical scalar four-point functions are the same as the identical scalar case, using Cauchy-Schwarz argument. We also make a comment on the main technical difficulty that arises in the case of spinning operators.

# 3 Euclidean CFT four-point function and its analytic continuation

In this section justify the first step of Wick rotation: analytically continuing the Euclidean CFT four-point function $G_4^E$ to the domain $\mathcal{D}$.

## 3.1 Euclidean CFT four-point function

Consider a Euclidean CFT four-point function of identical scalar primary operators $\phi$ with scaling dimension $\Delta$. By conformal symmetry we write it as

$$G_4^E(x_1, x_2, x_3, x_4) := \langle \phi(x_1)\phi(x_2)\phi(x_3)\phi(x_4) \rangle = \frac{g(u,v)}{\left(x_{12}^2\right)^\Delta \left(x_{34}^2\right)^\Delta} \tag{4}$$

where $x_{ij}^2$ are defined by

$$x_{ij}^2 := \sum_{\mu=0}^{d-1} (x_i^\mu - x_j^\mu)^2 = (\tau_i - \tau_j)^2 + (\mathbf{x}_i - \mathbf{x}_j)^2, \tag{5}$$

and $g(u,v)$ is a function of cross-ratios

$$u = \frac{x_{12}^2 x_{34}^2}{x_{13}^2 x_{24}^2}, \quad v = \frac{x_{14}^2 x_{23}^2}{x_{13}^2 x_{24}^2}. \tag{6}$$

$G_4^E$ is originally defined in the Euclidean signature. To extend $G_4^E$ to the domain of complex temporal variables,[4] we first analytically continue $x_{ij}^2$ by eq. (5). Since the functions $x_{ij}^2$ do not vanish when the configurations are in $\mathcal{D}$,[5] the cross-ratios do not diverge. As a result, cross-ratios have analytic continuation to $\mathcal{D}$. Since $x_{ij}^2 \neq 0$ and $\mathcal{D}$ is simply connected, the prefactor $(x_{12}^2 x_{34}^2)^{-\Delta}$ in (4) also has analytic continuation to $\mathcal{D}$.

Therefore, to show that $G_4^E(\tau_k, \mathbf{x}_k)$ has analytic continuation to the domain $\mathcal{D}$, it remains to show that $g(\tau_k, \mathbf{x}_k)$ has analytic continuation to the domain $\mathcal{D}$.[6]

In general, we do not know the exact expression of $g(\tau_k, \mathbf{x}_k)$. We want to extend its domain according to the basic properties of Euclidean unitary CFT [39, 40]:

- Conformal invariance (which has already been used in (4)).

- Reflection positivity (which is the Euclidean version of unitarity [2]).

- Convergence of operator product expansions in the Euclidean space.

Our main idea is to use the above properties via radial variables $\rho, \bar\rho$ [13], which we will introduce below. Roughly speaking, we want to first analytically continue the function $g(\rho, \bar\rho)$ by a good series expansion. Then we want to stick the analytic functions $\rho(\tau_k, \mathbf{x}_k)$ and $\bar\rho(\tau_k, \mathbf{x}_k)$ into $g(\rho, \bar\rho)$ to get analytic continuation of $g(\tau_k, \mathbf{x}_k)$.

---

[4]We want to remark here that in principle one can also consider analytically continuing the above functions to the domain of complex spatial variables. But since in this work we only want to discuss about the four-point functions in the Lorentzian signature, where $\mathbf{x}_k$ are always real, we will not consider the analytic continuation problem with respect to spatial variables. By analytically continuing some function $f(\tau_k, \mathbf{x}_k)$ to $\mathcal{D}$, we mean extending the domain of $f$ to $\mathcal{D}$, on which $f$ is analytic in $\tau_k$ and continuous in $\mathbf{x}_k$.

[5]Let $\tau_k = \epsilon_k + it_k$ ($k = 1, 2, 3, 4$). For $i \neq j$, since $x_{ij}^2 = (\epsilon_i - \epsilon_j + it_i - it_j)^2 + (\mathbf{x}_i - \mathbf{x}_j)^2$ and $\epsilon_i \neq \epsilon_j$, $x_{ij}^2$ is real only when $t_i = t_j$, but then $x_{ij}^2 > 0$.

[6]In this paper we will abuse the notation by writing $g(u,v)$, $g(z,\bar z)$, $g(\rho,\bar\rho)$ and $g(\tau_k, \mathbf{x}_k)$ for four-point function in different coordinates. For example, $g(\tau_k, \mathbf{x}_k) := g(u(\tau_k, \mathbf{x}_k), v(\tau_k, \mathbf{x}_k))$.

## 3.2 Expansion in $z, \bar{z}$

### 3.2.1 Definition of the coordinates

We pass from $u, v$ to $z, \bar{z}$ by

$$u = z\bar{z}, \quad v = (1-z)(1-\bar{z}). \tag{7}$$

The above definition has an ambiguity of interchanging $z$ and $\bar{z}$. We choose $z, \bar{z}$ to be one particular solution of (7):

$$
\begin{aligned}
z(u,v) &= \frac{1}{2}\left(1 + u - v + i\sqrt{4u - (1 + u - v)^2}\right), \\
\bar{z}(u,v) &= \frac{1}{2}\left(1 + u - v - i\sqrt{4u - (1 + u - v)^2}\right).
\end{aligned}
\tag{8}
$$

Since we are interested in the four-point configurations in $\mathcal{D}$, where the temporal variables $\tau_k$ are complex numbers, the cross-ratios $u(\tau_k, \mathbf{x}_k)$ and $v(\tau_k, \mathbf{x}_k)$ are also complex numbers. So we consider $z(u,v), \bar{z}(u,v)$ for $(u,v) \in \mathbb{C}^2$. For complex $(u,v)$, the expressions of $z(u,v), \bar{z}(u,v)$ have the same set of square-root branch points

$$\Gamma_{uv} := \left\{(u,v) \in \mathbb{C}^2 \,\middle|\, 4u - (1 + u - v)^2 = 0\right\}, \tag{9}$$

and (8) is not single-valued when $(u,v) \in \mathbb{C}^2 \backslash \Gamma_{uv}$. We define the variables $w, y$ by

$$w = 1 + u - v, \quad y^2 = 4u - (1 + u - v)^2, \tag{10}$$

and write (8) as

$$z = \frac{w}{2} + \frac{iy}{2}, \quad \bar{z} = \frac{w}{2} - \frac{iy}{2}. \tag{11}$$

As we discuss in appendix A, $y(\tau_k, \mathbf{x}_k)$, and hence $z(\tau_k, \mathbf{x}_k), \bar{z}(\tau_k, \mathbf{x}_k)$, are single-valued in $\mathcal{D}$ for $d = 2$ but not for $d \geq 3$. This leads to some complication in constructing the analytic continuation of $g(\tau_k, \mathbf{x}_k)$, which will be overcome below (see section 3.4).

### 3.2.2 Conformal frame

We would like to introduce a proper conformal frame configuration to understand the geometrical meaning of $z, \bar{z}$. Let $(\mu\nu)$-plane denote the 2d subspace of $\mathbb{R}^d$ which only has non-vanishing coordinates $x^\mu, x^\nu$ (recall that $x = (x^0, x^1, x^2, \ldots, x^{d-1})$). In Euclidean space, there exists a conformal transformation which maps the configuration $C = (x_1, x_2, x_3, x_4)$ onto the $(01)$-plane:

$$
\begin{aligned}
x_1' &= 0, \\
x_2' &= (a, b, 0, \ldots, 0), \\
x_3' &= (1, 0, \ldots, 0), \\
x_4' &= \infty.
\end{aligned}
\tag{12}
$$

Then $z = a + ib$, $\bar{z} = a - ib$. We call (12) a conformal frame configuration of $C$. Noticing that the reflection $(x^0, x^1, x^2, \ldots, x^{d-1}) \mapsto (x^0, -x^1, x^2, \ldots, x^{d-1})$ is a conformal transformation which preserves $(01)$-plane

and keeps $x_1', x_3', x_4'$ in (12) fixed, the conformal frame configuration is not unique: it is allowed to replace $b$ with $-b$ in (12). We see that changing $b$ to $-b$ is the same as interchanging $z$ and $\bar{z}$.

Let $\mathcal{D}_E$ be the Euclidean subset of $\mathcal{D}$:

$$\mathcal{D}_E := \left\{ (x_1, x_2, x_3, x_4) \in \mathcal{D} \,\middle|\, \tau_k \in \mathbb{R}, \quad k = 1, 2, 3, 4 \right\}, \quad (x_k = (\tau_k, \mathbf{x}_k)). \tag{13}$$

By using the conformal frame, one can show that

- All configurations in $\mathcal{D}_E$ have the following property:

$$z, \bar{z} \in \mathbb{C} \backslash [1, +\infty). \tag{14}$$

This follows from the fact that conformal transformations map circles to circles or lines, preserving cyclic order.[7] Suppose we have a configuration $C$ with $z = \bar{z} \in [1, +\infty)$, then the conformal frame configuration (12) of $C$ satisfies $x_2' = (a, 0, \ldots, 0)$ with $a > 1$, which means $(x_1', x_2', x_3', x_4')$ have cyclic order [1324]. However, the cyclic order [1324] does not exist in $\mathcal{D}_E$ because of the constraint $\tau_1 > \tau_2 > \tau_3 > \tau_4$.

### 3.2.3 Series expansion

By (7), we can think of $g(u(z, \bar{z}), v(z, \bar{z}))$ as a function of $z, \bar{z}$. Since $u, v$ are symmetric polynomials of $z, \bar{z}$, we have

$$g(z, \bar{z}) = g(\bar{z}, z), \quad (z^* = \bar{z}). \tag{15}$$

The constraint $z^* = \bar{z}$ in (15) is because our assumptions of conformal invariance are made in the Euclidean signature, where $z, \bar{z}$ are complex conjugate to each other. If $g(z, \bar{z})$ has analytic continuation to independent complex $z, \bar{z}$, then it is easy to remove this constraint by the Cauchy-Riemann equation, so that (15) will hold for any $z, \bar{z}$.

In the unitary CFT, the function $g(z, \bar{z})$ is known to have a series expansion in $z, \bar{z}$:

$$g(z, \bar{z}) = \sum_{h, \bar{h} \geq 0} a_{h, \bar{h}} z^h \bar{z}^{\bar{h}}, \tag{16}$$

where $a_{h, \bar{h}}$ are real non-negative coefficients. This expansion can be understood as follows. In the radial quantization picture, we insert a complete basis $\{|h, \bar{h}\rangle\}$ into the four-point function in the conformal frame (12):

$$\sum_{h, \bar{h}} \langle \phi(x_1') \phi(x_2') | h, \bar{h} \rangle \langle h, \bar{h} | \phi(x_3') \phi(x_4') \rangle = \frac{1}{(x_{12}'^2 x_{34}'^2)^\Delta} \sum_{h, \bar{h} \geq 0} a_{h, \bar{h}} z^h \bar{z}^{\bar{h}}, \tag{17}$$

where $h, \bar{h}$ are the eigenvalues of Virasoro generators $L_0, \bar{L}_0$ [14] ($L_0, \bar{L}_0$ also belong to $so(1, d+1)$, the Lie algebra of the global conformal group). Here $h, \bar{h} \geq 0$ are the consequences of unitarity [14].

The expansion (16) is absolutely convergent when $|z|, |\bar{z}| < 1$, so $g(z, \bar{z})$ has analytic continuation from its Euclidean domain $\{z^* = \bar{z}\}$ to the universal covering of $\{0 < |z|, |\bar{z}| < 1\}$. Our purpose is to extend the domain of $g(\tau_k, \mathbf{x}_k)$ from $\mathcal{D}_E$ to $\mathcal{D}$ by composing $g(z, \bar{z})$ with $z(\tau_k, \mathbf{x}_k)$ and $\bar{z}(\tau_k, \mathbf{x}_k)$. However, the set of $(x_1, x_2, x_3, x_4)$ with $|z(\tau_k, \mathbf{x}_k)|, |\bar{z}(\tau_k, \mathbf{x}_k)| < 1$ does not cover $\mathcal{D}$ (it does not even cover $\mathcal{D}_E$). To solve this issue, we will introduce a pair of radial coordinates $\rho, \bar{\rho}$ and expand the function $g$ in $\rho, \bar{\rho}$ (see the next subsection).

---

[7]Given four Euclidean points $x_1, x_2, x_3, x_4$, we say that they have the cyclic order $[i_1 i_2 i_3 i_4]$ if these four points lie on a circle or a line in the order $x_{i_1} x_{i_2} x_{i_3} x_{i_4}$.

### 3.3 Expansion in $\rho, \bar{\rho}$

#### 3.3.1 Definition of the coordinates

We define the radial coordinates $\rho, \bar{\rho}$ by

$$\rho(z) = \frac{(1 - \sqrt{1-z})^2}{z}, \quad \bar{\rho}(\bar{z}) = \frac{(1 - \sqrt{1-\bar{z}})^2}{\bar{z}}, \tag{18}$$

Eq. (18) defines a one-to-one holomorphic map $\rho(z)$ from $z \in \mathbb{C}\backslash[1, +\infty)$ to the open unit disc $|\rho| < 1$. The same is true for $\bar{z}$ and $\bar{\rho}$. By composing (18) with $z(\tau_k, \mathbf{x}_k)$ and $\bar{z}(\tau_k, \mathbf{x}_k)$, we get functions $\rho(\tau_k, \mathbf{x}_k)$ and $\bar{\rho}(\tau_k, \mathbf{x}_k)$. For configurations in $\mathcal{D}_E$, by the property (14) of $z, \bar{z}$ , we have:

- In the Euclidean region $\mathcal{D}_E$, we have $|\rho(\tau_k, \mathbf{x}_k)| , |\bar{\rho}(\tau_k, \mathbf{x}_k)| < 1$.

Analogously to the conformal frame for $z, \bar{z}$, for $\rho, \bar{\rho}$ there exists a conformal transformation which maps $C$ onto the (01)-plane:

$$\begin{aligned}
x_1'' &= (\alpha, \beta, 0, \ldots, 0), \\
x_2'' &= (-\alpha, -\beta, 0, \ldots, 0), \\
x_3'' &= (-1, 0, \ldots, 0), \\
x_4'' &= (1, 0, \ldots, 0).
\end{aligned} \tag{19}$$

with $\alpha^2 + \beta^2 < 1$. Then $\rho = \alpha + i\beta$, $\bar{\rho} = \alpha - i\beta$.

#### 3.3.2 Series expansion

By (18), the maps from $\rho, \bar{\rho}$ to $z, \bar{z}$ are given by

$$z = \frac{4\rho}{(1+\rho)^2}, \quad \bar{z} = \frac{4\bar{\rho}}{(1+\bar{\rho})^2}. \tag{20}$$

We get $g(\rho, \bar{\rho})$ by composing (20) with $g(z, \bar{z})$. By (15) and (20), we have

$$g(\rho, \bar{\rho}) = g(\bar{\rho}, \rho), \quad (\rho^* = \bar{\rho}). \tag{21}$$

In the Euclidean unitary CFT, the function $g(\tau_k, \mathbf{x}_k)$ is known to have an expansion in radial coordinates [9]:

$$g(\rho, \bar{\rho}) = \sum_{h, \bar{h} \geq 0} b_{h, \bar{h}} \rho^h \bar{\rho}^{\bar{h}} \tag{22}$$

where $b_{h, \bar{h}}$ are positive coefficients. This expansion can be obtained by inserting a complete basis $\{|h, \bar{h}\rangle\}$ into the four-point function at the configuration (19):

$$\sum_{h, \bar{h}} \langle \phi(x_1'') \phi(x_2'') | h, \bar{h} \rangle \langle h, \bar{h} | \phi(x_3'') \phi(x_4'') \rangle = \frac{1}{(16\rho\bar{\rho})^\Delta} \sum_{h, \bar{h} \geq 0} b_{h, \bar{h}} \rho^h \bar{\rho}^{\bar{h}}. \tag{23}$$

Furthermore, the $\rho$-expansion (22) can be rearranged in the following way:

$$g(\rho, \bar{\rho}) = \sum_{\Delta \geq 0} \sum_{l \in \mathbb{N}} c_{\Delta, l} (\rho\bar{\rho})^{\Delta/2} P^l(\rho, \bar{\rho}),$$

$$P^l(\rho, \bar{\rho}) = \sum_{k=0}^{l} p_k^l \left(\frac{\rho}{\bar{\rho}}\right)^{l/2-k}, \quad p_k^l = p_{l-k}^l, \tag{24}$$

where the function $P^l(\rho, \bar\rho)$ in (24) is another form of the Gegenbauer polynomial $C_l^{d/2-1}$ [13]:

$$P^l(\rho, \bar\rho) = C_l^{d/2-1}(\cos\theta), \quad \left(\rho = re^{i\theta}, \bar\rho = re^{-i\theta}\right). \tag{25}$$

The function $g(\rho, \bar\rho)$ is originally defined in the Euclidean region $\rho^* = \bar\rho$. Considering now $\rho, \bar\rho$ to be independent complex variables, since the expansion (22) is absolutely convergent when $|\rho|, |\bar\rho| < 1$, it defines an analytic function on the universal covering of the domain

$$R := \left\{ (\rho, \bar\rho) \in \mathbb{C}^2 \middle| 0 < |\rho|, |\bar\rho| < 1 \right\}. \tag{26}$$

We write radial coordinates as

$$\rho = e^{i\chi}, \bar\rho = e^{i\bar\chi}. \tag{27}$$

The universal covering of $R$ is characterized by a product of upper half planes

$$X := \left\{ (\chi, \bar\chi) \in \mathbb{C}^2 \middle| \mathrm{Im}(\chi), \mathrm{Im}(\bar\chi) > 0 \right\} \tag{28}$$

and the covering map is (27). The Euclidean region of $X$ corresponds to $\chi^* = -\bar\chi$. By (24), the function $g(\chi, \bar\chi)$ on $X$ has the following properties:

$$\begin{aligned} g(\chi, \bar\chi) &= g(\bar\chi, \chi), \\ g(\chi, \bar\chi) &= g(\chi + 2\pi, \bar\chi - 2\pi). \end{aligned} \tag{29}$$

## 3.4  Analytic continuation: case $d \geq 3$

### 3.4.1  Main idea

In this section we will study the analytic continuation of $g(\tau_k, \mathbf{x}_k)$ in $d \geq 3$. Our goal is to analytically continue $g(\tau_k, \mathbf{x}_k)$ to $\mathcal{D}$. Naively, one may want to construct this analytic continuation by the following compositions:

$$(\tau_k, \mathbf{x}_k) \mapsto (u, v) \mapsto (z, \bar z) \mapsto (\rho, \bar\rho) \mapsto (\chi, \bar\chi) \mapsto g(\chi, \bar\chi). \tag{30}$$

This construction requires two conditions:

1. The step $(\tau_k, \mathbf{x}_k) \mapsto (z, \bar z)$ in (30) should be well defined.

2. All configurations in $\mathcal{D}$ satisfy $|\rho|, |\bar\rho| < 1$, or equivalently, $z, \bar z \notin [1, +\infty)$.

The first condition holds if we have well-defined analytic functions $z(\tau_k, \mathbf{x}_k)$ and $\bar z(\tau_k, \mathbf{x}_k)$ on $\mathcal{D}$. However, as already mentioned in section 3.2.1, such analytic functions do not exist in $d \geq 3$ (they exist in 2d). We will discuss the 2d case in section 3.5. For the $d \geq 3$ case, we discuss the non-existence of $z(\tau_k, \mathbf{x}_k), \bar z(\tau_k, \mathbf{x}_k)$ in appendix A.

Concerning the second condition, we have a crucial observation:

**Theorem 3.1.** [37] The above condition 2 holds in any $d \geq 2$.

This theorem is one of the results in [37], where the proof is given (see [37], lemma 6.1 and the proof in its section 6.4). In this paper we will accept it as a fact. For readers who wish to get some intuition on why $|\rho|, |\bar{\rho}| < 1$ in $\mathcal{D}$, we will give a proof of the 2d case in section 3.5.

Since it is impossible to construct analytic functions $z(\tau_k, \mathbf{x}_k), \bar{z}(\tau_k, \mathbf{x}_k)$ on $\mathcal{D}$ in $d \geq 3$, we cannot naively do the composition like (30). We will construct analytic continuation of $g(\tau_k, \mathbf{x}_k)$ as follows.

Let $\Gamma$ be the preimage of $\Gamma_{uv}$ in $\mathcal{D}$ (see eq. (9)). Given a four-point configuration $C = (x_1, x_2, x_3, x_4) \in \mathcal{D}$, we choose a path

$$
\begin{aligned}
\gamma : \ & [0, 1] \ \longrightarrow \ \mathcal{D}, \\
& \gamma(0) \in \mathcal{D}_E \backslash \Gamma, \\
& \gamma(1) = C, \\
& \gamma(s) \in \mathcal{D} \backslash \Gamma, \quad s < 1.
\end{aligned}
\tag{31}
$$

The values of $u(s), v(s)$ along $\gamma$ are uniquely computable via (6). We would like to also define $z, \bar{z}, \rho, \bar{\rho}, \chi, \bar{\chi}$ along the path $\gamma$. For this we make some conventions: at the start point of $\gamma$ we choose

$$
\begin{aligned}
\operatorname{Im}(z) \geq 0, \quad \operatorname{Im}(\bar{z}) \leq 0, \\
0 \leq \operatorname{Re}(\chi) \leq \pi, \quad -\pi \leq \operatorname{Re}(\bar{\chi}) \leq 0.
\end{aligned}
\tag{32}
$$

This uniquely determines $z, \bar{z}, \rho, \bar{\rho}, \chi, \bar{\chi}$ at $s = 0$. Since we chose $\gamma$ such that $\gamma(s) \notin \Gamma$ before the final point, the subsequent paths of $z, \bar{z}, \rho, \bar{\rho}, \chi, \bar{\chi}$ at $s > 0$ are then uniquely determined by continuity. We define path-dependent variables

$$
z(C, \gamma), \bar{z}(C, \gamma), \rho(C, \gamma), \bar{\rho}(C, \gamma), \chi(C, \gamma), \bar{\chi}(C, \gamma)
\tag{33}
$$

to be the variables at the final point $\gamma(1) = C$. Our goal is to show that

- The function $g\left(\chi(\tau_k, \mathbf{x}_k, \gamma), \bar{\chi}(\tau_k, \mathbf{x}_k, \gamma)\right)$ is actually independent of the path $\gamma$, so we write it as $g(\tau_k, \mathbf{x}_k)$.

- The function $g(\tau_k, \mathbf{x}_k)$ is analytic in $\tau_k$.

This, then, is how we will analytically continue $g(\tau_k, \mathbf{x}_k)$ to the whole $\mathcal{D}$.

### 3.4.2 Path independent quantities

In the previous subsection we defined path dependent variables (33). Changing the path may change the values of these variables. In this subsection we are going to show that the following quantities are analytic functions on $\mathcal{D}$:

1. $(\rho\bar{\rho})^\Delta$ for any $\Delta$.

2. $\left(\dfrac{\rho}{\bar{\rho}}\right)^{k/2} + \left(\dfrac{\bar{\rho}}{\rho}\right)^{k/2}$, where $k \in \mathbb{Z}$.

We first need to show that the above quantities are path independent, then we need to show the analyticity. For $(\rho\bar{\rho})^\Delta$, by (7) and (18), we write it as

$$
(\rho\bar{\rho})^\Delta = \frac{u^\Delta}{I^{2\Delta}}, \quad I = (1 + \sqrt{1 - z})(1 + \sqrt{1 - \bar{z}}).
\tag{34}
$$

$u^\Delta$ is an analytic function on $\mathcal{D}$ because $(x_{ij}^2)^\Delta$ are non-zero analytic functions on $\mathcal{D}$.

**Lemma 3.2.** $I(\tau_k, \mathbf{x}_k)$ is an analytic function on $\mathcal{D}$.

*Proof.* By theorem 3.1, the variables $z, \bar{z}$ are always in the same branch of the square-root functions $\sqrt{1-z}, \sqrt{1-\bar{z}}$. Together with the fact that changing the path at most interchanges $z$ and $\bar{z}$ (this follows from (7) and from $u, v$ being functions of a point and not of a path), we conclude that $I$ is a path independent quantity.

To show that $I(\tau_k, \mathbf{x}_k)$ is an analytic function on $\mathcal{D}$, we first show that $I(\tau_k, \mathbf{x}_k)$ is an analytic function on $\mathcal{D}\backslash\Gamma$. For a given configuration $C \in \mathcal{D}\backslash\Gamma$, we choose a path $\gamma$ under conditions (31). Then the variables $z, \bar{z}$ at $C$ are determined by $\gamma$. By (8), we can find a neighbourhood $U \subset \mathcal{D}\backslash\Gamma$ of $C$ such that the map $(u, v) \mapsto (z, \bar{z})$ is locally analytic. Thus $I(\tau_k, \mathbf{x}_k)$ is an analytic function on $\mathcal{D}\backslash\Gamma$.

It remains to show that $I(\tau_k, \mathbf{x}_k)$ is analytic near $C \in \Gamma$. For $C \in \Gamma$ we have $z = \bar{z} = z_*$. Because $I(z, \bar{z})$ is symmetric in $z$ and $\bar{z}$, and because $I(z, \bar{z})$ is analytic in the variables $z, \bar{z}$ in the domain $z, \bar{z} \notin [1, +\infty)$, $I(z, \bar{z})$ has the following Taylor expansion near $(z_*, z_*)$:

$$I(z, \bar{z}) = \sum_{m,n \in \mathbb{N}} a_{m,n}(z + \bar{z} - 2z_*)^m (z - \bar{z})^{2n}. \tag{35}$$

By (8) we have

$$
\begin{aligned}
z + \bar{z} &= 1 + u - v \\
(z - \bar{z})^2 &= (1 + u - v)^2 - 4u
\end{aligned}
\tag{36}
$$

Although the map $(u, v) \mapsto (z, \bar{z})$ is not analytic near $(u, v) \in \Gamma_{uv}$, (35) and (36) imply that the function $I(u, v)$ is still locally analytic in the variables $u, v$. Thus $I(\tau_k, \mathbf{x}_k)$, which is the composition of analytic functions $I(u, v)$ and $(u(\tau_k, \mathbf{x}_k), v(\tau_k, \mathbf{x}_k))$, is analytic near $C \in \Gamma$. $\square$

To show $(\rho\bar{\rho})^\Delta$ is an analytic function on $\mathcal{D}$, it remains to show that $[I(\tau_k, \mathbf{x}_k)]^\Delta$ is an analytic function on $\mathcal{D}$. Since all configurations in $\mathcal{D}$ have $z, \bar{z} \notin [1, +\infty)$, we have

$$-\frac{\pi}{2} < \operatorname{Arg}(\sqrt{1-z}), \operatorname{Arg}(\sqrt{1-\bar{z}}) < \frac{\pi}{2}, \tag{37}$$

which implies

$$I(\mathcal{D}) \subset \mathbb{C}\backslash(-\infty, 0]. \tag{38}$$

Thus $I(\mathcal{D})$ does not contain any curve which goes around zero. Together with lemma 3.2, we conclude that $[I(\tau_k, \mathbf{x}_k)]^\Delta$ is an analytic function on $\mathcal{D}$.

Recall definition (27) of $\chi, \bar{\chi}$, the fact that $(\rho\bar{\rho})^\Delta$ is analytic in $\mathcal{D}$ is equivalent to the following lemma:

**Lemma 3.3.** $\chi + \bar{\chi}$ is an analytic function on $\mathcal{D}$.

*Proof.* The above discussion about analyticity of $(\rho\bar{\rho})^\Delta$ was actually proving that $\log(\rho\bar{\rho})$ is analytic in $\mathcal{D}$. By (27) we have

$$\chi + \bar{\chi} = -i \log(\rho\bar{\rho}). \tag{39}$$

Thus $\chi + \bar{\chi}$ is analytic in $\mathcal{D}$. $\square$

For $(\rho/\bar{\rho})^{k/2} + (\bar{\rho}/\rho)^{k/2}$, we write it as

$$\left(\frac{\rho}{\bar{\rho}}\right)^{k/2} + \left(\frac{\bar{\rho}}{\rho}\right)^{k/2} = (\rho\bar{\rho})^{-k/2}\left(\rho^k + \bar{\rho}^k\right). \tag{40}$$

The analyticity of $(\rho\bar{\rho})^{-k/2}$ has been proved. Changing the path at most interchanges $\rho$ and $\bar{\rho}$, so $\rho^k + \bar{\rho}^k$ is path independent. The analyticity of $\left(\rho^k + \bar{\rho}^k\right)$ in $\mathcal{D}$ follows from a similar argument as in the proof of lemma 3.2. Therefore, $(\rho/\bar{\rho})^{k/2} + (\bar{\rho}/\rho)^{k/2}$ is an analytic function on $\mathcal{D}$.

### 3.4.3 The end of the proof that $g(\tau_k, \mathbf{x}_k)$ is analytic in $\mathcal{D}$

In section 3.4.1 and 3.4.2, we introduced path dependent variables $z, \bar{z}, \rho, \bar{\rho}$ in $\mathcal{D}$ and showed that $(\rho\bar{\rho})^\Delta$ and $(\rho/\bar{\rho})^{k/2} + (\bar{\rho}/\rho)^{k/2}$ are analytic functions on $\mathcal{D}$. Now sticking them into the expansion (24), we conclude that

- $g(\tau_k, \mathbf{x}_k) = \sum\limits_{\Delta \geq 0} \sum\limits_{l \in \mathbb{N}} c_{\Delta, l} \, (\rho\bar{\rho})^\Delta P^l(\rho, \bar{\rho})$ is a series of analytic functions on $\mathcal{D}$.

For any $C \in \mathcal{D}$, we can find a neighbourhood $U \subset \mathcal{D}$ of $C$ such that the expansion (24) converges uniformly in $U$.[8] So we conclude that

- $g(\tau_k, \mathbf{x}_k)$ is an analytic function on $\mathcal{D}$.

## 3.5 Analytic continuation: case $d = 2$

In this section we would like to discuss separately the analytic continuation of $g(\tau_k, \mathbf{x}_k)$ in 2d. Although this case is covered by section 3.4, in the 2d case a much simpler construction can be given. This is because the analytic functions $z(\tau_k, \mathbf{x}_k), \bar{z}(\tau_k, \mathbf{x}_k)$ exist. In 2d we use the complex coordinates [14]:

$$z_k = x_k^0 + ix_k^1, \quad \bar{z}_k = x_k^0 - ix_k^1, \quad k = 1, 2, 3, 4. \tag{41}$$

We choose $z, \bar{z}$ to be

$$z = \frac{(z_1 - z_2)(z_3 - z_4)}{(z_1 - z_3)(z_2 - z_4)}, \quad \bar{z} = \frac{(\bar{z}_1 - \bar{z}_2)(\bar{z}_3 - \bar{z}_4)}{(\bar{z}_1 - \bar{z}_3)(\bar{z}_2 - \bar{z}_4)}. \tag{42}$$

One can check that (42) is consistent with (7). Furthermore, $z, \bar{z}$ in (42) are analytic in all variables $x_k^\mu$ except for

$$(z_1 - z_3)(\bar{z}_1 - \bar{z}_3)(z_2 - z_4)(\bar{z}_2 - \bar{z}_4) = 0. \tag{43}$$

In particular, $z(\tau_k, \mathbf{x}_k), \bar{z}(\tau_k, \mathbf{x}_k)$ are analytic functions of $x_k^\mu$ on $\mathcal{D}$.

Let $x_k^0 = \epsilon_k + it_k$. We write down the complex coordinates defined in (41):

$$z_k = \epsilon_k + it_k + ix_k^1, \quad \bar{z}_k = \epsilon_k + it_k - ix_k^1, \quad k = 1, 2, 3, 4. \tag{44}$$

---

[8]We can find a neighbourhood $U$ of $C$ such that $|\rho|, |\bar{\rho}| < R < 1$ in $U$. Then the uniform convergence follows from the fact that (24) is a rearrangement of the series expansion (22) which is absolutely convergent.

$$(\tau_k, \mathbf{x}_k) \in \mathcal{D} \quad \dashrightarrow^{\checkmark} \quad X \ni (\chi, \bar{\chi}) \xrightarrow{g(\chi, \bar{\chi})} \mathbb{C}$$

Figure 3.1: The diagram of maps in 2d.

Then $z, \bar{z}$ are computed via (42). Notice that the Euclidean configuration $C' = (x_1', x_2', x_3', x_4')$,

$$\left(x_k'\right)^0 = \epsilon_k, \quad \left(x_k'\right)^1 = t_k + x_k^1 \tag{45}$$

gives the same $z_k$, and hence the same $z$. Applying eq. (14) to $C'$, we conclude that $z \neq [1, +\infty)$ ($\bar{z} \neq [1, +\infty)$ follows from the same argument). This proves the 2d case of theorem 3.1.

Furthermore, $z, \bar{z} \neq 0$ in $\mathcal{D}$ because $z_i - z_j, \bar{z}_i - z_j \neq 0$ for $i \neq j$. So we conclude that

$$z(\mathcal{D}), \bar{z}(\mathcal{D}) \subset \mathbb{C} \backslash (\{0\} \cup [1, +\infty)) \tag{46}$$

Based on (46), we safely map $z, \bar{z}$ to $\rho, \bar{\rho}$ via (18), and we have $\rho, \bar{\rho} \neq 0$. Then consider the function $g(\chi, \bar{\chi})$ which is analytic in the domain $\text{Im}(\chi), \text{Im}(\bar{\chi}) > 0$ (see section 3.3.2). Figure 3.1 shows the procedure of analytic continuation.

Since $\mathcal{D}$ is simply connected, by the lifting properties of the covering map [41], there exists a continuous map (dashed arrow in figure 3.1) from $\mathcal{D}$ to $X$, which lifts the map from $\mathcal{D}$ to $R$. Such a map is unique if we fix $\chi(\tau_k, \mathbf{x}_k)$ and $\bar{\chi}(\tau_k, \mathbf{x}_k)$ at one configuration in $\mathcal{D}_E$.[9] Because (42), (18), (27) are analytic functions, the map $\mathcal{D} \to X$ defines an analytic function $(\chi(\tau_k, \mathbf{x}_k), \bar{\chi}(\tau_k, \mathbf{x}_k))$. By composing $\chi(\tau_k, \mathbf{x}_k), \bar{\chi}(\tau_k, \mathbf{x}_k)$ with the function $g(\chi, \bar{\chi})$, we get a function $g(\tau_k, \mathbf{x}_k)$ which is analytic in the variables $\tau_k$ and $\mathbf{x}_k = x_k^1$.

# 4 Lorentzian CFT four-point function

## 4.1 Some preparations

In this section we are going to study the OPE convergence of Lorentzian CFT four-point functions. Since we have analytically continued the Euclidean CFT four-point function to the domain $\mathcal{D}$, the next step is to take the limit (2) to the Lorentzian configurations, which are at the boundary of $\mathcal{D}$ (denoted by $\partial \mathcal{D}$).

The main differences between $\mathcal{D}$ and $\partial \mathcal{D}$ are as follows:

- (*Position space*) For any $x_i, x_j$ pair in $\mathcal{D}$, $x_{ij}^2$ is always non-zero, while for $x_i, x_j$ pairs in $\partial \mathcal{D}$, $x_{ij}^2 = 0$ when $x_i$ and $x_j$ are light-like separated:

$$\epsilon_i = \epsilon_j, \quad (t_i - t_j)^2 = (\mathbf{x}_i - \mathbf{x}_j)^2.$$

- (*Cross-ratio space*) For all configurations in $\mathcal{D}$, the variables $z, \bar{z}$ never belong to $\{0\} \cup [1, +\infty)$, while the configurations in $\partial \mathcal{D}$ may have $z$ or $\bar{z} \in \{0\} \cup [1, +\infty)$.

---

[9]In $\mathcal{D}_E$, we choose $\chi(\tau_k, \mathbf{x}_k), \bar{\chi}(\tau_k, \mathbf{x}_k)$ with the constraint $\chi^* = -\bar{\chi}$ because $\rho^*(\tau_k, \mathbf{x}_k) = \bar{\rho}(\tau_k, \mathbf{x}_k)$ for configurations in the Euclidean region.

We call (4), (22) the s-channel expansion of the CFT four-point function. A priori we can also use the t- and u-channel expansions to construct the analytic continuation of the four-point function, starting from the t- and u-channel versions of (4):

$$\langle \phi(x_1)\phi(x_2)\phi(x_3)\phi(x_4) \rangle = \frac{g(u_t, v_t)}{\left(x_{14}^2\right)^\Delta \left(x_{23}^2\right)^\Delta} = \frac{g(u_u, v_u)}{\left(x_{13}^2\right)^\Delta \left(x_{24}^2\right)^\Delta}, \tag{47}$$

where

$$u_t = v, \quad v_t = u, \quad u_u = \frac{1}{u}, \quad v_u = \frac{v}{u}. \tag{48}$$

We want to remark that only the s-channel expansion could be used to extend $g(\tau_k, \mathbf{x}_k)$ to the whole $\mathcal{D}$, since theorem 3.1 holds only for the s-channel. We can use t- and u-channel expansion to analytically continue the four-point function to part of $\mathcal{D}$, but not to the whole $\mathcal{D}$. We will also consider t- and u-channel expansions because there are Lorentzian configurations where the s-channel expansion does not converge, but the t- or u-channel expansion converges (see section 4.3.2).

## 4.2 Excluding light-cone singularities

When $x_{ij}^2 = 0$ for some $x_i, x_j$ pair, since at least one of the scaling factors in (4) and (47) is infinity, we expect the four-point function to be infinity. The configurations which contain at least one light-like $x_i, x_j$ pair are called light-cone singularities.

One example, for which the correlation functions are divergent at light-cone singularities, is the generalized free field (GFF). Since we are interested in the Lorentzian configurations where the four-point functions are genuine functions for all unitary CFTs, we only consider the configurations which are not light-cone singularities. In other words, in this paper we will only consider the following set of Lorentzian configurations:

$$\mathcal{D}_L := \left\{ (x_1, x_2, x_3, x_4) \in \partial \mathcal{D} \Big| x_k = (it_k, \mathbf{x}_k), \ \forall k; \quad x_{ij}^2 \neq 0, \ \forall i \neq j \right\}. \tag{49}$$

## 4.3 Criteria of OPE convergence

Now that $x_{ij}^2 \neq 0$ for all configurations in $\mathcal{D}_L$, all the cross-ratios defined in (6) and (48) are finite and non-zero, which implies

$$z, \bar{z} \neq 0, 1, \infty. \tag{50}$$

So the real axis in the $z, \bar{z}$-space is divided into three parts:

$$(-\infty, 0) \cup (0, 1) \cup (1, +\infty). \tag{51}$$

In this section, we are going to establish criteria of OPE convergence in s-, t- and u-channels. The three intervals in (51) will play important roles because each of them is the place where one OPE channel stops being convergent.

### 4.3.1 s-channel

We have analytically continued the four-point function $G_4$ to $\mathcal{D}$. So far, as already mentioned, we only used the s-channel expansion because of theorem 3.1. Actually by using the s-channel expansion, we are able to extend $G_4$ to a larger domain $\mathcal{D}^s \supset \mathcal{D}$ according the constraint $0 < |\rho|, |\bar{\rho}| < 1$ (or equivalently, $z, \bar{z} \neq \{0\} \cup [1, +\infty)$). $\mathcal{D}^s$ contains some but not all Lorentzian configurations. In other words, the Lorentzian four-point function has convergent s-channel OPE on the set $\mathcal{D}^s \cap \mathcal{D}_L$.

By (18) and (50), $\rho, \bar{\rho} \neq 0, \pm 1$ for all configurations in $\mathcal{D}_L$. Because of theorem 3.1 and the continuity, all configurations in $\mathcal{D}_L$ have $|\rho|, |\bar{\rho}| \leq 1$. To check the convergence of s-channel OPE, it suffices to check whether $|\rho|, |\bar{\rho}| \neq 1$ or not. Equivalently, it suffices to check whether $z, \bar{z} \notin (1, +\infty)$ or not.

Therefore, given a Lorentzian configuration $C_L \in \mathcal{D}_L$, we have the following criterion of s-channel OPE convergence:

**Theorem 4.1.** (*s-channel OPE convergence*) If neither $z$ nor $\bar{z}$ computed from $C_L$ belong to $(1, +\infty)$, then the Lorentzian four-point function $G_4$ is analytic at $C_L$ and is given by the formula

$$G_4(C_L) = \frac{g(\chi, \bar{\chi})}{[x_{12}^2 x_{34}^2]^\Delta}. \tag{52}$$

Here $g(\chi, \bar{\chi})$ is the same function as described in section 3.3.2, and the variables $\chi, \bar{\chi}$ are defined by the algorithm in section 3.4.1. The function $g(\chi, \bar{\chi})$ can be computed by the convergent series expansion (22).

### 4.3.2 t-channel and u-channel

We define the variables $z_t, \bar{z}_t$ and $z_u, \bar{z}_u$ by replacing $u, v$ with $u_t, v_t$ and $u_u, v_u$ in (7). By (7) and (48), we choose proper solutions to the t- and u-channel versions of (7), and get the following relations[10]

$$z_t = 1 - z, \quad \bar{z}_t = 1 - \bar{z}, \quad z_u = 1/z, \quad \bar{z}_u = 1/\bar{z}. \tag{53}$$

Then we define the t- and u-channel versions of radial coordinates $\rho_t, \bar{\rho}_t, \rho_u, \bar{\rho}_u$ by replacing $z, \bar{z}$ with $z_t, \bar{z}_t$ and $z_u, \bar{z}_u$ in (18). By (53) and the fact that $z, \bar{z}$ are not real for configurations in $\mathcal{D}_E \backslash \Gamma$, $z_t, \bar{z}_t, z_u, \bar{z}_u$ are also not real for configurations in $\mathcal{D}_E \backslash \Gamma$. In particular, $z_t, \bar{z}_t, z_u, \bar{z}_u \notin [1, +\infty)$ for all configurations in $\mathcal{D}_E \backslash \Gamma$, which allows us to choose $|\rho_t| = |\bar{\rho}_t| < 1$ and $|\rho_u| = |\bar{\rho}_u| < 1$ to start with convergent t- and u-channel expansions. Analogously to the s-channel expansion, the t- and u-channel expansions are defined by replacing $\rho, \bar{\rho}$ with $\rho_t, \bar{\rho}_t$ and $\rho_u, \bar{\rho}_u$ in the series expansion (22).

For all configurations in $\mathcal{D}_E \backslash \Gamma$, the s-, t- and u-channel expansions converge to the same Euclidean CFT four-point function. This consistency condition is called the crossing symmetry [42, 43]. Now let us analytically continue the four-point function via the t-channel expansion. Suppose we have a path $\gamma$ in $\mathcal{D} \backslash \Gamma$ such that $\gamma(0) \in \mathcal{D}_E \backslash \Gamma$, we can find a neighbourhood $U_\gamma \subset \mathcal{D} \backslash \Gamma$ of the set $\{\gamma(s) \mid 0 \leq s \leq 1\}$ and perform the analytic continuation of $z, \bar{z}$ in $U_\gamma$ via (6) and (8).[11] Then we get the analytic continuation of $z_t, \bar{z}_t$ in $U_\gamma$ by the relation in (53). If $z_t, \bar{z}_t \notin (1, +\infty)$ in $U_\gamma$, or equivalently, $|\rho_t|, |\bar{\rho}_t| < 1$ in $U_\gamma$, then the t-channel expansion of $G_4$ is convergent in $U_\gamma$, and gives the analytic continuation to $U_\gamma$. Since the start point $\gamma(0)$ is a Euclidean configuration, $U_\gamma \cap \mathcal{D}_E$ is an open subset of $\mathcal{D}_E$, where the temporal variables $\tau_k$ are independent real numbers. According the crossing symmetry, the s- and t-channel expansions agree in $U_\gamma \cap \mathcal{D}_E$, so they also agree in $U_\gamma$, where $\tau_k$ are independent complex numbers. Furthermore, by taking

---

[10]The other solutions of $z_t, \bar{z}_t, z_u, \bar{z}_u$ differ from (53) by interchanging $z_t, \bar{z}_t$ or $z_u, \bar{z}_u$, which will give the same conclusions of convergence properties in t- and u-channel expansions.

[11]As long as $\gamma(s) \notin \mathcal{D} \backslash \Gamma$ along the path $\gamma$, such a neighbourhood $U_\gamma$ always exists.

the limit from $\mathcal{D}\backslash\Gamma$ to $\Gamma$, we can also use the t-channel expansion to compute the four-point function for configurations in $\Gamma$ with the constraint $|\rho_t|, |\bar{\rho}_t| < 1$, and the result also agrees with the s-channel expansion by continuity. So we conclude that

- Given a configuration $C$ in $\mathcal{D}$, the t-channel expansion gives the same analytic continuation of $G_4$ as the s-channel expansion if there exists a path $\gamma$ in $\mathcal{D}$ such that $\gamma(0) \in \mathcal{D}_E\backslash\Gamma$, $\gamma(1) = C$ and $z_t, \bar{z}_t \notin (1, +\infty)$ along $\gamma$.

Analogously, by replacing $z_t, \bar{z}_t$ with $z_u, \bar{z}_u$, we have the same conclusion for the u-channel expansion.

While theorem 3.1 holds for $z, \bar{z}$, it does not hold for $z_t, \bar{z}_t$ or $z_u, \bar{z}_u$, which means that the t- and u-channel expansions may diverge in $\mathcal{D}$. Unlike the s-channel, the convergence properties of t- and u-channel expansions require not only the values of $z_t, \bar{z}_t, z_u, \bar{z}_u$ of a configuration, but also the values of these variables along a path. For convenience we use the relation (53) to translate $z_t, \bar{z}_t, z_u, \bar{z}_u \notin (1, +\infty)$ to equivalent conditions in $z, \bar{z}$:

$$
\begin{aligned}
z_t, \bar{z}_t \notin (1, +\infty) &\quad\Rightarrow\quad z, \bar{z} \notin (-\infty, 0) \\
z_u, \bar{z}_u \notin (1, +\infty) &\quad\Rightarrow\quad z, \bar{z} \notin (0, 1)
\end{aligned}
\tag{54}
$$

Then it suffices to compute and watch $z, \bar{z}$-curves along the path.

To give criteria of convergence properties in t- and u-channel expansions, we define some quantities which count how $z, \bar{z}$-curves cross the intervals $(-\infty, 0)$ and $(0, 1)$. Given a path $\gamma$ defined as follows

$$
\begin{aligned}
\gamma : [0, 1] &\longrightarrow \overline{\mathcal{D}}, \\
\gamma(0) &\in \mathcal{D}_E\backslash\Gamma, \\
\gamma(s) &\in \mathcal{D}\backslash\Gamma, \quad s < 1,
\end{aligned}
\tag{55}
$$

if the variables $z, \bar{z}$ at the final point $\gamma(1)$ satisfy $z, \bar{z} \notin (-\infty, 0)$, we define

$$
\begin{aligned}
n_t(\gamma) :=\ &\text{number of times } z \text{ crosses } (-\infty, 0) \text{ from above} \\
&- \text{number of times } z \text{ crosses } (-\infty, 0) \text{ from below}, \\
\bar{n}_t(\gamma) :=\ &\text{number of times } \bar{z} \text{ crosses } (-\infty, 0) \text{ from above} \\
&- \text{number of times } \bar{z} \text{ crosses } (-\infty, 0) \text{ from below},
\end{aligned}
\tag{56}
$$

and

$$
N_t(\gamma) := n_t(\gamma) + \bar{n}_t(\gamma).
\tag{57}
$$

Analogously, if the variables $z, \bar{z}$ at the final point $\gamma(1)$ satisfy $z, \bar{z} \notin (0, 1)$, we define

$$
\begin{aligned}
n_u(\gamma) :=\ &\text{number of times } z \text{ crosses } (0, 1) \text{ from above} \\
&- \text{number of times } z \text{ crosses } (0, 1) \text{ from below}, \\
\bar{n}_u(\gamma) :=\ &\text{number of times } \bar{z} \text{ crosses } (0, 1) \text{ from above} \\
&- \text{number of times } \bar{z} \text{ crosses } (0, 1) \text{ from below},
\end{aligned}
\tag{58}
$$

and

$$
N_u(\gamma) := n_u(\gamma) + \bar{n}_u(\gamma).
\tag{59}
$$

Let us consider the t-channel expansion. We claim that $N_t$ is a path independent quantity:

**Lemma 4.2.** Given a configuration $C \in \overline{\mathcal{D}}$ with $z, \bar{z} \notin (-\infty, 0]$, $N_t$ is independent of the choice of the path. Therefore, we can write $N_t$ as $N_t(C)$.

*Proof.* Suppose we have a path $\gamma$ under condition (55) and $\gamma(1) = C$. Under convention (32), the path $\gamma$ uniquely determines the paths of $z, \bar{z}, \rho, \bar{\rho}, \chi, \bar{\chi}$. By (18) and (27), we have

$$
\begin{aligned}
z \in (-\infty, 0) &\iff \rho \in (-1, 0) \\
&\iff \mathrm{Re}(\chi) = (2k+1)\pi \text{ for some } k \in \mathbb{Z},
\end{aligned}
\tag{60}
$$

which implies that the final point of $\chi(s)$ contains the information about $n_t$:

$$
(2n_t - 1)\pi < \mathrm{Re}\left(\chi(1)\right) < (2n_t + 1)\pi.
\tag{61}
$$

Analogously we have

$$
(2\bar{n}_t - 1)\pi < \mathrm{Re}\left(\bar{\chi}(1)\right) < (2\bar{n}_t + 1)\pi.
\tag{62}
$$

Now we pick another path $\gamma'$ under condition (55) and $\gamma'(1) = C$. We let $\chi', \bar{\chi}', n_t', \bar{n}_t', N_t'$ denote the corresponding variables of the path $\gamma'$. By lemma 3.3, we have

$$
\chi(1) + \bar{\chi}(1) = \chi'(1) + \bar{\chi}'(1).
\tag{63}
$$

Since $\rho, \bar{\rho}$ at $C$ at most interchange with each other, the relation (63) implies that there only two possibilities:

1. $\chi(1) = \chi'(1) + 2k\pi$, $\bar{\chi}(1) = \bar{\chi}'(1) - 2k\pi$ for some $k \in \mathbb{Z}$.

2. $\chi(1) = \bar{\chi}'(1) + 2k\pi$, $\bar{\chi}(1) = \chi'(1) - 2k\pi$ for some $k \in \mathbb{Z}$.

which, by (61) and (62), are equivalent to

1. $n_t = n_t' + k$, $\bar{n}_t = \bar{n}_t' - k$ for some $k \in \mathbb{Z}$.

2. $n_t = \bar{n}_t' + k$, $\bar{n}_t = n_t' - k$ for some $k \in \mathbb{Z}$.

Thus we have $N_t(\gamma) = N_t(\gamma')$. $\qquad\square$

Suppose $C$ is a configuration in $\mathcal{D} \cup \mathcal{D}_L$ with $z, \bar{z} \notin (-\infty, 0)$ and $N_t = 0$. By choosing an arbitrary path $\gamma$ with conditions (55) and $\gamma(1) = C$, we get the paths $\chi(s), \bar{\chi}(s)$ along $\gamma$. We define a pair of new variables $\tilde{\chi}, \tilde{\bar{\chi}}$ by

$$
\tilde{\chi} = \chi(1) - 2n_t\pi, \quad \tilde{\bar{\chi}} = \bar{\chi}(1) + 2n_t\pi.
\tag{64}
$$

Since $N_t = 0$ (which implies $n_t = -\bar{n}_t$), by (61) and (62), the construction (64) gives

$$
-\pi < \mathrm{Re}\left(\tilde{\chi}\right) < \pi, \quad -\pi < \mathrm{Re}\left(\tilde{\bar{\chi}}\right) < \pi.
\tag{65}
$$

We have the following lemma.

**Lemma 4.3.** The following maps

$$\chi \mapsto \rho = e^{i\chi} \mapsto z = \frac{4\rho}{(1+\rho)^2} = \frac{1}{\cos^2 \frac{\chi}{2}} \tag{66}$$

are biholomorphic maps from $\{\chi \in \mathbb{C} | -\pi < \mathrm{Re}\,(\chi) < \pi,\; \mathrm{Im}\chi > 0\}$ to $\{\rho \in \mathbb{C} | |\rho| < 1,\; \rho \neq (-1,0]\}$, then to the double-cut plane $\{z \in \mathbb{C} | z \notin (-\infty,0] \cup [1,+\infty)\}$.

*Proof.* For the map $\chi \mapsto \rho$, since $-\pi < \mathrm{Re}\,(\chi) < \pi$, its image in the $\rho$-space does not contain $(-\infty,0)$, hence does not contain curves which go around 0. So the inverse $\chi = -i \ln \rho$ exists. The constraint $\mathrm{Im}\chi > 0$ is equivalent to $0 < |\rho| < 1$.

The map $\rho \mapsto z$ is known to be a biholomorphic map from the open unit disc to $\mathbb{C}\backslash[1,+\infty)$ [13]. One can show by direct computation that $\rho \in (-1,0]$ is equivalent to $z \in (-\infty,0]$. $\qquad\square$

Since the double-cut plane is preserved under the map $z \to 1 - z$, by lemma 4.3 we define a pair of t-channel variables $\tilde{\chi}_t, \tilde{\bar{\chi}}_t$ by

$$\begin{aligned}
\tilde{\chi} \mapsto \tilde{z} \mapsto \tilde{z}_t = 1 - \tilde{z} \mapsto \tilde{\chi}_t, \\
\tilde{\bar{\chi}} \mapsto \tilde{\bar{z}} \mapsto \tilde{\bar{z}}_t = 1 - \tilde{\bar{z}} \mapsto \tilde{\bar{\chi}}_t,
\end{aligned} \tag{67}$$

where the maps $\tilde{\chi} \mapsto \tilde{z}$, $\tilde{\bar{\chi}} \mapsto \tilde{\bar{z}}$ are the same as (66), and the maps $\tilde{z}_t \mapsto \tilde{\chi}_t$, $\tilde{\bar{z}}_t \mapsto \tilde{\bar{\chi}}_t$ are the inverse of (66).

Since $\mathrm{Im}\tilde{\chi}, \mathrm{Im}\tilde{\bar{\chi}} > 0$ for all configurations in $\mathcal{D}$, above we defined the (path dependent) variables $\tilde{\chi}_t, \tilde{\bar{\chi}}_t$ for configurations in $\mathcal{D}$ with the constraints $z, \bar{z} \notin (-\infty,0)$ and $N_t = 0$. In fact such definition can be extended to Lorentzian configurations in $\mathcal{D}_L$ with the same constraints. This is because any configuration $C_L$ with $z, \bar{z} \notin (-\infty,0)$ and $N_t = 0$ can be approached by configurations in $\mathcal{D}$ with the same constraints, and then $\tilde{\chi}_t, \tilde{\bar{\chi}}_t$ at $C_L$ are defined by continuity.

Note that if $z$ nor $\bar{z}$ do not cross $(-\infty,0)$ at all, then $|\rho_t|, |\bar{\rho}_t| < 1$ along the whole path, and the t-channel OPE is guaranteed to converge. The criterion we give is more general in that it allows some crossings. Let us prove that this more general criterion is indeed sufficient.

**Theorem 4.4.** (*t-channel OPE convergence*) Given a Lorentzian configuration $C_L \in \mathcal{D}_L$. If the variables $z, \bar{z}$ of $C_L$ do not belong to $(-\infty,0)$, and furthermore if $N_t(C_L) = 0$, then the Lorentzian four-point function $G_4$ is analytic at $C_L$ and is given by the formula

$$G_4(C_L) = \frac{g\left(\tilde{\chi}_t, \tilde{\bar{\chi}}_t\right)}{\left[x_{23}^2 x_{14}^2\right]^\Delta}. \tag{68}$$

Here $g$ is the same function as described in section 3.3.2, and the variables $\tilde{\chi}_t, \tilde{\bar{\chi}}_t$ are defined by the algorithm in (64) and (67). The function $g(\tilde{\chi}_t, \tilde{\bar{\chi}}_t)$ can be computed by the convergent series expansion (22).

Before the proof of theorem 4.4, we introduce the following lemma:

**Lemma 4.5.** Given a Lorentzian configuration $C \in \mathcal{D}$. If the variables $z, \bar{z}$ of $C_L$ do not belong to $(-\infty,0)$, and furthermore if $N_t(C) = 0$, then we have

$$\frac{g\left(\chi, \bar{\chi}\right)}{\left[x_{12}^2 x_{34}^2\right]^\Delta} = \frac{g\left(\tilde{\chi}_t, \tilde{\bar{\chi}}_t\right)}{\left[x_{23}^2 x_{14}^2\right]^\Delta} \tag{69}$$

We would like to postpone the proof of lemma 4.5. Let us first see how lemma 4.5 implies theorem 4.4.

Suppose we have a Lorentzian configuration $C_L$ which satisfies the conditions of theorem 4.4. Since $C_L \in \overline{\mathcal{D}}$, we can approach $C_L$ by a sequence of configurations $\{C_n\}$ in $\mathcal{D}$ such that $C_n$ satisfies $z, \bar{z} \notin (-\infty, 0)$ and $N_t(C_n) = 0$. By construction $\text{Im}\tilde{\chi}_t, \text{Im}\tilde{\bar{\chi}}_t > 0$ for $C_L$ and all $C_n$ in the sequence, so $g(\tilde{\chi}_t, \tilde{\bar{\chi}}_t)$ can be computed by the convergent series expansion (22). We choose $\tilde{\chi}_t(C_n), \tilde{\bar{\chi}}_t(C_n)$ such that they form two sequences which approach $\tilde{\chi}_t(C_L), \tilde{\bar{\chi}}_t(C_L)$, then by continuity, the limit of $g(\tilde{\chi}_t(C_n), \tilde{\bar{\chi}}_t(C_n))$ is exactly $g(\tilde{\chi}_t(C_L), \tilde{\bar{\chi}}_t(C_L))$.[12] By lemma 4.5, we have eq. (69) for $C_n$. Note that the LHS of (69) is the four-point function in the s-channel expansion, thus we have

$$G_4(C_L) = \lim_{n \to \infty} G_4(C_n) = \lim_{n \to \infty} \frac{g(\tilde{\chi}_t, \tilde{\bar{\chi}}_t)}{[x_{23}^2 x_{14}^2]^\Delta}\bigg|_{C_n} = \frac{g(\tilde{\chi}_t, \tilde{\bar{\chi}}_t)}{[x_{23}^2 x_{14}^2]^\Delta}\bigg|_{C_L}. \tag{70}$$

So we finish the proof of theorem 4.4. We would like to make two comments. First, theorem 4.4 covers the case when s-channel expansion is not convergent. For this case we have $z$ or $\bar{z} \in (1, +\infty)$. To compute $G_4$ it is important to know whether $\text{Arg}(1 - \tilde{z})$ or $\text{Arg}(1 - \tilde{\bar{z}})$ is equal to $\pi$ or $-\pi$. A crucial point is that the information about these phases are contained in $\tilde{\chi}_t$ and $\tilde{\bar{\chi}}_t$.[13] Second, eq. (68) indeed corresponds to the t-channel expansion because each term in the series expansion (22) of $g(\tilde{\chi}_t, \tilde{\bar{\chi}}_t)$ corresponds to a state which appears in the $\phi(x_2)\phi(x_3)$ OPE.

It remains to prove lemma 4.5.

*Proof.* We have

$$\frac{g(\chi, \bar{\chi})}{[x_{12}^2 x_{34}^2]^\Delta} = \frac{g(\tilde{\chi}, \tilde{\bar{\chi}})}{[x_{12}^2 x_{34}^2]^\Delta}$$

$$= \frac{1}{[x_{12}^2 x_{34}^2]^\Delta} \times \left[\frac{\tilde{z}\tilde{\bar{z}}}{(1 - \tilde{z})(1 - \tilde{\bar{z}})}\right]^\Delta g(\tilde{\chi}_t, \tilde{\bar{\chi}}_t)$$

$$= \frac{1}{[x_{12}^2 x_{34}^2]^\Delta} \times \left[\frac{z\bar{z}}{(1 - z)(1 - \bar{z})}\right]^\Delta g(\tilde{\chi}_t, \tilde{\bar{\chi}}_t) \tag{71}$$

$$= \frac{1}{[x_{12}^2 x_{34}^2]^\Delta} \times \left[\frac{u}{v}\right]^\Delta g(\tilde{\chi}_t, \tilde{\bar{\chi}}_t)$$

$$= \frac{g(\tilde{\chi}_t, \tilde{\bar{\chi}}_t)}{[x_{23}^2 x_{14}^2]^\Delta}$$

The first equality is a consequence of eq. (29). The second equality follows from the crossing symmetry

$$g(z, \bar{z}) = \left[\frac{z\bar{z}}{(1 - z)(1 - \bar{z})}\right]^\Delta g(1 - z, 1 - \bar{z}), \quad z, \bar{z} \in \mathbb{C}\backslash(-\infty, 0] \cup [0, +\infty). \tag{72}$$

Here we also use the fact that both s- and t-channel expansions are convergent if $z, \bar{z}$ are in the double-cut plane, and in the same branch as the Euclidean case.

---

[12]For this claim we choose a path $\gamma(s)$ with $\gamma(1) = C_L$ and let the sequence $\{C_n\}$ be along the path $\gamma$. Let the sequences of $\chi(C_n), \bar{\chi}(C_n)$ also be along the path, then it is natural to see that the sequences of $\tilde{\chi}_t(C_n), \tilde{\bar{\chi}}_t(C_n)$ have the limits $\tilde{\chi}_t(C_L), \tilde{\bar{\chi}}_t(C_L)$.

[13]Since $\tilde{\rho}_t = e^{i\tilde{\chi}_t}$, we have $\text{Arg}(\tilde{\rho}_t) = \text{Re}\tilde{\chi}_t$. If $z = \bar{z} \in (1, +\infty)$, then we have $\text{Arg}(1 - \tilde{z}) = \text{Re}\tilde{\chi}_t$. The argument is similar for $\bar{z}$.

For the third equality, we recall our definition of $\tilde{\chi}, \tilde{\bar{\chi}}$ in (64): the variables $z, \bar{z}$ acquire extra phases and $1 - z, 1 - \bar{z}$ do not go around 0. In other words, we have

$$\tilde{z} = e^{-2n_t\pi i} z, \quad \tilde{\bar{z}} = e^{2n_t\pi i}\bar{z},$$
$$1 - \tilde{z} = 1 - z, \quad 1 - \tilde{\bar{z}} = 1 - \bar{z}. \tag{73}$$

The remaining steps in (71) are trivial. $\qquad\square$

The criterion of u-channel convergence is similar to t-channel. Given a configuration $C_L$ with $z, \bar{z} \notin (0, 1)$ and $N_u = 0$. We choose a path $\gamma$ to get $\chi(1), \bar{\chi}(1)$, then the u-channel versions of (61) and (62) are given by

$$-2n_u\pi < \operatorname{Re}(\chi(1)) < -2n_u\pi + 2\pi,$$
$$-2\bar{n}_u\pi - 2\pi < \operatorname{Re}(\bar{\chi}(1)) < -2\bar{n}_u\pi. \tag{74}$$

The u-channel variables $\tilde{\chi}_u, \tilde{\bar{\chi}}_u$ are defined by following algorithm, which is analogous to (67):

$$\tilde{\chi} = \chi(1) + 2n_u\pi \mapsto \tilde{z} \mapsto \tilde{z}_u = \frac{1}{\tilde{z}} \mapsto \tilde{\chi}_u,$$
$$\tilde{\bar{\chi}} = \bar{\chi}(1) - 2n_u\pi \mapsto \tilde{\bar{z}} \mapsto \tilde{\bar{z}}_u = \frac{1}{\tilde{\bar{z}}} \mapsto \tilde{\bar{\chi}}_u. \tag{75}$$

We give the u-channel criterion without proof.

**Theorem 4.6.** (*u-channel OPE convergence*) Given a Lorentzian configuration $C_L \in \mathcal{D}_L$. If the variables $z, \bar{z}$ of $C_L$ do not belong to $(0, 1)$, and furthermore if $N_u(C_L) = 0$, then the Lorentzian four-point function $G_4$ is analytic at $C_L$ and is given by the formula

$$G_4(C_L) = \frac{g\left(\tilde{\chi}_u, \tilde{\bar{\chi}}_u\right)}{\left[x_{13}^2 x_{24}^2\right]^\Delta}. \tag{76}$$

Here $g$ is the same function as described in section 3.3.2, and the variables $\tilde{\chi}_u, \tilde{\bar{\chi}}_u$ are defined by the algorithm (75). The function $g(\tilde{\chi}_u, \tilde{\bar{\chi}}_u)$ can be computed by the convergent series expansion (22).

Unlike the s-channel case, even if we only want to check the convergence properties of t- and u-channel expansions, we have to choose a path to compute $N_t$ and $N_u$.

Before finishing this subsection, we want to remark that actually the condition (55) of the path $\gamma$ can be relaxed in the way that $\gamma$ is allowed to touch $\Gamma$:

$$\gamma : [0, 1] \longrightarrow \overline{\mathcal{D}},$$
$$\gamma(0) \in \mathcal{D}_E \backslash \Gamma, \tag{77}$$
$$\gamma(1) \in \mathcal{D}_L.$$

Suppose we have a path $\gamma$ which intersects with $\Gamma$. Let $\gamma(s_*) \in \Gamma$ be the first intersection point. At $s_*$ we have $z(s_*) = \bar{z}(s_*)$, then $z(s), \bar{z}(s)$ become indistinguishable for $s > s_*$, so the quantities $n_t, \bar{n}_t, n_u, \bar{n}_u, \chi, \bar{\chi}$ are not well defined for $\gamma$. However, by manually choosing $z, \bar{z}$ after each intersection, we still get two curves $z(s), \bar{z}(s)$: they may not be smooth at intersection points, but they are still continuous. By this trick we get $n_t, \bar{n}_t, n_u, \bar{n}_u, \chi, \bar{\chi}$, so that we are able to compute $N_t, N_u$ and the four-point function. On the other hand, we can always deform $\gamma$ to a path $\gamma'$, such that $\gamma'$ has the same start and final points as $\gamma$ but $\gamma'$ does not intersect with $\Gamma$. By doing proper deformation, we can make $\gamma'$ have the same $n_t, \bar{n}_t, n_u, \bar{n}_u, \chi, \bar{\chi}$ as selected on $\gamma$. Therefore, our manual selection will give the correct OPE convergence properties and the correct value of the four-point function.

## 4.4 What happens if there is no convergent OPE channel?

We want to make a comment that theorem 4.1, 4.4 and 4.6 give sufficient conditions for OPE convergence. For a Lorentzian configuration $C_L$ which is not a light-cone singularity and which does not satisfy the conditions in these theorems, it does not mean that $G_4$ cannot be a function at $C_L$. It just means that for general CFT, we are not able to use the radial coordinates $\rho, \bar{\rho}$ ($\rho_t, \bar{\rho}_t,$ $\rho_u, \bar{\rho}_u$) and the expansion (22) to prove the analyticity of $G_4$ at $C_L$. The four-point function still has a chance to be analytic at $C_L$. For example, the four-point function of generalized free fields has analytic continuation to the whole Lorentzian region except for the light-cone singularities.

An interesting related open question is: can we relax the conditions in theorem 4.1, 4.4 and 4.6?

### 4.4.1 s-channel condition

In theorem 4.1, we only assume the condition $z, \bar{z} \notin (1, +\infty)$ (equivalently, $|\rho|, |\bar{\rho}| < 1$). The Lorentzian configurations which violate this condition has $|\rho| = 1$ or $|\bar{\rho}| = 1$, then the proof of theorem 4.1 fails because in the proof we used the fact that the series expansion (22) is absolutely convergent when $|\rho|, |\bar{\rho}| < 1$.

We are interested in the Lorentzian configurations where the s-channel expansion is convergent for all unitary CFTs. For configurations with $|\rho| = 1$ or $|\bar{\rho}| = 1$, we may exhibit an explicit CFT four-point function, for which the s-channel expansion is divergent (then such configurations are ruled out). The generalized free field (GFF) theory is such an example. The GFF four-point function of identical scalar operators (with scaling dimension $\Delta$) is defined by

$$(G_4)_{GFF}(x_1, x_2, x_3, x_4) = \frac{1}{\left[x_{12}^2 x_{34}^2\right]^\Delta} + \frac{1}{\left[x_{23}^2 x_{14}^2\right]^\Delta} + \frac{1}{\left[x_{13}^2 x_{24}^2\right]^\Delta}. \tag{78}$$

By (4), the conformal invariant part of $(G_4)_{GFF}$ is given by

$$g_{GFF}(\rho, \bar{\rho}) = 1 + \left(\frac{16\rho\bar{\rho}}{(1+\rho)^2(1+\bar{\rho})^2}\right)^\Delta + \left(\frac{16\rho\bar{\rho}}{(1-\rho)^2(1-\bar{\rho})^2}\right)^\Delta. \tag{79}$$

It has the series expansion

$$g_{GFF}(\rho, \bar{\rho}) = 1 + (16\rho\bar{\rho})^\Delta \sum_{m,n=0}^\infty \frac{(1+(-1)^{m+n})\,\Gamma(\Delta+m)\Gamma(\Delta+n)}{m!n!\Gamma(\Delta)^2} \rho^m \bar{\rho}^n \tag{80}$$

which diverges when $|\rho| = 1$ or $|\bar{\rho}| = 1$. It follows that theorem 4.1 cannot be extended to configurations with $|\rho| = 1$ or $|\bar{\rho}| = 1$ without extra assumptions on the theory. One such extra assumption will be mentioned in section 5.3.2 (locality of 2d CFT).

### 4.4.2 t- and u-channel conditions

In theorem 4.4, we assumed two conditions: $z, \bar{z} \notin (-\infty, 0)$ and $N_t = 0$. For Lorentzian configurations which violate the first condition, (analogously to the s-channel case) we can use GFF to conclude that these configurations do not have convergent t-channel expansion for some unitary CFTs.

Let us explain more about our motivation for assuming $N_t = 0$. By the s-channel series expansion (22) and crossing symmetry (72), the function $g(\chi_t, \bar{\chi}_t)$ has analytic continuation to the universal covering of the domain

$$\begin{aligned} -\pi < \mathrm{Re}\,(\chi_t) < \pi, \quad \chi_t \neq 0 \\ -\pi < \mathrm{Re}\,(\bar{\chi}_t) < \pi, \quad \bar{\chi}_t \neq 0 \end{aligned} \tag{81}$$

The series expansion of $g(\chi_t, \bar{\chi}_t)$ is absolutely convergent in the region where

$$
\begin{aligned}
-\pi < \mathrm{Re}\,(\chi_t) < \pi, \quad 0 < \mathrm{Arg}\,(\chi_t) < \pi \\
-\pi < \mathrm{Re}\,(\bar{\chi}_t) < \pi, \quad 0 < \mathrm{Arg}\,(\bar{\chi}_t) < \pi
\end{aligned}
\tag{82}
$$

Suppose we have a configuration $C_L$ with $N_t \neq 0$. By choosing a path $\gamma$, we compute the paths $\chi_t(s), \bar{\chi}_t(s)$, and then determine the final points $\chi_t(1), \bar{\chi}_t(1)$. When the path $z(s)$ crosses $(-\infty, 0)$ from above, $\chi_t(s)$ either crosses $(-\pi, 0)$ from above, or crosses $(0, \pi)$ from below. Since the start points $\chi_t(0), \bar{\chi}_t(0)$ are in the region (82), we have

$$
n_t \pi < \mathrm{Arg}\,(\chi_t) < (n_t + 1)\pi, \quad \bar{n}_t \pi < \mathrm{Arg}\,(\bar{\chi}_t) < (\bar{n}_t + 1)\,\pi,
\tag{83}
$$

which are the t-channel versions of (61) and (62). We see that $\chi_t(1)e^{-n_t\pi i}, \chi_t e^{-\bar{n}_t\pi i}$ are in the region (82). The property (29) says that we have[14]

$$
g\left(\chi_t(1), \bar{\chi}_t(1)\right) = g\left(\chi_t(1)e^{\pi i}, \bar{\chi}_t(1)e^{-\pi i}\right).
\tag{84}
$$

However, if $N_t \neq 0$, then $n_t \neq -\bar{n}_t$ for any path $\gamma$. So in this case we cannot use (84) to move $\chi_t(1), \bar{\chi}_t(1)$ to the region (82). This is where the proof of theorem 4.4 fails for $N_t \neq 0$.

The arguments for theorem 4.6 are similar.

# 5 Classifying the Lorentzian configurations

In the previous section we gave the criteria of convergence properties of OPE in various channels for Lorentzian CFT four-point functions. These criteria say that given a Lorentzian configuration $C_L$, one can just start with an arbitrary Euclidean configuration in $\mathcal{D}_E \backslash \Gamma$ and choose an arbitrary path towards $C_L$, then decide if the conditions in theorem 4.1, 4.4 and 4.6 hold or not by watching the $z, \bar{z}$-curves (in theorem 4.1 one does not even have to choose a path).

However, it would be frustrating if we have to check the analytic continuation curves for all Lorentzian configurations in $\mathcal{D}_L$ (recall definition (49)). We expect that these Lorentzian configurations can be classified such that for each class it suffices to choose one representative configuration to see if various OPE channels converge or not. There are two natural classification methods, one according to the range of $z$ and $\bar{z}$, the other according to the causal orderings. We will show that combining these two methods leads to a complete classification for the convergence properties of Lorentzian CFT four-point functions.

## 5.1 $z, \bar{z}$ of Lorentzian configurations

For all Lorentzian configurations, since $x_{ij}^2$ are real, the cross-ratios $u, v$ are also real. By (8), there are only two possibilities for $z, \bar{z}$:

1. $z, \bar{z}$ are independent real variables.

2. $z, \bar{z}$ are complex conjugate to each other.

---

[14]Recall the map $\chi \mapsto \chi_t$ in (67), the transformation $\chi \to \chi + 2\pi$ corresponds to $\chi_t \to \chi_t e^{i\pi}$.

In addition, we have already excluded light-cone singularities in $\mathcal{D}_L$ (recall definition (49)), so the configurations in $\mathcal{D}_L$ have $z, \bar{z} \neq 0, 1, \infty$. According to the range of the $z, \bar{z}$ variables, we divide $\mathcal{D}_L$ into four classes:

$$\mathcal{D}_L = \mathrm{S} \sqcup \mathrm{T} \sqcup \mathrm{U} \sqcup \mathrm{E}, \tag{85}$$

where the classes are defined as follows.

- Class S: configurations with $0 < z < 1$, $\bar{z} < 0$ or $z < 0$, $0 < \bar{z} < 1$.

- Class T: configurations with $z > 1$, $0 < \bar{z} < 1$ or $0 < z < 1$, $\bar{z} > 1$.

- Class U: configurations with $z > 1$, $\bar{z} < 0$ or $z < 0$, $\bar{z} > 1$.

- Class E: configurations with $z, \bar{z} < 0$ or $0 < z, \bar{z} < 1$ or $z, \bar{z} > 1$ or $z^* = \bar{z}$.

We use the name "S" (resp. "T", "U") because it corresponds to the configurations where only the s-channel (resp. t-channel, u-channel) expansion has a chance to converge. The name "E" means "Euclidean", since the variables $z, \bar{z}$ in class E can be realized by the configurations with totally space-like separation. In addition, we divide the class E into four subclasses:

$$\mathrm{E} = \mathrm{E}_{\mathrm{su}} \sqcup \mathrm{E}_{\mathrm{st}} \sqcup \mathrm{E}_{\mathrm{tu}} \sqcup \mathrm{E}_{\mathrm{stu}}, \tag{86}$$

where the subclasses are defined as follows.

- Subclass $\mathrm{E}_{\mathrm{su}}$: configurations with $z, \bar{z} < 0$.

- Subclass $\mathrm{E}_{\mathrm{st}}$: configurations with $0 < z, \bar{z} < 1$.

- Subclass $\mathrm{E}_{\mathrm{tu}}$: configurations with $z, \bar{z} > 1$.

- Subclass $\mathrm{E}_{\mathrm{stu}}$: configurations with $z^* = \bar{z}$ not real.

The subscripts in above names indicate the possible convergent channels. Figure 5.1 shows the range of $(z, \bar{z})$ pair corresponding to each class/subclass. Let $P(\mathcal{C})$ denote the subset of $(z, \bar{z})$ pairs corresponding to class/subclass $\mathcal{C}$. Under identification $(z, \bar{z}) \sim (\bar{z}, z)$, $P(\mathcal{C})$ are connected subsets of $\mathbb{C}/\mathbb{Z}_2$. $P(\mathrm{S}), P(\mathrm{T}), P(\mathrm{U}), P(\mathrm{E})$ are disconnected from each other, but $P(\mathrm{E}_{\mathrm{su}})$, $P(\mathrm{E}_{\mathrm{st}})$ and $P(\mathrm{E}_{\mathrm{tu}})$ are connected to $P(\mathrm{E}_{\mathrm{stu}})$ (also note that $P(\mathrm{E}_{\mathrm{su}})$, $P(\mathrm{E}_{\mathrm{st}})$ and $P(\mathrm{E}_{\mathrm{tu}})$ are disconnected from each other).

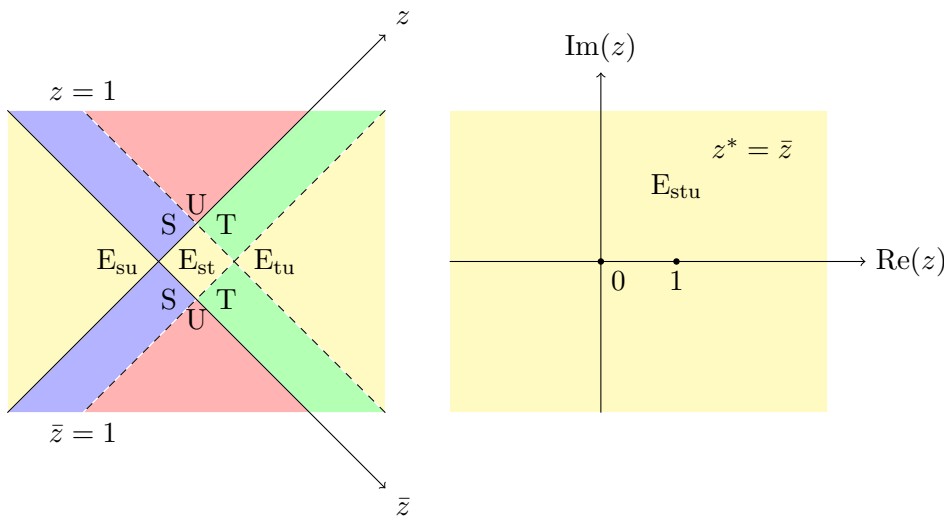

Figure 5.1: The corresponding range of $(z, \bar{z})$ pair of each class/subclass.

For each class/subclass, we immediately get some information about OPE convergence properties by theorem 4.1, 4.4 and 4.6 (see table 1).

| class/subclass | s-channel | t-channel | u-channel |
|:---:|:---:|:---:|:---:|
| S | ✓ | ✗ | ✗ |
| T | ✗ | | ✗ |
| U | ✗ | ✗ | |
| $E_{st}$ | ✓ | | ✗ |
| $E_{su}$ | ✓ | ✗ | |
| $E_{tu}$ | ✗ | | |
| $E_{stu}$ | ✓ | | |

Table 1: OPE convergence properties of classes/subclasses

In table 1, the check mark means that the sufficient conditions in theorem 4.1 or 4.4 or 4.6 holds, hence the corresponding channel is convergent. The cross mark means that the sufficient conditions do not hold, we cannot conclude that the corresponding channel is convergent or not (basically because one or both $\rho, \bar{\rho}$ variables are on the unit circles). The blank means that there is room for convergence but we need to check $N_t, N_u$ conditions.

## 5.2 Causal orderings

In Minkowski space $\mathbb{R}^{d-1,1}$, causal ordering is a binary relation between two arbitrary points. Let $x_1 = (it_1, \mathbf{x}_1)$ and $x_2 = (it_2, \mathbf{x}_2)$ be two points in $\mathbb{R}^{d-1,1}$,[15] we say $x_1 \to x_2$ if $x_2$ is in the open forward light-cone of $x_1$, or equivalently, $t_2 - t_1 > |\mathbf{x}_1 - \mathbf{x}_2|$.

By the triangle inequality, the causal ordering is transitive: if $x_1 \to x_2$ and $x_2 \to x_3$, then $x_1 \to x_3$.

Causal orderings are preserved by translations, Lorentz transformations and dilatations. But special conformal transformations may violate causal orderings. Given a pair of time-like separated points $x_i, x_j$ in $\mathbb{R}^{d-1,1}$, there exists a special transformation such that the images $x_i', x_j'$ are space-like separated [44].

By "the causal ordering of a configuration $C = (x_1, x_2, x_3, x_4)$", we will mean the directed graph $(V, E)$, where $V = \{1, 2, 3, 4\}$ is the set of indices and $E = \{(ij)\}$ is the set of arrows $i \to j$ encoding the causal orderings $x_i \to x_j$. For example, the causal ordering of the configuration

$$
\begin{aligned}
x_1 &= (0, 0, \ldots, 0), \\
x_2 &= (i, 0, \ldots, 0), \\
x_3 &= (2i, 0, \ldots, 0), \\
x_4 &= (3i, 0, \ldots, 0),
\end{aligned}
\tag{87}
$$

_______________

[15]Since in this work our discussions start from the Euclidean signature, we use the Euclidean coordinates $x = (\epsilon + it, \mathbf{x})$. The Euclidean points correspond to $t = 0$ and the Lorentzian points correspond to $\epsilon = 0$.

is given by

$$
\begin{array}{ccc}
1 & \longrightarrow & 2 \\
\downarrow & \times & \downarrow \\
4 & \longleftarrow & 3
\end{array}
\tag{88}
$$

Since causal ordering is transitive, some arrows in the graph (88) are redundant and we will drop them. E.g. the graph

$$
1 \rightarrow 2 \rightarrow 3 \rightarrow 4
\tag{89}
$$

represents the same causal ordering as (88). For simplicity, we will use the graphic notation with the least number of arrows like (89).

## 5.3 Classifying convergent OPE channels

We decompose the set $\mathcal{D}_L$ according to the causal orderings of the configurations:

$$
\mathcal{D}_L = \bigsqcup_\alpha \mathcal{D}_L^\alpha,
\tag{90}
$$

where each $\mathcal{D}_L^\alpha$ is the set of configurations with the same causal ordering, labelled by the index $\alpha$.

### 5.3.1 Case $d \geq 3$

In $d \geq 3$, each $\mathcal{D}_L^\alpha$ in (90) is a connected component of $\mathcal{D}_L$. it is not hard to see that different $\mathcal{D}_L^\alpha$ are disconnected to each other. The proof that each $\mathcal{D}_L^\alpha$ is connected is given in appendix B.

Since $\mathcal{D}_L^\alpha$ is connected, with the identification $(z, \bar{z}) \sim (\bar{z}, z)$, the set of corresponding $(z, \bar{z})$ pairs is a connected subset of $\mathbb{C}^2 / \mathbb{Z}_2$. Recalling our classification in section 5.1, we conclude that

**Lemma 5.1.** For $d \geq 3$, all configurations with the same causal ordering belong to the same class S, T, U, E (see section 5.1).

By the lemma, we can assign class S, T, U and E to each causal ordering of the configurations. In addition, if $\mathcal{D}_L^\alpha$ is in class E, we subdivide $\mathcal{D}_L^\alpha$ according to the subclasses of class E. We summarize these relations in figure 5.2.

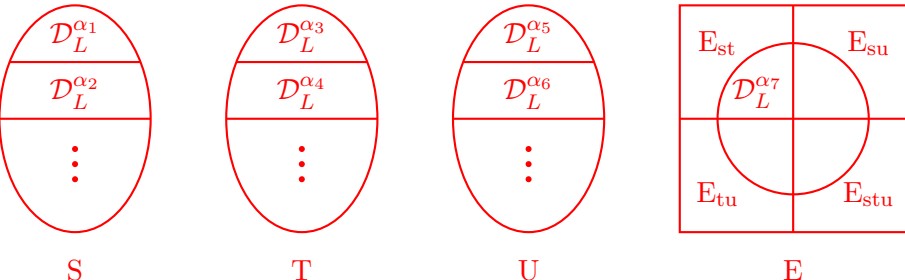

Figure 5.2: The class S, T, U and E are subdivided according causal orderings. For each $\mathcal{D}_L^\alpha$ in class E, $\mathcal{D}_L^\alpha$ is subdivided according to subclasses.

Now we are ready to state the classification of convergent OPE channels for Lorentzian CFT four-point functions.

**Theorem 5.2.** Let $G_4^L$ be the Lorentzian four-point function which is defined by the Wick rotation (2) from a Euclidean unitary CFT in $d \geq 3$. Let $\alpha$ be a causal ordering and let $\mathcal{D}_L^\alpha$ be the set of all configurations with this causal ordering.

- If $\mathcal{D}_L^\alpha$ is in class S, then all configurations in $\mathcal{D}_L^\alpha$ only have convergent s-channel expansion for $G_4^L$.

- If $\mathcal{D}_L^\alpha$ is in class T, then all configurations in $\mathcal{D}_L^\alpha$ have the same $N_t$.

- If $\mathcal{D}_L^\alpha$ is in class U, then all configurations in $\mathcal{D}_L^\alpha$ have the same $N_u$.

- If $\mathcal{D}_L^\alpha$ is in class E, then

  - All configurations in $\mathcal{D}_L^\alpha \cap \mathrm{E}_{\mathrm{st}}$ have the convergent s-channel expansion and the same $N_t$.
  - All configurations in $\mathcal{D}_L^\alpha \cap \mathrm{E}_{\mathrm{su}}$ have the convergent s-channel expansion and the same $N_u$.
  - All configurations in $\mathcal{D}_L^\alpha \cap \mathrm{E}_{\mathrm{tu}}$ have the same $N_t, N_u$.
  - All configurations in $\mathcal{D}_L^\alpha \cap \mathrm{E}_{\mathrm{stu}}$ have the convergent s-channel expansion and the same $N_t, N_u$.

*Proof.* Let us check the conclusions case by case.

Case 1: $\mathcal{D}_L^\alpha$ is in class S.

The s-channel convergence follows from theorem 4.1. For other cases, the s-channel arguments are the same, and we will only focus on $N_t$ and $N_u$.

Case 2: $\mathcal{D}_L^\alpha$ is in class T.

It remains to show that $N_t$ is a constant in $\mathcal{D}_L^\alpha$. For any $C_L, C_L' \in \mathcal{D}_L^\alpha$, since $\mathcal{D}_L^\alpha$ is connected, there exists a path $\gamma_1$ which connects $C_L$ and $C_L'$:

$$
\begin{aligned}
\gamma_1 : \ [0,1] \ &\longrightarrow \ \mathcal{D}_L^\alpha, \\
\gamma_1(0) = C_L, \quad &\gamma_1(1) = C_L'.
\end{aligned}
\tag{91}
$$

Since $\gamma_1(s)$ are always configurations in class T, the corresponding $z, \bar{z}$ never touch the interval $(-\infty, 0)$. So $n_t(\gamma_1) = \bar{n}_t(\gamma_1) = 0$, which implies $N_t(\gamma_1) = 0$. On the other hand, given a path $\gamma_2$ from $\mathcal{D}_E \backslash \Gamma$ to $C_L$, we get a path from $\mathcal{D}_E \backslash \Gamma$ to $C_L'$ by connecting $\gamma_1$ and $\gamma_2$. So we have

$$
N_t(C_L') = N_t(\gamma_1) + N_t(\gamma_2) = N_t(C_L).
\tag{92}
$$

In other words, $N_t$ is a constant in $\mathcal{D}_L^\alpha$.

Case 3: $\mathcal{D}_L^\alpha$ is in class U.

It remains to show that $N_u$ is a constant in $\mathcal{D}_L^\alpha$. The argument is similar to case 2.

Case 4: $\mathcal{D}_L^\alpha$ is in class E.

Suppose $C_L, C_L'$ are two configurations in $\mathcal{D}_L^\alpha \cap \mathrm{E}_{\mathrm{st}}$. It remains to show that $N_t(C_L) = N_t(C_L')$. Analogously to case 2, there exists a path $\gamma_1$ satisfying the condition (91), and it suffices to show that $N_t(\gamma_1) = 0$. Here it is different from case 2 because $\gamma_1(s)$ may go through the other subclasses of the class E, and the curves of $z, \bar{z}$ may touch the interval $(-\infty, 0)$. In class E, the curves $z(s), \bar{z}(s)$ touch the interval $(-\infty, 0)$ only when $\gamma(s)$ enters the subclass $\mathrm{E}_{\mathrm{su}}$. However, $\gamma(s)$, which starts from $\mathrm{E}_{\mathrm{st}}$, must go through $\mathrm{E}_{\mathrm{stu}}$ before

entering $E_{su}$. When $\gamma(s)$ leaves $E_{su}$, it must go through $E_{stu}$ again. Since in $E_{stu}$, the variables $z, \bar{z}$ are complex conjugate to each other, the curves of $z, \bar{z}$ must cross $(-\infty, 0)$ from opposite directions, see figure 5.3.

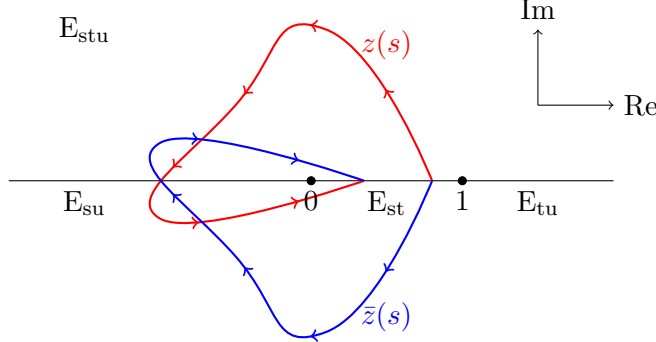

Figure 5.3: An example of $z(s), \bar{z}(s)$ along $\gamma_1$ in case 4.

So we get

$$n_t(\gamma_1) = -\bar{n}_t(\gamma_1), \tag{93}$$

which implies $N_t(\gamma_1) = 0$, hence $N_t(C_L) = N_t(C'_L)$.

The arguments for $\mathcal{D}_L^\alpha \cap E_{su}$, $\mathcal{D}_L^\alpha \cap E_{tu}$ and $\mathcal{D}_L^\alpha \cap E_{stu}$ are similar. $\qquad \square$

An immediate consequence of theorem 5.2 is that for a fixed causal ordering (say $\mathcal{D}_L^\alpha$), each blank space in table 1 satisfy the all-or-none law: either check mark for all configurations in $\mathcal{D}_L^\alpha$ or cross mark for all configurations in $\mathcal{D}_L^\alpha$. Therefore, if $\mathcal{D}_L^\alpha$ is in class S or T or U, then all its configurations have the same OPE convergence properties; if $\mathcal{D}_L^\alpha$ is in class E, then all its configurations in the same subclass have the same OPE convergence properties.[16]

### 5.3.2 Comments on the 2d case

In 2d unitary local CFTs, we have Al. Zamolodchikov's uniformizing variables $q, \bar{q}$ [11]. The function $g$ in (4) has a convergent expansion in terms of Virasoro blocks, and Virasoro blocks have convergent series expansions in $q, \bar{q}$ if $0 < |q|, |\bar{q}| < 1$, which includes the configurations with $0 \leq |\rho|, |\bar{\rho}| \leq 1$ except for $\rho$ or $\bar{\rho} = \pm 1$. However, $\rho$ or $\bar{\rho} = \pm 1$ only happens at light-cone singularities.[17] So we conclude that in the Lorentzian signature, the s-channel OPE is always convergent aside from light-cone singularities [12].

The above CFT argument is valid only for 2d unitary local CFTs, where by local we mean there exists a stress tensor $T_{\mu\nu}(x)$, which has the mode expansion in Virasoro generators [14]. There are also non-local CFTs, e.g. the generalized free field theories. These non-local CFTs have only global conformal symmetry, for which we can only use $\rho, \bar{\rho}$ instead of $q, \bar{q}$.

We claim that the conclusions in theorem 5.2 are still true for 2d unitary CFT (here we only assume global conformal symmetry). Unlike the case $d \geq 3$, the sets $\mathcal{D}_L^\alpha$ are usually disconnected in 2d. This is

---

[16]By configurations having the same OPE convergence properties, we mean that in each OPE channel, all or none of these configurations have the convergent expansion for the four-point function.

[17]If $\rho$ or $\bar{\rho} = 1$, then $v = 0$. If $\rho$ or $\bar{\rho} = -1$, then $u$ or $v = \infty$. Thus, for any configuration with $\rho$ or $\bar{\rho} = \pm 1$, there exists at least one $x_i, x_j$ pair such that $(x_i - x_j)^2 = 0$.

because in 2d, there are two disconnected space-like separations. So we cannot copy the proof of theorem 5.2. However, any 2d configuration can be embedded into $d \geq 3$. Since our criteria of OPE convergence properties are based on counting how the analytic continuation curves of $z, \bar{z}$ cross the intervals $(-\infty, 0)$, $(0, 1)$ and $(1, +\infty)$, which is dimension independent, the 2d path gives the same counting of $N_t, N_u$ as in $d \geq 3$. Therefore, theorem 5.2 also covers the 2d case.

The only little difference is that in the 2d case, the Lorentzian four-point configurations only have real $z, \bar{z}$. This follows from (41) and (42). So the subclass $E_{\text{stu}}$, where $z, \bar{z}$ are not real, does not exist in 2d.

## 5.4  Time reversals

In theorem 5.2, we have classified the Lorentzian configurations in $\mathcal{D}_L$ into a finite number of cases. For each case, we will have to choose a representative configuration and a path from $\mathcal{D}_E \backslash \Gamma$, then check if conditions of theorem 4.1, 4.4 and 4.6 hold. Actually, there are some further simplifications which will reduce the number of checks to perform. We are going to show that different $\mathcal{D}_L^\alpha$ which are related by time reversals have the same convergent OPE channels.

We define two time reversals:

$$
\begin{aligned}
\theta_E &: (\epsilon + it, \mathbf{x}) \mapsto (-\epsilon + it, \mathbf{x}) \\
\theta_L &: (\epsilon + it, \mathbf{x}) \mapsto (\epsilon - it, \mathbf{x})
\end{aligned}
\tag{94}
$$

They correspond to the time reversals in Euclidean and Minkowski space. Under time reversals, $x_{ij}^2$ takes its complex conjugate

$$
(\theta_E x_i - \theta_E x_j)^2 = (\theta_L x_i - \theta_L x_j)^2 = \left[(x_i - x_j)^2\right]^*
\tag{95}
$$

Given a configuration $C = (x_1, x_2, x_3, x_4)$, we define the time reversals of the configuration by (notice the change of order of points in $\theta_E C$)

$$
\begin{aligned}
\theta_E C &= (\theta_E x_4, \theta_E x_3, \theta_E x_2, \theta_E x_1), \\
\theta_L C &= (\theta_L x_1, \theta_L x_2, \theta_L x_3, \theta_L x_4).
\end{aligned}
\tag{96}
$$

Then the following properties are easily checked:

- The sets $\mathcal{D}$, $\mathcal{D}_E$ and $\mathcal{D}_L$ are preserved by $\theta_E$ and $\theta_L$.

- Under the transformation $C \mapsto \theta_E C$ or $C \mapsto \theta_L C$, the conformal invariants $u, v, z, \bar{z}, \rho, \bar{\rho}$ become their complex conjugates.

Suppose we have a path $\gamma$ from $\mathcal{D}_E \backslash \Gamma$ to $\mathcal{D}_L$. Then $\theta_E \gamma$ and $\theta_L \gamma$ are still paths from $\mathcal{D}_E \backslash \Gamma$ to $\mathcal{D}_L$. The curves of $z, \bar{z}$ are reflected with respect to the real axis, which implies

$$
\begin{aligned}
N_t(\theta_E \gamma), N_t(\theta_L \gamma) &= -N_t(\gamma), \\
N_u(\theta_E \gamma), N_u(\theta_L \gamma) &= -N_u(\gamma).
\end{aligned}
\tag{97}
$$

By theorem 4.1, 4.4 and 4.6, we conclude that

- Different Lorentzian configurations which are related by $\theta_E, \theta_L$ have the same convergent OPE channels.

By lemma 5.1 and theorem 5.2, we translate the above results to the level of causal orderings:

- If two different sets $\mathcal{D}_L^\alpha, \mathcal{D}_L^\beta$ are related by $\theta_E, \theta_L$, then they belong to the same class (S, T, U, E).

- If two different sets $\mathcal{D}_L^\alpha, \mathcal{D}_L^\beta$ are in class S or T or U and are related by $\theta_E, \theta_L$, then they have the same convergent OPE channels.

- If two different sets $\mathcal{D}_L^\alpha, \mathcal{D}_L^\beta$ are in class E and are related by $\theta_E, \theta_L$, then their intersections with each subclass have the same convergent OPE channels.

Given a Lorentzian configuration $C = (x_1, x_2, x_3, x_4)$, $\theta_E$ interchanges $x_1 \leftrightarrow x_4$ and $x_2 \leftrightarrow x_3$. At the level of causal orderings, $\theta_E$ is the permutation of indices

$$1 \leftrightarrow 4, \quad 2 \leftrightarrow 3, \tag{98}$$

with all the arrows kept fixed. For example, under $\theta_E$ we have

$$1 \longrightarrow 2 \begin{array}{c} \nearrow 3 \\ \searrow 4 \end{array} \quad \Rightarrow \quad 4 \longrightarrow 3 \begin{array}{c} \nearrow 2 \\ \searrow 1 \end{array}. \tag{99}$$

Under $\theta_L$, the Lorentzian configuration $x_k = (it_k, \mathbf{x}_k)$ is mapped to $C' = (x_1', x_2', x_3', x_4')$ with

$$x_k' = \theta_L x_k = (-it_k, \mathbf{x}_k), \quad k = 1, 2, 3, 4. \tag{100}$$

So the operator ordering does not change but the causal ordering is reversed. For example, under $\theta_L$ we have

$$1 \longrightarrow 2 \begin{array}{c} \nearrow 3 \\ \searrow 4 \end{array} \quad \Rightarrow \quad 1 \longleftarrow 2 \begin{array}{c} \nwarrow 3 \\ \nwarrow 4 \end{array}. \tag{101}$$

By definitions (94) and (96), we have the following properties for $\theta_E, \theta_L$:

$$\theta_E^2 = id, \quad \theta_L^2 = id, \quad \theta_E \theta_L = \theta_L \theta_E. \tag{102}$$

So the group generated by $\theta_E, \theta_L$ is $\mathbb{Z}_2 \times \mathbb{Z}_2$. Under the $\mathbb{Z}_2 \times \mathbb{Z}_2$-actions, the orbit of a given causal ordering contains 1 or 2 or 4 causal orderings. In each orbit, it suffices to check the OPE convergence properties of only one causal ordering and make the same conclusions for other causal orderings. This simplifies our work.

## 5.5 The table of four-point causal orderings

Given two Lorentzian configurations $(x_1, \ldots, x_n)$ and $(y_1, \ldots, y_n)$, we say that they are in the same causal type if there is a permutation $\sigma \in S_n$ such that $(x_{\sigma(1)}, \ldots, x_{\sigma(n)})$ has the same causal ordering as $(y_1, \ldots, y_n)$ or $(\theta_L y_1, \ldots, \theta_L y_n)$.

In table 2, we give a classification of four-point causal orderings according to the causal types. The vertices labelled by $a, b, c, d$ can be any permutation of $1, 2, 3, 4$. In the end we will give a table about OPE convergence properties for each causal type in table 2.

Table 2: Classification of four-point causal orderings

| Type No. | causal ordering | $\theta_L$ time reversal |
|---|---|---|
| 1 | $a \longrightarrow b \longrightarrow c \longrightarrow d$ | same |
| 2 | $a \longrightarrow b \nearrow^{c}_{\searrow d}$ | $^{c}_{d}\searrow^{\nearrow} b \longrightarrow a$ |
| 3 | $a \rightrightarrows b \longrightarrow c$ (with $d$ below) | $c \longrightarrow b \rightrightarrows a$ (with $d$ below) |
| 4 | $a \nearrow^{b}\searrow_{\nearrow}^{\searrow} d$ ($c$ below) | same |
| 5 | $a \rightrightarrows^{b}_{c, d}$ | $^{b}_{c,d} \rightrightarrows a$ |
| 6 | $a \longrightarrow b \longrightarrow c$ (with $d$ below) | same |
| 7 | $a \nearrow^{b}\searrow_{c}$ ($d$ below) | $^{b}\searrow_{\nearrow c}^{a}$ ($d$ below) |
| 8 | $a \nearrow^{b}\searrow_{d}$, $c \nearrow d$ | same |
| 9 | $a \longrightarrow b$ ($c$, $d$ below) | same |
| 10 | $^{a}\searrow_{\nearrow}^{\searrow} c \longrightarrow d$ ($b$ below) | same |
| 11 | $a \longrightarrow b$, $c \longrightarrow d$ | same |
| 12 | $a \quad b \quad c \quad d$ | same |

Each causal type thus represents at most $4! \times 2$ causal orderings ($4!$ for possible assignments of $1, 2, 3, 4 \rightarrow a, b, c, d$ and $\times 2$ for two columns). This maximal number is realized for type 3, while for other types it is smaller because often second column is equivalent to the first and because of little group (see appendix C.0.3).

It makes sense to do this grouping of causal orderings into causal types for two reasons:

- causal orderings related by $\theta_E$ and $\theta_L$ action (and which thus have same OPE convergence properties) belong to the same causal type.

- if we know class/subclass of $\mathcal{D}_L^\alpha$ for one $\alpha$ in a given causal type, it is easy to determine the class/subclass of any other $\mathcal{D}_L^\alpha$ in the same causal type (see appendix C.0.1).

## 5.6 Examples

The tables which classify the OPE convergence properties will be particularly large, we leave them in appendix C. Readers can pick the cases they are interested in. To make it easy for readers to check, we also share the Mathematica code which contain the OPE convergence results of all causal orderings, see the file "/anc/OPE_check.nb". In this section we only give some examples.

The Lorentzian four-point correlation functions defined in (2) are either time-ordered ($t_1 > t_2 > t_4 > t_4$) or out-of-time-order (not $t_1 > t_2 > t_4 > t_4$). The time-ordered correlators have applications in scattering theories [45, 46], and the out-of-time-order correlation functions have applications in the study of many-body systems [47, 48, 49, 50, 51, 52, 53, 54]. An example in [36] shows the existence of out-of-time-order correlators which do not have a convergent OPE channel (see appendix A in [36]).[18] Our first example is to show that not all time-ordered correlators have a convergent OPE channel.

Then we will discuss two other examples from AdS/CFT. One is the Regge kinematics [55, 56], the other is related to the bulk-point singularities [12].

### 5.6.1 A time-orderd correlation function

Let us consider the following two-dimensional configuration:

$$x_1 = (0,0), \quad x_2 = (-0.1i, 1), \quad x_3 = (-2i, -1.5), \quad x_4 = (-2.1i, 1.5). \tag{103}$$

The four-point function $G_4^L(x_1, x_2, x_3, x_4)$ at the configuration (103) is time-ordered. The causal ordering of (103) is given by

$$
\begin{array}{c}
4 \diagup^{\displaystyle 2} \\
\diagdown_{\displaystyle 1} \\
3 \diagup
\end{array}
, \tag{104}
$$

which is of causal type 8 in table 2. A quick way to know the OPE convergence property is to look up the table of OPE convergence in appendix C.8. The causal ordering (104) corresponds to the label "(4231)" in table 12. We see from table 12 that there is no convergent OPE channel for this causal ordering.

Let us also choose a start point in $\mathcal{D}_E \backslash \Gamma$ and a path to compute the $z, \bar{z}$-curves, and directly check the OPE convergence properties. Figure 5.4 shows the $z, \bar{z}$-curves along the path.[19]

---

[18]By "a configuration do not have a convergent OPE channel" we mean the configuration do not satisfy the conditions of theorem 4.1, 4.4 and 4.6.

[19]We choose the start point $x_1^E = (0, -0.8)$, $x_2^E = (-1, -0.2)$, $x_3^E = (-2, -0.6)$ and $x_4^E = (-3, -0.3)$. The path is given by the straight line.

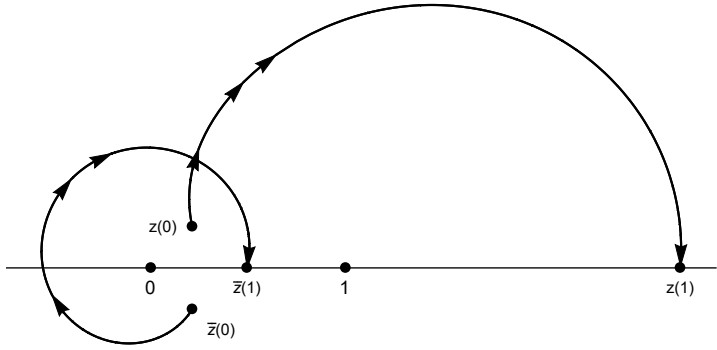

Figure 5.4: $z, \bar{z}$-curves of the configuration (103).

We see from figure 5.4 that $z > 1$, $0 < \bar{z} < 1$ at the final point, which implies that the configuration (103) is in class T. The curve of $z$ variable crosses $(-\infty, 0)$ from below, which gives $N_t = -1$. So the t-channel OPE (the only undetermined case by table 1) is not convergent. Thus, as already mentioned, there is no convergent OPE channel for the four-point function at the configuration (103).

This example shows that not all time-ordered correlation functions have a convergent OPE channel.

### 5.6.2   Regge kinematics

The second example is the Lorentzian four-point function in the Regge regime [55, 56, 57]. Let $x_1, x_4$ and $x_2, x_3$ pairs be time-like separated, while other pairs be space-like separated (see figure 5.5).

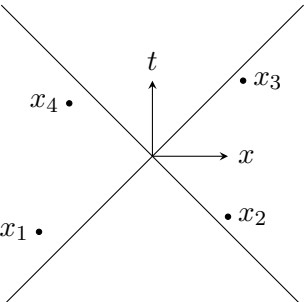

Figure 5.5: Regge kinematics.

It is well known that the four-point function at Regge kinematics only has convergent t-channel expansion [55]. Here we just review this result. The causal ordering of the Regge kinematics is given by

$$\begin{aligned} 1 &\longrightarrow 4 \\ 2 &\longrightarrow 3 \end{aligned} \tag{105}$$

The Regge kinematics belongs to causal type 11 in table 2. Let us look up this causal ordering in appendix C.11. The causal ordering (105) corresponds to the label "(1423)" in table 15. We see that only t-channel OPE is convergent.

We would like to also choose a representative configuration and a path to compute the curves of $z, \bar{z}$. The

plot is given by figure 5.6. [20]

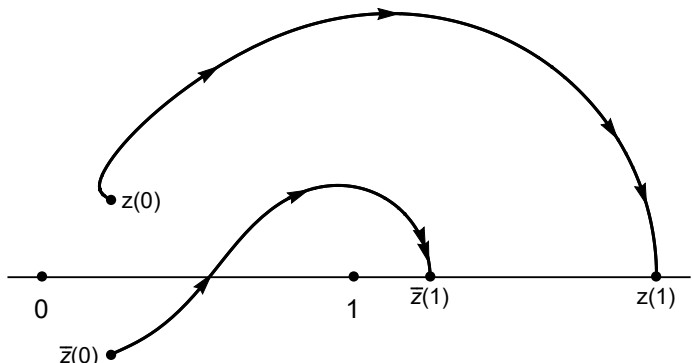

Figure 5.6: The plot of $z, \bar{z}$-curves of the Regge kinematics.

We see from figure 5.6 that $z, \bar{z} > 1$ at the final point,[21] which implies that the Regge kinematics is in class E. In fact the Regge kinematics can only be in the subclass $E_{tu}$, where $z, \bar{z} > 1$ [55], so only t- and u-channel expansions have a chance to converge. We see from figure 5.6 that the $\bar{z}$-curve crosses $(0, 1)$ from below, and $z, \bar{z}$-curves do not cross $(-\infty, 0)$. So we get

$$N_t = 0, \quad N_u = -1, \tag{106}$$

which implies that only the t-channel expansion is convergent.

### 5.6.3 Causal ordering of bulk-point singularities

The third example is as follows. Let $x_1, x_2$ and $x_3, x_4$ pairs be space-like separated. We put the $x_1, x_2$ pair in the open backward light-cone of some base point and $x_3, x_4$ pair in the open forward light-cone of the base point (see figure 5.7).

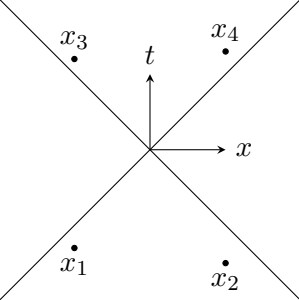

Figure 5.7: Configuration of example 3.

---

[20]We choose the Euclidean configuration $x_1 = 0$, $x_2 = (-1, 0, 0, 0)$, $x_3 = (-2, 0.9, 0, 0)$, $x_4 = (-4, 0, 0, 0)$ and the representative Lorentzian configuration $y_1 = 0$, $y_2 = (0, 0, 0.6, 0)$, $y_3 = (2i, 0, 0, 0.7)$, $y_4 = (2i, -0.05, 0, -3)$. We choose the path to be the straight line between them.

[21]The definition of $z, \bar{z}$ in [55] is different from this paper. In their work, $0 < z, \bar{z} < 1$ at Regge kinematics, while in this paper, $z, \bar{z} > 1$. One can compare the definitions and get the relation of $z, \bar{z}$ between [55] and our work: $z \to 1/z$, $\bar{z} \to 1/\bar{z}$.

Such configurations have the causal ordering

$$
\begin{array}{c}
1 \searrow \\
\phantom{1} 3 \rightarrow 4 \ . \\
2 \nearrow
\end{array}
\tag{107}
$$

The causal ordering (107) is of causal type 10 in table 2. We look up the OPE convergence properties in appendix C.10. The causal ordering (107) corresponds to the label "(1234)" in table 14. We see that this causal ordering is in class E, which has four subclasses. From table 14 we also see that the configurations with the causal ordering (107) exist in each subclass. We wish to consider the subclass $E_{ut}$, where $z, \bar{z} > 1$. In table 14, we see that the configurations with causal ordering (107) and in subclass $E_{ut}$ have no convergent OPE channels.

Let us also choose a representative configuration to check this result. We want to remark that such case does not exist in 2d (see appendix C.10 for the proof). We choose the following three-dimensional configuration

$$
x_1 = (0, 0, 0), \quad x_2 = (0, 1, 0), \quad x_3 = (i, 0.2, 0.5), \quad x_4 = (i, 0.5, 0.8).
\tag{108}
$$

Figure 5.8 shows the plot of $z, \bar{z}$-curves.

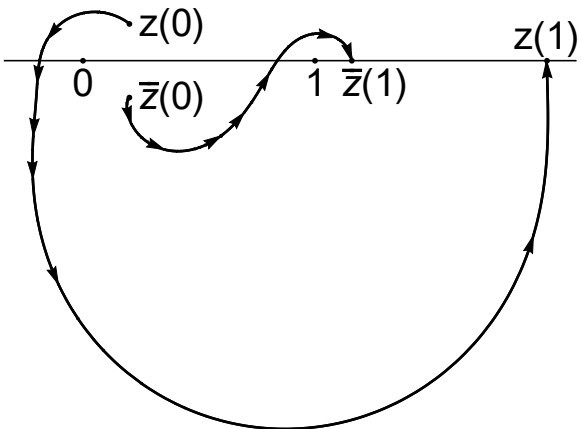

Figure 5.8: The plot of $z, \bar{z}$-curves of the configuration (108).

We see that along the path, $z$ crosses the interval $(-\infty, 0)$ and $\bar{z}$ crosses the interval $(0,1)$. We get

$$
N_t = 1, \quad N_u = -1.
\tag{109}
$$

which implies that the t- and u-channel expansions do not converge.

We conclude that there is no convergent OPE channel for the causal ordering (107) with $z, \bar{z} > 1$.

Here we give a hint why this example is related to the bulk-point singularities in AdS/CFT [12]. The bulk-point singularities are not exactly the configurations in Minkowski space $\mathbb{R}^{d-1,1}$, instead they are configurations on the Minkowski cylinder $\mathbb{R} \times S^{d-1}$ [58]. The Minkowski space can be embedded into a patch of the Minkowski cylinder in a Weyl equivalent way, this patch is called the Poincaré patch[59]. The Minkowski cylinder also admits a causal ordering which is equivalent to the causal ordering of the Minkowski space in the Poincaré patch [60, 61]. One can show the following facts:

- The bulk-point singularities have the causal ordering (107) and $z, \bar{z} > 1$.

- One can find a path from an arbitrary bulk-point singularity to a configuration in the Poincaré patch, such that the causal ordering (107) is preserved along the path.

- The CFT four-point function in the Poincaré patch is the same as the CFT four-point function in the Minkowski space up to a scaling factor.[22]

Based on the above facts, the OPE convergence properties of the bulk-point singularities are exactly the same as this example: there is no convergent OPE channel. More details will be given in [38]. Our result does not contradict the two-dimensional result in [12] (see the beginning of section 5.3.2) because here we only use the global conformal symmetry instead of the Virasoro symmetry.

### 5.6.4 Digression

We can see from figure 5.4, 5.6 and 5.8 that the $z, \bar{z}$-curves do not touch the interval $(1, +\infty)$ until the end. One can also see this phenomenon by picking representative configurations of other causal orderings and compute the $z, \bar{z}$-curves. This numerical observation is consistent with theorem 3.1.

## 6  Wightman functions: a brief review

In this section we will review some classical results about regular points (points where Wightman distributions are genuine functions) in a general QFT [6, 5, 62]. For simplicity let us still consider a scalar theory in the Minkowski space, which is characterized by a collection of Lorentzian correlators:

$$\mathcal{W}_n(x_1, x_2, \ldots, x_n) \coloneqq \langle 0| \phi(x_1)\phi(x_2)...\phi(x_n) |0\rangle \tag{110}$$

where $x_i$ are Lorentzian coordinates.[23]  We will introduce the Wightman axioms for QFTs, and then review the domain of correlation functions which can be derived from Wightman axioms. In the end, we will compare these classical results with our results for CFT four-point functions.

This section is logically independent from the rest of the paper. Here we assume Wightman axioms while in the rest we did not. The only connection is to justify the definition of Wick rotation (steps 1 and 2 in section 2.1).

### 6.1  Wightman axioms for Lorentzian correlators

We assume the Wightman axioms for correlators $\{\mathcal{W}_n\}$:

$(W1)$ Temperedness.

$\mathcal{W}_n$ is a tempered distribution (called Wightman distribution). It becomes a complex number after being smeared with rapidly decreasing test functions $f_n$:

$$\mathcal{W}_n(f_n) = \int f(x_1, \ldots, x_n)\mathcal{W}_n(x_1, \ldots, x_n)dx_1 \ldots dx_n \tag{111}$$

---

[22]The definition of the CFT four-point function on the Minkowski cylinder is similar to Minkowski space. We replace the planar time variables $\tau_k$ by the cylindrical time variables. Then do Wick rotations.

[23]In the rest of the paper we the Lorentzian points were denoted by $(it_k, \mathbf{x}_k)$. Only in this section we use the notation $x_k = (t_k, \mathbf{x}_k)$.

The Fourier transform $\hat{\mathcal{W}}_n$ of $\mathcal{W}_n$ is well defined since the space of rapidly decreasing test functions (Schwartz space) is closed under Fourier transform [63]. One has the definition $\hat{\mathcal{W}}_n(f_n) := \mathcal{W}_n\left(\hat{f}_n\right)$.

($W2$) Poincaré invariance.

The correlators transform invariantly under action of the Poincaré group:

$$\mathcal{W}_n(g \cdot x_1, \ldots, g \cdot x_n) = \mathcal{W}_n(x_1, \ldots, x_n) \tag{112}$$

for all $n \geq 0$ and $g$ in the Poincaré group.

($W3$) Unitarity.

The vector space generated by the states of the form

$$\Psi\left(\underline{f}\right) = \sum_{n \geq 0} \int f_n(x_1, \ldots, x_n) \phi(x_1) \ldots \phi(x_n) |0\rangle \, dx_1 \ldots dx_n \tag{113}$$

has a non-negative inner product. Here $\underline{f}$ is an arbitrary finite sequence of complex valued Schwartz functions: $\underline{f} = (f_0, f_1, f_2, \ldots)$ and $f_n$ denotes the Schwartz function with $n$ Lorentzian points as variables. If we assume that $\phi(x)$ are Hermitian operators, i.e. $[\phi(x)]^\dagger = \phi(x)$, then the unitarity condition is written as

$$\sum_{n,m} \int \overline{f_n(x_1 \ldots, x_n)} f_m(y_1, \ldots, y_m) \mathcal{W}_{n+m}(x_n, \ldots, x_1, y_1, \ldots, y_m) dx dy \geq 0 \tag{114}$$

($W4$) Spectral condition.

The open forward light-cone $V_+$ is defined by the collection of vectors $x \in \mathbb{R}^d$ such that

$$x^0 > \sqrt{\sum_{\mu \geq 1} (x^\mu)^2}. \tag{115}$$

In a general QFT we have self-adjoint momentum operators $P^\mu$. The spectral condition says that the spectrum of $P = (P^0, P^1, \ldots, P^{d-1})$ is inside the closed forward light-cone $\overline{V_+}$, and the normalized eigenvector of $P = 0$ is unique (up to a phase factor), denoted by $|0\rangle$.

We define the reduced correlators $W_{n-1}$ by

$$W_n(x_2 - x_1, \ldots, x_{n+1} - x_n) = \mathcal{W}_{n+1}(x_1, \ldots, x_{n+1}) \tag{116}$$

Since $\mathcal{W}_{n+1}$ is a translational invariant tempered distribution, $W_n$ is well defined and is also a tempered distribution. The spectral condition implies that the Fourier transforms $\hat{W}_n$ of $W_n$ is supported in the forward light-cone. That is to say, $\hat{W}_n(p_1, \ldots, p_n) \neq 0$ only if all the momentum variables $p_k$ are inside $\overline{V}_+$.

($W5$) Microscopic causality.

$\mathcal{W}_n(x_1, \ldots, x_k, x_{k+1}, \ldots, x_n) = \mathcal{W}_n(x_1, \ldots, x_{k+1}, x_k, \ldots, x_n)$ if $x_k$ and $x_{k+1}$ are space-like separated.

## 6.2 Wightman functions and their domains

### 6.2.1 Forward tube

Let us consider the "reduced correlator" $W_n$ defined in eq. (116). $W_n$ has Fourier transform

$$W_n(x_1, \ldots, x_n) = \int \frac{dp_1}{(2\pi)^d} \cdots \frac{dp_n}{(2\pi)^d} \hat{W}_n(p_1, \ldots, p_n) e^{-i(p_1 \cdot x_1 + \ldots + p_n \cdot x_n)}, \tag{117}$$

where $\hat{W}_n$ is also a tempered distribution, and the Lorentzian inner product is defined by $p \cdot x = -p^0 x^0 + p^1 x^1 + \ldots + p^{d-1} x^{d-1}$. In general, $W_n$ is not a function if $x_k$ are real. However, if we replace $x_k$ with complex coordinates $x_k \to z_k = x_k + iy_k$, because of the spectral condition $(W4)$, $W_n(z_1, \ldots, z_n)$ is indeed a function if the imaginary parts of $z_k$ belong to $V_+$. The argument is as follows. Suppose $y_k \in V_+$ for all $k = 1, 2, \ldots, n$, then there exists a Schwartz function $f(p_1, \ldots, p_n)$ in the momentum space such that $f(p_1, \ldots, p_n) = e^{-i(p_1 \cdot z_1 + \ldots + p_n \cdot z_n)}$ when all the momentum variables $p_k$ are inside the closure of forward light-cone.[24] Since $\hat{W}_n$ is supported in $\overline{V}_+^n$, $\hat{W}_n(f)$ is exactly in the form of (117) with $x_k$ replaced by $z_k$. So $W_n(z_1, \ldots, z_n)$ is a well-defined complex number when $Im(z_k) \in V_+$ for all $k$.

Furthermore, since $-i (p_k)_\mu f(p_1, \ldots, p_n)$ is also a Schwartz test function, we have

$$\frac{\partial}{\partial z_k^\mu} W_n(z_1, \ldots, z_n) = \hat{W}_n[-i (p_k)_\mu f],$$
$$k = 1, \ldots, n \text{ and } \mu = 0, \ldots, d-1. \tag{118}$$

As a result $W_n(z_1, \ldots, z_n)$ is an analytic function inside the "forward tube", denoted as $I_n$

$$I_n = \left\{ (z_1, \ldots, z_n) \in \mathbb{C}^{nd} \middle| Im(z_k) \in V_+, \ k = 1, 2, \ldots, n \right\}, \tag{119}$$

The distribution $W_n(x_1, \ldots, x_n)$ is the boundary value of the analytic function $W_n(z_1, \ldots, z_n)$ on the forward tube $I_n$:

$$W_n(x_1, \ldots, x_n) = \lim_{\substack{y \to 0 \\ y \in V_+^n}} W_n(x_1 + iy_1, \ldots, x_n + iy_n). \tag{120}$$

### 6.2.2 Bargmann-Hall-Wightman theorem, extended tube

Now let us use the Lorentz invariance $(W2)$ to analytically continue $W_n$ to a larger domain. By $(W2)$, $W_n$ is invariant under the action of real Lorentz group $SO^+(1, d-1)$:[25]

$$W_n(g \cdot x_1, \ldots, g \cdot x_n) = W_n(x_1, \ldots, x_n), \quad \forall g \in SO^+(1, d-1). \tag{121}$$

The Lorentz transformations preserve the inner product $p_k \cdot x_k$ and the measure $dx_k$, so the Fourier transform $\hat{W}_n$ is also Lorentz invariant

$$W_n(g \cdot p_1, \ldots, g \cdot p_n) = W_n(p_1, \ldots, p_n), \quad \forall g \in SO^+(1, d-1). \tag{122}$$

Since $W_n(z_1, \ldots, z_n)$ is defined by the Fourier transform (117) (replace $x_k$ with $z_k$), we have

$$W_n(g \cdot z_1, \ldots, g \cdot z_n) = W_n(z_1, \ldots, z_n), \quad \forall g \in SO^+(1, d-1). \tag{123}$$

Here we remark that the real Lorentz group actions preserve the forward tube $I_n$.

An important observation is that (123) remains true if we replace the real Lorentz group $SO^+(1, d-1)$ by the proper complex Lorentz group $L_+(\mathbb{C})$.[26] Given an arbitrary $g \in L_+(\mathbb{C})$, we define

$$W_n^g(z_1, \ldots, z_n) := W_n\left(g^{-1} \cdot z_n, \ldots, g^{-1} \cdot z_n\right) \tag{124}$$

---

[24]The crucial point is that if all $y_k$ are inside the forward light-cone, then $f(p_1, \ldots, p_n)$ decays exponentially fast when some $p_k$ goes to infinity inside the forward light-cone.

[25]By $SO^+(1, d-1)$ we mean the connected component of the identity element in $O(1, d-1)$.

[26]Let $d$ be the spacetime dimension. The complex Lorentz group $L(\mathbb{C})$ is defined by the set of all $d \times d$ complex matrices $M$ such that $M^t \eta M = \eta$. Here $\eta = \text{diag}(-1, 1, \ldots, 1)$ is the matrix of Lorentzian inner product, and $M^t$ is the transpose of $M$. $L_+(\mathbb{C})$ is the subgroup of $L(\mathbb{C})$ with constraint $\det M = 1$. $L_+(\mathbb{C})$ is connected, unlike the real case where we need to introduce the constraints "proper", "orthochronous" for connectedness.

for $(z_1, \ldots, z_n) \in gI_n$. The Bargmann-Hall-Wightman theorem [64] tells us that if we choose different complex Lorentz group elements $g_1, g_2$, the functions $W_n^{g_1}$ and $W_n^{g_2}$ coincide in the domain $g_1 I_n \cap g_2 I_n$. So $W_n(z_1, \ldots, z_n)$ has analytic continuation to the "extended forward tube", denoted by $\tilde{I}_n$:

$$\tilde{I}_n := \left\{ (z_1, \ldots, z_n) \in \mathbb{C}^{nd} \Big| (g \cdot z_1, \ldots, g \cdot z_n) \in I_n \text{ for some } g \in L_+(\mathbb{C}) \right\}. \tag{125}$$

Here we only give the idea of the proof. It suffices to show that for any $g \in L_+(\mathbb{C})$, the function $W_n^g$ coincides with $W_n$ in the domain $gI_n \cap I_n$. Since $I_n$ and $gI_n$ are two convex sets, their intersection $gI_n \cap I_n$ is also convex, thus connected. So it suffices to show that $W_n^g$ coincides with $W_n$ in the neighbourhood of one point. This is obvious for $g$ near the identity element, but the proof for an arbitrary $g$ is based on the fact that the set $\{g \in L_+(\mathbb{C}) \mid gI_n \cap I_n \neq \emptyset\}$ is connected, which follows from the group structure of the complex Lorentz group $L_+(\mathbb{C})$ (for more details, see [1]).

### 6.2.3 Jost points

While $I_n$ does not contain Lorentzian points (i.e. points with $\text{Im}(z_k) = 0$ for all $k$), $\tilde{I}_n$ contains a region of Lorentzian points. These points are called Jost points [6], and they are defined by the configurations $(x_1, \ldots, x_n)$ such that the following cone

$$\left\{ \lambda_1 x_1 + \ldots + \lambda_n x_n \Big| \lambda_k \geq 0 \text{ for all } k, \quad \sum_{k=1}^{n} \lambda_k > 0 \right\} \tag{126}$$

contains only space-like points (see [1], the theorem on page 81 and the corollary on page 82).

### 6.2.4 Microscopic causality, envelope of holomorphy

Now let us go back to $\mathcal{W}_n(x_1, \ldots, x_n)$ via (116). We define $\mathcal{J}_n$ as the set of $(x_1, \ldots, x_n)$ such that $(x_2 - x_1, \ldots, x_n - x_{n-1})$ are Jost points. The configurations in $\mathcal{J}_n$ have totally space-like separations. To see this we rewrite $x_i - x_j$ $(i > j)$ as

$$x_i - x_j = (x_i - x_{i-1}) + (x_{i-1} - x_{i-2}) + \ldots + (x_{j+1} - x_j), \tag{127}$$

which is in the form of (126). By the definition of Jost points, we have $(x_i - x_j)^2 > 0$ for $i \neq j$. It is obvious that $\mathcal{J}_2$ contains all totally space-like configurations. For $n \geq 3$, $\mathcal{J}_n$ does not contain all totally space-like configurations.

Since Jost points are the configurations with totally space-like separations, by the microscopic causality condition (W5), $\mathcal{W}_n(x_1, \ldots, x_n)$ is also regular at $(x_1, \ldots, x_n)$ if there exists a permutation $\sigma \in S_n$ such that

$$\left( x_{\sigma(1)}, x_{\sigma(2)}, \ldots, x_{\sigma(n)} \right) \in \mathcal{J}_n. \tag{128}$$

Then the equation $\mathcal{W}_n(x_1, \ldots, x_n) = \mathcal{W}_n(x_{\sigma(1)}, \ldots, x_{\sigma(n)})$ in $\mathcal{J}_n$ can be analytically continued to

$$\mathcal{W}_n(z_1, \ldots, z_n) = \mathcal{W}_n(z_{\sigma(1)}, \ldots, z_{\sigma(n)}), \quad (z_2 - z_1, \ldots, z_n - z_{n-1}) \in \tilde{I}_{n-1}. \tag{129}$$

Therefore, $\mathcal{W}_n$ has analytic continuation to the following domain of complex coordinates $(z_1, \ldots, z_n)$:

$$\mathcal{U}_n = \left\{ (z_1, \ldots, z_n) \in \mathbb{C}^{nd} \Big| \exists \sigma \in S_n \text{ s.t. } (z_{\sigma(2)} - z_{\sigma(1)}, \ldots, z_{\sigma(n)} - z_{\sigma(n-1)}) \in \tilde{I}_{n-1} \right\}. \tag{130}$$

$\mathcal{U}_n$ have the following properties:[27]

---

[27] We were unable to track properties 1,2 in prior literatures. Readers are welcome to provide us with references.

1. In 2d, $\mathcal{U}_n$ contains all totally space-like configurations.

2. $\mathcal{U}_3$ contains all totally space-like configurations.

3. In $d \geq 3$ and $n \geq 4$, $\mathcal{U}_n$ does not contain all totally space-like configurations [1].

To show the first property, we use an analogous version of the complex coordinates (44):

$$w_k = \mathbf{x}_k + t_k, \quad \bar{w}_k = \mathbf{x}_k - t_k. \tag{131}$$

Given an arbitrary totally space-like configuration, we have

$$(w_j - w_k)(\bar{w}_j - \bar{w}_k) > 0, \quad (j \neq k) \tag{132}$$

which implies $w_j > w_k, \bar{w}_j > \bar{w}_k$ or $w_j < w_k, \bar{w}_j < \bar{w}_k$. So we can find a permutation $\sigma$ such that

$$w_{\sigma(k)} < w_{\sigma(k+1)}, \quad \bar{w}_{\sigma(k)} < \bar{w}_{\sigma(k+1)}, \quad k = 1, 2, \ldots, n-1. \tag{133}$$

We see from (133) that $w_{\sigma(k+1)} - w_{\sigma(k)}, \bar{w}_{\sigma(k+1)} - \bar{w}_{\sigma(k)}$ are positive, so the cone (126) generated from these vectors only contain points with positive components. Thus the configuration $(x_{\sigma(1)}, \ldots, x_{\sigma(n)})$ is in $\mathcal{J}_n$, or equivalently, $(x_1, \ldots, x_n)$ is in $\mathcal{U}_n$.

To show the second property, we consider the totally space-like three-point configurations in the following form:

$$x_1 = 0, \quad x_2 = (0, 1, 0), \quad x_3 = (a, b, c). \tag{134}$$

To check that all totally space-like configurations are in $\mathcal{U}_3$, it suffices to check the totally space-like configurations in the form of (134) because $\mathcal{U}_n$ is Poincaré invariant and scale invariant, and any totally space-like configuration can be mapped to a configuration in the form of (134) by Poincaré transformations and dilatation. If $(x_1, x_2, x_3) \in \mathcal{J}_3$ then we are done. Suppose $(x_1, x_2, x_3) \notin \mathcal{J}_3$, which means that there exists a positive $\lambda$ such that $\lambda(x_2 - x_1) + (x_3 - x_2)$ is a null vector:

$$a^2 = (b - 1 + \lambda)^2 + c^2. \tag{135}$$

The above equation implies $a^2 \geq c^2$, so there exists a Lorentz transformation which maps the configuration (134) to

$$x_1' = 0, \quad x_2' = (0, 1, 0), \quad x_3' = (a', b, 0). \tag{136}$$

We see that the problem is reduced to the 2d case. According second property, the configuration (136) is in $\mathcal{U}_3$. So we conclude that all totally space-like configurations are in $\mathcal{U}_3$.

To show the third property, it suffices to give a counterexample for $d = 3$ and $n = 4$:

$$\begin{aligned} x_1 &= (1 - \epsilon, 1, 1), \quad x_2 = (1 - \epsilon, -1, -1), \\ x_3 &= (\epsilon - 1, 1, -1), \quad x_4 = (\epsilon - 1, -1, 1). \end{aligned} \tag{137}$$

where $\epsilon > 0$ is small. (137) is a totally space-like configuration but it does not belong to $\mathcal{U}_4$ (see [1], p. 89).

$\mathcal{W}_n$ has analytic continuation from $\mathcal{U}_n$ to its envelope of holomorphy $H(\mathcal{U}_n)$ [65], which is defined by the following property:

- Any holomorphic function on $\mathcal{U}_n$ has analytic continuation to $H(\mathcal{U}_n)$.

A theorem proved by Ruelle [7] shows that $H(\mathcal{U}_n)$ contains all totally space-like configurations.

We conclude that Wightman distributions $\mathcal{W}_n(x_1, \ldots, x_n)$ are analytic functions at all totally space-like configurations.

## 6.3 Comparison with CFT

### 6.3.1 Justifying the definition of Wick rotation

In section 3 of this paper, we used CFT arguments (not using Wightman axioms) to show that the CFT four-point function is analytic in the domain $\mathcal{D}$, where only the temporal variables are complex numbers (see section 2.1). Consider the points $(x_2 - x_1, x_3 - x_2, x_4 - x_3)$ where $(x_1, \ldots, x_4) \in \mathcal{D}$. The notation we use in the rest of the paper is $x_k = (\epsilon_k + it_k, \mathbf{x}_k)$, so we have

$$x_{k+1} - x_k = (\epsilon_{k+1} - \epsilon_k + it_{k+1} - it_k, \mathbf{x}_{k+1} - \mathbf{x}_k). \tag{138}$$

By translating (138) to the notation in this section (see footnote 23), we have

$$x_{k+1} - x_k = (t_{k+1} - t_k, \mathbf{x}_{k+1} - \mathbf{x}_k) + i(\epsilon_k - \epsilon_{k+1}, 0). \tag{139}$$

We see from (139) that the points $(x_2 - x_1, x_3 - x_2, x_4 - x_3)$ are in the forward tube $I_3$ if $(x_1, x_2, x_3, x_4) \in \mathcal{D}$ (because $\epsilon_1 > \epsilon_2 > \epsilon_3 > \epsilon_4$). Recalling that the Wightman distritbutions $\mathcal{W}_n(x_1, \ldots, x_n)$ can be obtained by taking the limit of the analytic functions $\mathcal{W}_n(z_1, \ldots, z_n)$ from the domain $\{(z_2 - z_1, \ldots, z_n - z_{n-1}) \in I_{n-1}\}$, we see that our definition of Wick rotation (2) is consistent with Wightman QFT.

### 6.3.2 Osterwalder-Schrader theorem

In fact we use the same analytic continuation path as in the Osterwalder-Schrader (OS) theorem [2]. The OS theorem shows that under certain conditions, a Euclidean QFT can be Wick rotated to a Wightman QFT.

In this paper we focus on the domain of analyticity of the four-point function, and we do not explore the distributional properties of it. To show that the limit (2) of the CFT four-point functions defines a Wightman four-point distribution, one needs to deal with the four-point function not only at regular points (where $|\rho|, |\bar{\rho}| < 1$ in s- or t- or u-channel), but also at all the other Lorentzian configurations where $\rho$ and/or $\bar{\rho}$ is 1 in absolute value (this needs a lot of extra work). Readers can go to [37], where we show that Wick rotating a Euclidean CFT four-point function indeed results in a Wightman four-point distribution.

Let us contrast our construction and the OS construction. Our construction extends $G_4$ in a CFT to domain $\mathcal{D}$ using only information about the four-point function itself. The OS construction can extend $G_4$ (in fact any $G_n$) to domain $\mathcal{D}$ in a general QFT. But the price to pay is that analytic continuation involves infinitely many steps and needs information about higher-point functions [3, 4].

### 6.3.3 Domain of analyticity: Wightman axioms + conformal invariance

Let us summarize how the domains of Wightman functions are derived in Wightman QFT. We first use the temperedness property (W1), translational invariance (W2) and the spectral condition (W4) to show that the reduced Wightman distribution $W_n$ is an analytic function in the forward tube $I_n$. Then we use the Lorentz invariance (W2) to show that $W_n$ has analytic continuation to the "extended tube" $\tilde{I}_n$, which includes the set of Jost points. Finally we use the microscopic causality condition (W5) to show that $\mathcal{W}_{n+1}$ has analytic continuation to all configurations with totally space-like separations.

The unitarity condition (W3) is not involved in the above argument, only the conditions (W1), (W2), (W4) and (W5) are used.

In the rest of the paper we explored the domain of CFT four-point functions by assuming unitarity, Euclidean conformal invariance and OPE (not assuming Wightman axioms). Here we would like to discuss a related but different situation:

- What is the domain of the four-point function if we only assume Wightman axioms and conformal invariance (not assuming OPE)?

We want to emphasize that global conformal invariance does not hold for general CFT in $\mathbb{R}^{d-1,1}$ because Lorentzian special conformal transformations may violate causal orderings. The precise meaning of conformal invariance here is the Euclidean global conformal invariance: we Wick rotate Wightman functions to the Euclidean signature, then the corresponding Euclidean correlation functions are invariant under Euclidean global conformal transformations. This assumption is called *weak conformal invariance*[58].

It is obvious that the Wightman function $\mathcal{W}_4$ is analytic in $\mathcal{U}_4$ (as discussed in section 6.2). In section 6.2, a crucial step is to extend the real Poincaré invariance to complex Poincaré invariance. Then the reduced Wightman function $W_3$ has analytic continuation from the forward tube $I_3$ to the extended forward tube $\tilde{I}_3$. Now that we assume weak conformal invariance, given any Euclidean conformal transformation $g$ and Euclidean configuration $C = (x_1, x_2, x_3, x_4)$, we have

$$
\begin{aligned}
\mathcal{W}_4(C) =& \Omega(x_1)^\Delta \Omega(x_2)^\Delta \Omega(x_3)^\Delta \Omega(x_4)^\Delta \mathcal{W}_4(g \cdot C), \\
g \cdot C =& (g \cdot x_1, g \cdot x_2, g \cdot x_3, g \cdot x_4),
\end{aligned} \tag{140}
$$

where $\Omega(x)$ is the scaling factor of the conformal transformation. It is not hard to show that given any configuration $C$ (which can be non-Euclidean) in the domain of $\mathcal{W}_4$, eq. (140) holds for $g$ in a neighbourhood of the identity element in the Euclidean conformal group (this neighbourhood depends on configuration). Then we can show that for a fixed $C$, eq. (140) holds not only in a neighbourhood of identity element in the Euclidean conformal group, but also in a neighourhood in the complex conformal group.[28] Therefore, by using (140) with $g$ in the complex conformal group, we can extend $\mathcal{W}_4$ to a bigger domain than $\mathcal{U}_4$ (recall definition (130)).[29] We say that configurations $C = (x_1, x_2, x_3, x_4), C' = (x_1', x_2', x_3', x_4')$ are conformally equivalent if there exists a path $g(s)$ in the complex conformal group such that

- $g(0) =$ id, and $g(1) \cdot C_1 = C_2$.

- The scaling factors $\Omega(x_1), \Omega(x_2), \Omega(x_3), \Omega(x_4)$ along $g(s)$ do not go to 0 or $\infty$.

We define

$$
\mathcal{U}_4^c = \left\{ C = (x_1, x_2, x_3, x_4) \in \mathbb{C}^{4d} \mid C \text{ is conformally equivalent to some C}' \in \mathcal{U}_4 \right\}, \tag{141}
$$

where the superscript "c" in $\mathcal{U}_4^c$ means "conformal". Naively, one may expect that $\mathcal{W}_4$ has analytic continuation to $\mathcal{U}_4^c$ by (140).[30] However, for each conformally equivalent $C, C'$ pair in $\mathcal{U}_4^c$, the path $g(s)$ in the complex conformal group is not unique, which means choosing different $g(s)$ may give different analytic continuation. In other words, $\mathcal{W}_4$ may not be a single-valued function on $\mathcal{U}_4^c$.

---

[28]The complex conformal group is generated by translations $x \to x+a$, rotations $x \to \exp\left[\alpha^{\mu\nu} M_{\mu\nu}\right] x$, dilatations $x \to e^\tau x$, special conformal translations $x \to x'^\mu = \frac{x^\mu - x^2 b^\mu}{1 - 2b \cdot x + b^2 x^2}$ with complex parameters.

[29]This can be called conformal extension of Bargmann-Hall-Wightman theory. We were unable to find this idea in prior literature.

[30]As a next step, one could consider the Lorentzian configurations in envelope of holomorphy $H(\mathcal{U}_4^c)$.

There is one very simple partial case which is guaranteed not to lead to multi-valuedness. It is the case when the whole curve $C(s) = g(s) \cdot C = (x_1(s), x_2(s), x_3(s), x_4(s))$ has point differences $(x_2(s) - x_1(s), x_3(s) - x_2(s), x_4$ in the forward tube $I_3$, except for the end point $g(1) \cdot C$.

In this paper we do not fully explore the Lorentzian domain of $\mathcal{W}_4$ by using the above method. We left it for future work. Here we only give a simple example which shows that by assuming weak conformal invariance in Wightman QFT, the domain of the Wightman function contains more Lorentzian configurations than totally space-like configurations.

We would like to show that the following Lorentzian configurations are in the domain of (conformally invariant) $\mathcal{W}_4$:

$$x_k = (t_k, \mathbf{x}_k), \quad 1 \to 2 \to 3 \to 4. \tag{142}$$

We act with complex dilatation on (142):

$$x'_k = e^{i\alpha} x_k. \tag{143}$$

Then the point differences are given by $x'_{jk} = x_{jk} \cos\alpha + i x_{jk} \sin\alpha$. By the causal ordering in (142), we have $(x'_2 - x'_1, x'_3 - x'_2, x'_4 - x'_3) \in I_3$ when $0 < \alpha \leq 1$. On the other hand, the scaling factors of complex dilatations (143) are constants: $\Omega(x) = e^{i\alpha}$, which are finite and non-zero. Thus, we can use the above-mentioned simple partial case. We conclude that the Lorentzian configurations in the form of (142) are in the domain of (conformally invariant) $\mathcal{W}_4$.

### 6.3.4 Domain of analyticity: unitarity + conformal invariance + OPE

Now back to our CFT construction in this paper. We assumed unitarity, conformal invariance and OPE, but did not assume Wightman axioms.

With the help of OPE, we are able to control the upper bound of the CFT four-point function more efficiently [9]. It seems to be rather difficult to apply the unitarity condition (114) in a general Wightman QFT because it is a non-linear constraint. While, in the expansion (22), we are able to use the unitarity condition for CFT four-point functions.

The domain of the Lorentzian CFT four-point function $G_4$ contains all configurations where there exists at least one convergent OPE channel. The results in appendix C show that the domain of $G_4$ contains much richer set of causal orderings than just the totally space-like causal ordering obtainable from Wightman axioms alone.

One interesting point is that if we act with conformal transformations on the configurations which have point differences in the forward tube $I_3$ (let us call this set $\mathcal{I}_4$), we can get at most Lorentzian configurations whose cross-ratios can be realized by configurations in $\mathcal{I}_4$ because conformal transformations do not change cross-ratios.[31] However, if we additionally assume OPE, then it is not necessary that the cross-ratios of the Lorentzian configurations can be realized by configurations in $\mathcal{I}_4$ (the only requirement is that $|\rho|, |\bar{\rho}| < 1$ in the corresponding OPE channel). It would be interesting to figure out:

- can we get more Lorentzian configurations by assuming unitarity + conformal invariance + OPE than by assuming Wightman axioms + conformal invariance?

We left it for future work.

---

[31] The similar idea has been used to look for the domain of analyticity of the Wightman functions in a general QFT. E.g. G. Källén explored the domain of the Wightman four-point function by studying configurations $(x_1, x_2, x_3, x_4)$ such that the Poincaré invariants $x_{ij} \cdot x_{kl}$ can be realized by configurations in $\mathcal{I}_4$ [66].

# 7 Four-point functions of non-identical scalar or spinning operators

## 7.1 Generalization to the case of non-identical scalar operators

In this section we generalize our results of section 3, 4 and 5 to the CFT four-point functions of non-identical scalar operators. We start from a Euclidean CFT four-point function

$$G_{1234}(x_1, x_2, x_3, x_4) := \langle \mathcal{O}_1(x_1)\mathcal{O}_2(x_2)\mathcal{O}_3(x_3)\mathcal{O}_4(x_4)\rangle . \tag{144}$$

The scalar operators $\mathcal{O}_i$ in (144) have scaling dimensions $\Delta_i$. By conformal symmetry, (144) can be factorized as

$$G_{1234}(x_1, x_2, x_3, x_4) = \frac{1}{\left(x_{12}^2\right)^{\frac{\Delta_1+\Delta_2}{2}} \left(x_{34}^2\right)^{\frac{\Delta_3+\Delta_4}{2}}} \left(\frac{x_{24}^2}{x_{14}^2}\right)^{\frac{\Delta_1-\Delta_2}{2}} \left(\frac{x_{14}^2}{x_{13}^2}\right)^{\frac{\Delta_3-\Delta_4}{2}} g_{1234}(\rho, \bar{\rho}) \tag{145}$$

As we discussed in section 3.1, the prefactor multiplying $g_{1234}(\rho, \bar{\rho})$ has analytic continuation to the domain $\mathcal{D}$. It remains to show that $g_{1234}(\tau_k, \mathbf{x}_k)$ has analytic continuation to $\mathcal{D}$.

In the region $|\rho|, |\bar{\rho}| < 1$ in the Euclidean signature, $g_{1234}$ has a convergent expansion in conformal blocks:

$$g_{1234}(\rho, \bar{\rho}) = \sum_{\mathcal{O}} C_{12\mathcal{O}}C_{34\mathcal{O}} g_{\mathcal{O}}^{12,34}(\rho, \bar{\rho}) \tag{146}$$

where $C_{ij\mathcal{O}}$ are OPE coefficients and $\mathcal{O}$ are the primary operators which appear in the $\mathcal{O}_1 \times \mathcal{O}_2$ OPE. In the unitary CFTs, we can assume $C_{ij\mathcal{O}}$ to be real by choosing proper basis of operators.

Each conformal block $g_{\mathcal{O}}^{12,34}(\rho, \bar{\rho})$ has a series expansion

$$g_{\mathcal{O}}^{12,34}(\rho, \bar{\rho}) = \sum_{\psi \in [\mathcal{O}]} a_{12\psi} b_{34\psi} \rho^{h(\psi)} \bar{\rho}^{\bar{h}(\psi)} \tag{147}$$

where $\psi \in [\mathcal{O}]$ form an orthonormal basis in the highest weight representation $[\mathcal{O}]$ of the conformal group $SO(1, d+1)$. Here we choose $\psi$ to be eigenstates of Virasoro generators $L_0, \bar{L}_0 \in so(1, d+1)$ and $h(\psi), \bar{h}(\psi)$ are the corresponding eigenvalues. The real coefficients $a_{12\psi}$, $b_{34\psi}$ in (147) are totally fixed by conformal symmetry and $a_{12\mathcal{O}} = b_{34\mathcal{O}} = 1$, where $\mathcal{O}$ is the primary state in $[\mathcal{O}]$.

We stick (147) into (146) and apply the Cauchy-Schwartz inequality:

$$
\begin{aligned}
\left|g^{12,34}(\rho, \bar{\rho})\right|^2 &= \left|\sum_{\mathcal{O}} C_{12\mathcal{O}}C_{34\mathcal{O}} \sum_{\psi \in [\mathcal{O}]} a_{12\psi} b_{34\psi} \rho^{h(\psi)} \bar{\rho}^{\bar{h}(\psi)}\right|^2 \\
&\leq \left|\sum_{\mathcal{O}} \sum_{\psi \in [\mathcal{O}]} \left|C_{12\mathcal{O}}C_{34\mathcal{O}} a_{12\psi} b_{34\psi} \rho^{h(\psi)} \bar{\rho}^{\bar{h}(\psi)}\right|\right|^2 \\
&\leq \left(\sum_{\mathcal{O}} \sum_{\psi \in [\mathcal{O}]} C_{12\mathcal{O}}^2 a_{12\psi}^2 |\rho|^{h(\psi)} |\bar{\rho}|^{\bar{h}(\psi)}\right) \\
&\quad \times \left(\sum_{\mathcal{O}} \sum_{\psi \in [\mathcal{O}]} C_{34\mathcal{O}}^2 b_{34\psi}^2 |\rho|^{h(\psi)} |\bar{\rho}|^{\bar{h}(\psi)}\right) .
\end{aligned}
\tag{148}
$$

Since the primaries $\mathcal{O}$ are in the intersection of $\mathcal{O}_1 \times \mathcal{O}_2$ and $\mathcal{O}_3 \times \mathcal{O}_4$ OPE, we have

$$
\begin{aligned}
\sum_{\mathcal{O}} \sum_{\psi \in [\mathcal{O}]} C_{12\mathcal{O}}^2 a_{12\psi}^2 |\rho|^{h(\psi)} |\bar{\rho}|^{\bar{h}(\psi)} &\leq g^{12,21}(r,r) \\
\sum_{\mathcal{O}} \sum_{\psi \in [\mathcal{O}]} C_{34\mathcal{O}}^2 b_{34\psi}^2 |\rho|^{h(\psi)} |\bar{\rho}|^{\bar{h}(\psi)} &\leq g^{43,34}(r,r)
\end{aligned}
\tag{149}
$$

where $r = \max\{|\rho|, |\bar{\rho}|\}$. The above inequalities show that the $\rho$-expansion of $g^{12,34}$ is absolutely convergent in the domain $|\rho|, |\bar{\rho}| < 1$. As a result we have an analytic function $g_{1234}(\chi, \bar{\chi})$ in the domain (28), where $\chi, \bar{\chi}$ are defined in (27).

On the other hand, for the case of non-identical scalar operators, we can still rearrange the series (147) in the same way as (24). This follows from the fact that only the operators in traceless totally-symmetric representations of $SO(d)$ can appear in the OPE of two scalar operators, which leads to $\rho$-expansion in the form of Gegenbauer polynomials (25) [13]. Therefore, by the results in section 3.4.2, the function $g_{1234}(\tau_k, \mathbf{x}_k)$ is analytic in $\mathcal{D}$.

The remaining steps are the same as the case of identical scalar operators. We conclude that for the case of four-point functions with non-identical scalar operators, the OPE convergence properties are the same as the case of identical scalar operators.

## 7.2 Comments on the case of spinning operators

Before finishing this section we want to make some comments on the case of four-point functions with spinning operators:

$$
G_{1234}^{a_1 a_2 a_3 a_4}(x_1, x_2, x_3, x_4) := \langle \mathcal{O}_1^{a_1}(x_1) \mathcal{O}_2^{a_2}(x_2) \mathcal{O}_3^{a_3}(x_3) \mathcal{O}_4^{a_4}(x_4) \rangle .
\tag{150}
$$

where $\mathcal{O}_i^{a_i}$ are primary operators with scaling dimensions $\Delta_i$ and $SO(d)$-representation $\rho_i$. $a_i$ are the indices for the spin representations $\rho_i$. In the Euclidean signature, the Jacobian of any conformal transformation $f$ in $SO(1, d+1)$ can be factorized as

$$
J_\nu^\mu(x) := \frac{\partial f^\mu(x)}{\partial x^\nu} = \Omega(x) \mathcal{R}_\nu^\mu(x),
\tag{151}
$$

where $\Omega(x) > 0$ is a scaling factor and $\mathcal{R}$ is a rotation matrix. The four-point function $G_{1234}^{a_1 a_2 a_3 a_4}$ is invariant if we replace all $\mathcal{O}_i^{a_i}(x)$ in (150) with

$$
\mathcal{O}_i^{a_i}(x) \to \Omega(x)^{\Delta_i} \left[ \rho_i \left( \mathcal{R}(x) \right)^{-1} \right]_{b_i}^{a_i} \mathcal{O}_i^{b_i}(f(x)) .
\tag{152}
$$

If we choose the conformal transformation $f$ to be the one which maps $(x_1, x_2, x_3, x_4)$ to its $\rho, \bar{\rho}$-configuration (19), then we get

$$
\begin{aligned}
G_{1234}^{a_1 a_2 a_3 a_4}(x_1, x_2, x_3, x_4) = &\frac{1}{\left(x_{12}^2\right)^{\frac{\Delta_1 + \Delta_2}{2}} \left(x_{34}^2\right)^{\frac{\Delta_3 + \Delta_4}{2}}} \left(\frac{x_{24}^2}{x_{14}^2}\right)^{\frac{\Delta_1 - \Delta_2}{2}} \left(\frac{x_{14}^2}{x_{13}^2}\right)^{\frac{\Delta_3 - \Delta_4}{2}} \\
&\times T_{b_1 b_2 b_3 b_4}^{a_1 a_2 a_3 a_4}(\mathcal{R}_1, \mathcal{R}_2, \mathcal{R}_3, \mathcal{R}_4) g_{1234}^{b_1 b_2 b_3 b_4}(\rho, \bar{\rho})
\end{aligned}
\tag{153}
$$

where $\mathcal{R}_k$ is the rotation matrix of $f$ at $x_k$, and $T$ is a function of rotation matrices, which is determined by the representations $\rho_i$ of the spinning operators $\mathcal{O}_i$.

Analogously to the scalar case, the function $g_{1234}^{b_1 b_2 b_3 b_4}(\rho, \bar{\rho})$ can be bounded by Cauchy-Schwarz inequality. We can still write $g_{1234}^{b_1 b_2 b_3 b_4}(\rho, \bar{\rho})$ in the form of (24), but in general the functions $P^l(\rho, \bar{\rho})$ are not Gegenbauer polynomials of $\cos\theta$ (recall that $\frac{\rho}{\bar{\rho}} = e^{2i\theta}$). The difficulty is that there is the function $T(\mathcal{R}_k)$ in (153) because of the non-trivial representations of the spinning operators. If we think of the above conformal transformation $f$ as a conformal-group-valued function of $(x_1, x_2, x_3, x_4)$,[32] then $\mathcal{R}_k$ are also functions of $(x_1, x_2, x_3, x_4)$. In general, the entries of $\mathcal{R}_k$ have singularities in the domain $\mathcal{D}$, e.g. at $\Gamma$ [67, 68]. In a word, it is not obvious that $T(\mathcal{R}_k)$ in (153) is under control. Some extra work is required for a good estimate on the object

$$T_{b_1 b_2 b_3 b_4}^{a_1 a_2 a_3 a_4}(\mathcal{R}_1, \mathcal{R}_2, \mathcal{R}_3, \mathcal{R}_4) g_{1234}^{b_1 b_2 b_3 b_4}(\rho, \bar{\rho})$$

In this paper we do not study the correlators of spinning operators. We leave these for future work.

# 8 Conclusions and outlooks

In this work we studied the convergence properties of various OPE channels for Lorentzian CFT four-point functions of scalar operators in $d \geq 2$, assuming global conformal symmetry. Our analysis is based on the convergence properties of OPE in the Euclidean unitary CFTs. We classified the Lorentzian four-point configurations according to their causal orderings and the range of the variables $z, \bar{z}$. The Lorentzian correlators are analytic functions in a neighbourhood of some Lorentzian configuration as long as there exists at least one convergent OPE channel in the sense of functions. We showed that the convergence properties of various OPE channels are determined by the causal orderings and the range of $z, \bar{z}$ of the four-point configurations. The CFT four-point functions are analytic in a very big domain, including configurations with totally space-like separations and configurations with some other causal orderings. All the results of OPE convergence properties are given in Appendix C.

Before ending, we would like to point out some related open questions. We list these questions in the order of difficulty (based on personal perspective):

1. (Easy) CFTs can also live on the Minkowski cylinder [58]. We start from CFT four-point function defined on the Euclidean cylinder and then Wick rotate the cylindrical time variables. The corresponding counterpart questions are:

   - Which configurations have convergent s-, t- and u-channel OPE in the sense of functions on the Minkowski cylinder?
   - Can we still classify the OPE convergence properties according to the range of $z, \bar{z}$ and the causal orderings on the Minkowski cylinder?

2. We mainly used the radial coordinates $\rho, \bar{\rho}$ in our analysis. We have seen that by using the $q, \bar{q}$-variables in 2d, the domain of CFT four-point functions are larger than the domain derived by using the $\rho, \bar{\rho}$-variables. A natural question is

   - For CFTs with only global conformal symmetry, are there any other coordinates which allow us to extend $G_4$ to some other domains which are not covered by using radial variables?

   Our conjecture is that there are no such coordinates.

---

[32]Since such a conformal transformation $f$ is not unique, the definition of this group-valued function depends on how we construct it.

3. (Probably hard) Our results provide some safe Lorentzian regions where conformal bootstrap approach can be applied. One can use bootstrap equations to analyze the four-point functions at Lorentzian configurations with at least two convergent OPE channels. It is also interesting to play with crossing symmetry at Lorentzian configurations with

   - One convergent OPE channel in the sense of functions, another one in the sense of distributions.
   - Two convergent OPE channels in the sense of distributions.[33]

   The above situations are closely related to the topics on analytic functional bootstrap when the functionals touch the boundaries of the regions with convergent OPE [24, 25, 26, 27, 70, 71, 72].

4. (Hard) There are also Lorentzian configurations with no convergent OPE channels. For these cases we do not know whether the general four-point correlators are genuine functions or not. We may need other techniques to handle these situations. For example, there are questions similar to section 6.2.4:

   - One can derive a complex domain $\mathcal{D}^{stu}$ which is the union of the domains of three OPE channels. Then what is $\mathcal{D}^{stu}$ and what is its envelope of holomorphy $H\left(\mathcal{D}^{stu}\right)$? Does $H\left(\mathcal{D}^{stu}\right)$ contain more Lorentzian configurations than those provided by the results in this paper?

   Once we are able to construct $H\left(\mathcal{D}^{stu}\right)$, one can ask

   - Given a Lorentzian configuration $C_L \in \mathcal{D}\backslash H\left(\mathcal{D}^{stu}\right)$, can we find a CFT example such that the four-point function is divergent at $C_L$?

5. (Hard) One can also consider higher-point correlation functions in CFTs. A natural question is:

   - For $n \geq 5$, what is the Lorentzian domain of $G_n$ in the sense of functions?

We leave these questions for future work.

# Acknowledgments

JQ thanks Slava Rychkov and Petr Kravchuk for a lot of helpful discussions. This work is a by-product of our collaboration work of a series of papers on distributional properties of four-point functions in CFTs. JQ also thanks Slava Rychkov for the great help in improving the writing of this paper.

JQ thanks Zechuan Zheng, Emilio Trevisani, Dan Mao and Yikun Jiang for helpful discussions and comments.

The work of JQ is supported by the Simons Foundation grant 488655 (Simons Collaboration on the Nonperturbative Bootstrap).

# A  Why $z(\tau_k, \mathbf{x}_k)$, $\bar{z}(\tau_k, \mathbf{x}_k)$ are not well defined on $\mathcal{D}$ in $d \geq 3$ ?

## A.1  Main idea

Recall that we want to analytically continue the function $g(\tau_k, \mathbf{x}_k)$ to $\mathcal{D}$ by using the compositions (30). In fact, by theorem 3.1 and the fact that $\mathcal{D}$ is simply connected, this would have been possible if analytic functions $z(\tau_k, \mathbf{x}_k), \bar{z}(\tau_k, \mathbf{x}_k)$ existed (see the 2d case in section 3.5).

---

[33]The similar idea was proposed recently by Gillioz et al, see [69], section 5.

We are going to show that in $d \geq 3$, there is no analytic function $(z(\tau_k, \mathbf{x}_k), \bar{z}(\tau_k, \mathbf{x}_k))$ on $\mathcal{D}$ which is compatible with (6) and (7). Consequently, some modification of the strategy (30) is needed to construct the analytic continuation of $g(\tau_k, \mathbf{x}_k)$ in $d \geq 3$ (and that's what we did in section 3.4). In this section we do not identify $(z, \bar{z}) \sim (\bar{z}, z)$.

Let us first give the main idea why $z(\tau_k, \mathbf{x}_k), \bar{z}(\tau_k, \mathbf{x}_k)$ are not globally well-defined on $\mathcal{D}$. This is similar to the reason why the square-root function $f(w) = \sqrt{w}$ is not globally well-defined on $\mathbb{C}$: there exits a loop of $w$ such that $f(w)$ starts with $f(w_0)$ ($w_0$ is the start and final point of the loop) but ends with $-f(w_0)$. In $d \geq 3$, there exists a loop $\gamma$ in $\mathcal{D}$ such that the value $4u - (1 + u - v)^2$ goes around 0 once. Then by (8), we see that $(z(\gamma(s)), \bar{z}(\gamma(s)))$ starts with $(z_0, \bar{z}_0)$ but ends with $(\bar{z}_0, z_0)$. The construction of such a path $\gamma$ is given by (161) and (162).[34]

## A.2 Proof

Let $\Gamma$ be the preimage of $\Gamma_{uv}$ in $\mathcal{D}$ (i.e. the set of configurations where $4u - (1 + u - v)^2 = 0$, see the definition of $\Gamma_{uv}$ in (9)). A necessary condition for the existence of such a $(z, \bar{z})$ function is that

- $(z(\tau_k, \mathbf{x}_k), \bar{z}(\tau_k, \mathbf{x}_k))$ is a continuous function from $\mathcal{D}\backslash\Gamma$ to $\mathbb{C}^2\backslash\{z = \bar{z}\}$.

It suffices to show that the above condition does not hold in $d \geq 3$.

If the continuous function $(z(\tau_k, \mathbf{x}_k), \bar{z}(\tau_k, \mathbf{x}_k))$ exists, then since (11) is an invertible linear map, the continuous function $\tilde{\alpha}(\tau_k, \mathbf{x}_k) = (w(\tau_k, \mathbf{x}_k), y(\tau_k, \mathbf{x}_k))$ also exists. Then there is the following commutative diagram (the map $\alpha$ is defined by (6), while $p$ is a composition of (11) and (7)):

$$
\begin{array}{ccc}
(w, y) \in \ \mathbb{C} \times (\mathbb{C}\backslash\{0\}) & \xrightarrow{\ (11)\ } & \mathbb{C}^2\backslash\{z = \bar{z}\} \\
{\scriptstyle\tilde{\alpha}?}\nearrow\ \Big\downarrow{\scriptstyle p} & {\scriptstyle(7)}\swarrow & \\
(\epsilon_k + it_k, \mathbf{x}_k) \in \mathcal{D}\backslash\Gamma & \xrightarrow{\ \alpha\ } & \mathbb{C}^2\backslash\Gamma_{uv} \ni (u, v)
\end{array}
\tag{154}
$$

The map $\tilde{\alpha}$ in (154) satisfies $p \circ \tilde{\alpha} = \alpha$. Since $p$ is a double covering map from $\mathbb{C} \times (\mathbb{C}\backslash\{0\})$ to $\mathbb{C}^2\backslash\Gamma_{uv}$, recalling the lifting properties of the covering map [41], a sufficient and necessary condition for the existence of $\tilde{\alpha}$ is the following relation of the fundamental groups

$$
\alpha_* \Big( \pi_1 \big( \mathcal{D}\backslash\Gamma, C_0 \big) \Big) \subset p_* \Big( \pi_1 \big( \mathbb{C} \times (\mathbb{C}\backslash\{0\}), (w_0, y_0) \big) \Big),
\tag{155}
$$

where $C_0$, $(w_0, y_0)$ are base points of the corresponding spaces such that $\alpha(C_0) = p(w_0, y_0)$, and $\alpha_*$, $p_*$ are the fundamental group homomorphisms induced by $\alpha$, $p$.

It remains to show that the condition (155) does not hold in $d \geq 3$. By (10), $\mathbb{C}^2\backslash\Gamma_{uv}$ is homeomorphic to $\mathbb{C} \times (\mathbb{C}\backslash\{0\})$. So $\mathbb{C}^2\backslash\Gamma_{uv}$ and $\mathbb{C} \times (\mathbb{C}\backslash\{0\})$ have the same non-trivial fundamental group

$$
\pi_1 \big( \mathbb{C} \times (\mathbb{C}\backslash\{0\}), (w_0, y_0) \big) \simeq \pi_1 \big( \mathbb{C}^2\backslash\Gamma_{uv}, (u_0, v_0) \big) \simeq \mathbb{Z}.
\tag{156}
$$

where $(u_0, v_0)$ is a base point in $\mathbb{C}^2\backslash\Gamma_{uv}$ such that $p(w_0, y_0) = (u_0, v_0)$. Since $p$ is a double covering map, the group homomorphism

$$
p_* : \ \pi_1 \big( \mathbb{C} \times (\mathbb{C}\backslash\{0\}), (w_0, y_0) \big) \ \longrightarrow \ \pi_1 \big( \mathbb{C}^2\backslash\Gamma_{uv}, (u_0, v_0) \big)
\tag{157}
$$

---

[34] In the proof we will use a formal way to reformulate the idea described here. Readers who got the idea can just go to the construction of the path. We thank Petr Kravchuk for the discussion on this problem.

is not surjective. In fact, by choosing proper basis, the group homomorphism (157) can be written as

$$p_* : \begin{aligned} \mathbb{Z} &\longrightarrow \mathbb{Z} \\ n &\mapsto 2n \end{aligned} \tag{158}$$

Now to show that the condition (155) does not hold, it suffices to show that the group homomorphism

$$\alpha_* : \pi_1\big(\mathcal{D}\backslash\Gamma, C_0\big) \longrightarrow \pi_1\big(\mathbb{C}^2\backslash\Gamma_{uv}, (u_0, v_0)\big) \tag{159}$$

is surjective. By (10), the generator of $\pi_1\big(\mathbb{C}^2\backslash\Gamma_{uv}, (u_0, v_0)\big)$ corresponds to a loop in $\mathbb{C}^2\backslash\Gamma_{uv}$ such that along this loop, $4u - (1 + u - v)^2$ goes around 0 only once. Let us construct a loop $\gamma$ in $\mathcal{D}\backslash\Gamma$:

$$\gamma : [0, 1] \longrightarrow \mathcal{D}\backslash\Gamma, \quad \gamma(0) = \gamma(1) = C_0, \tag{160}$$

which induces a loop in $\mathbb{C}^2\backslash\Gamma_{uv}$ such that $4u - (1 + u - v)^2$ goes around 0 only once.

We pick 4 configurations $C_0, C_1, C_2, C_3$ in $\mathcal{D}\backslash\Gamma$:

$$\begin{aligned} C_0 : \quad & x_1 = 0, \ x_2 = (-1, 1, 0, \dots, 0), \ x_3 = (-2, 0, \dots, 0), \ x_4 = (-5, 0, \dots, 0). \\ C_1 : \quad & x_1 = (0, -1, 0, \dots, 0), \ x_2 = (-1 + 3i, -1, 0, \dots, 0), \ x_3 = (-2, 0, \dots, 0), \\ & x_4 = (-5, 0, \dots, 0). \\ C_2 : \quad & x_1 = 0, \ x_2 = (-1, -1, 0, \dots, 0), \ x_3 = (-2, 0, \dots, 0), \\ & x_4 = (-5, 0, \dots, 0). \\ C_3 : \quad & x_1 = (0, 1, 0, \dots, 0), \ x_2 = (-1, 0, 1, 0, \dots, 0), \ x_3 = (-2, 0, \dots, 0), \\ & x_4 = (-5, 0, \dots, 0). \end{aligned} \tag{161}$$

We let $\gamma$ be consisting of four straight lines:

$$\gamma(s) = \begin{cases} (1 - 4s)C_0 + 4sC_1, & 0 \le s \le 1/4, \\ (2 - 4s)C_1 + (4s - 1)C_2, & 1/4 \le s \le 1/2, \\ (3 - 4s)C_2 + (4s - 2)C_3, & 1/2 \le s \le 3/4, \\ (4 - 4s)C_3 + (4s - 3)C_0, & 3/4 \le s \le 1. \end{cases} \tag{162}$$

Figure A.1 shows the curve of $4u - (1 + u - v)^2$ along the path $\gamma$. We see that $4u - (1 + u - v)^2$ is non-zero everywhere, implies that the configurations along $\gamma$ are always in $\mathcal{D}\backslash\Gamma$ (because the image $(u, v)$ of $\gamma(s)$ is not in $\Gamma_{uv}$). Furthermore, $4u - (1 + u - v)^2$ goes around 0 only once. Therefore, fundamental group element $[\gamma] \in \pi_1\big(\mathcal{D}\backslash\Gamma, C_0\big)$ is mapped to the generator of $\pi_1\big(\mathbb{C}^2\backslash\Gamma_{uv}, (u_0, v_0)\big)$, so (159) is surjective. As a result, the lifted map $\tilde{\alpha}$ does not exist.

We conclude that in $d \ge 3$, there is no analytic function $(z(\tau_k, \mathbf{x}_k), \bar{z}(\tau_k, \mathbf{x}_k))$ on $\mathcal{D}$.

We also want to remark that in 2d, the analytic function $(z, \bar{z})$ does exist (see section 3.5). The part of our proof in this section that fails in 2d is the construction of the loop $\gamma$ in (162). In our construction of $C_3$ in (161), the third component of $x_2$ is non-zero. This is where we use the condition $d \ge 3$. In fact, one can show that in 2d, the group homomorphism (159) is not surjective, and the condition (155) holds.

# B    Connectedness of $\mathcal{D}_L^\alpha$ in $d \ge 3$

In this section we are going to show that in $d \ge 3$, each $\mathcal{D}_L^\alpha$ in (90) is connected.

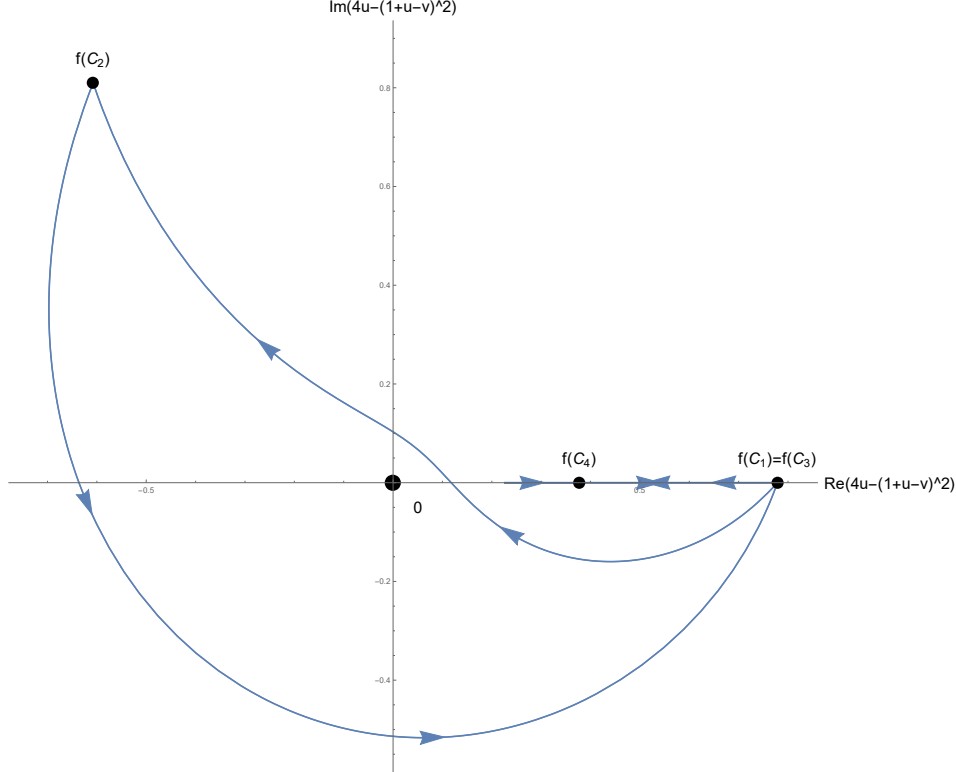

Figure A.1: The curve of $4u - (1 + u - v)^2$ along $\gamma(s)$. We use $f(C_i)$ to denote the value $4u - (1 + u - v)^2$ at the configuration $C_i$.

Observe first of all that all $\mathcal{D}_L^\alpha$ of the same causal type in table 2 have the same connectedness property (this is obvious because they are related by renumbering points), so it suffices to prove the connectedness property for one $\mathcal{D}_L^\alpha$ in each causal type.

Given a $\mathcal{D}_L^\alpha$, we define $(\mathcal{D}_L^\alpha)_3$ to be the set of all three-point Lorentzian configurations which have the causal ordering of the first three points of the configurations in $\mathcal{D}_L^\alpha$. Then there is a natural projection from $\mathcal{D}_L^\alpha$ to $(\mathcal{D}_L^\alpha)_3$:

$$
\begin{aligned}
\pi : \mathcal{D}_L^\alpha &\longrightarrow (\mathcal{D}_L^\alpha)_3 , \\
(x_1, x_2, x_3, x_4) &\mapsto (x_1, x_2, x_3),
\end{aligned}
\tag{163}
$$

Then $\mathcal{D}_L^\alpha$ has the following decomposition

$$
\mathcal{D}_L^\alpha = \bigcup_{(x_1, x_2, x_3) \in (\mathcal{D}_L^\alpha)_3} \left\{ x_4 \mid (x_1, x_2, x_3, x_4) \in \mathcal{D}_L^\alpha \right\} .
\tag{164}
$$

For each causal type in table 2, we want to show that there exists a $\mathcal{D}_L^\alpha$ in this causal type such that

1. For fixed $(x_1, x_2, x_3) \in (\mathcal{D}_L^\alpha)_3$, the set $F_{x_1, x_2, x_3}^\alpha = \left\{ x_4 \mid (x_1, x_2, x_3, x_4) \in \mathcal{D}_L^\alpha \right\}$ is non-empty and connected.

2. $(\mathcal{D}_L^\alpha)_3$ is connected.

## B.1  Step 1

For a fixed three-point configuration $(x_1, x_2, x_3)$, the set $F^\alpha_{x_1,x_2,x_3} = \left\{ x_4 \mid (x_1, x_2, x_3, x_4) \in \mathcal{D}^\alpha_L \right\}$ is non-empty and connected if one of the following conditions holds as a consequence of causal ordering imposed by $\mathcal{D}^\alpha_L$:

1. $x_i \rightarrow x_4$ for $i = 1, 2, 3$.

2. $x_4 \rightarrow x_i$ for $i = 1, 2, 3$.

3. $x_4$ is space-like separated from all of $x_1, x_2, x_3$.

For the first case, $F^\alpha_{x_1,x_2,x_3}$ is given by the intersection of the open forward light-cones of $x_1, x_2, x_3$, which is non-empty. Since cones are convex, $F^\alpha_{x_1,x_2,x_3}$ is also convex, thus connected. The connectedness for the second case follows from a similar argument. For the third case, we use the fact that the connectedness property does not change under Poincaré transformations, which allows us to move $x_1$ to 0 by translation

$$x_k \mapsto x_k - x_1, \quad k = 1, 2, 3, 4, \tag{165}$$

and then move $x_2, x_3$ onto a 2d subspace by a Lorentz transformation. We enumerate all possible three-point causal orderings

$$a \longrightarrow b \longrightarrow c \ , \quad a \begin{matrix} \nearrow b \\ \searrow c \end{matrix} \ , \quad \begin{matrix} b \searrow \\ c \nearrow \end{matrix} a \ , \quad b \longrightarrow c \ , \quad \begin{matrix} a \\ b \\ c \end{matrix} \ , \tag{166}$$

and check case by case that in $d \geq 3$ we can always move the extra point $x_4$ from any position to $\infty$, preserving the constraint that $x_4$ is space-like to $a, b, c$. This observation implies that $F^\alpha_{x_1,x_2,x_3}$ is connected for the third case.

In table 2, we find a $\mathcal{D}^\alpha_L$ satisfying one of the above conditions for some but not all causal types. That's why the connectedness of $\mathcal{D}^\alpha_L$ is not so obvious. The exceptional cases are causal type 8, 10 and 11, for which we need to discuss case by case. Without loss of generality we set $a = 1$, $b = 2$, $c = 3$, $d = 4$ (comparing with table 2) in the following discussion.

**Type 8.** By translations, Lorentz transformations and dilatations we fix the configurations to

$$x_1 = 0, \quad x_2 = (x^0, x^1, 0, \ldots, 0), \quad x_3 = (0, 1, 0, \ldots, 0). \tag{167}$$

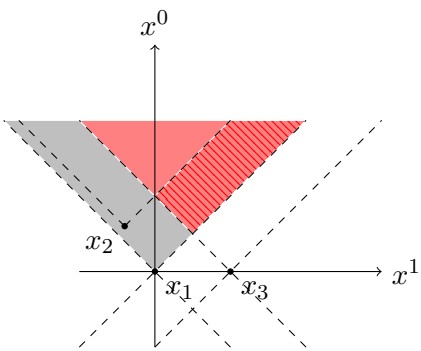

Figure B.1: Type 8

Then $x_2$ is in the open forward light-cone of $x_1$, but out of the light-cones of $x_3$ (see the grey region in figure B.1), and $x_4$ is in the intersection of open forward light-cones of $x_1$ and $x_3$ (see the red region in figure B.1). Once $x_2$ is fixed somewhere in the grey region, the space of allowed positions for $x_4$ is given by the red region minus the forward light-cone of $x_2$, so the remaining region for $x_4$, which is $F^{\alpha}_{x_1,x_2,x_3}$, is the red dashed region in figure B.1. Figure B.1 shows the 2d situation but a similar 3d figure shows that $F^{\alpha}_{x_1,x_2,x_3}$ is non-empty and connected in 3d, thus also non-empty and connected in higher d (because we can always find a spatial rotation which preserves $x_1, x_2, x_3$ and maps $x_4$ to $(x, y, z, 0, \ldots, 0)$).

**Type 10.** By translations, Lorentz transformations and dilatations we fix the configurations to

$$x_1 = 0, \quad x_2 = (0, 1, 0 \ldots, 0), \quad x_3 = (x^0, x^1, 0, \ldots, 0). \tag{168}$$

Figure B.2: Type 10

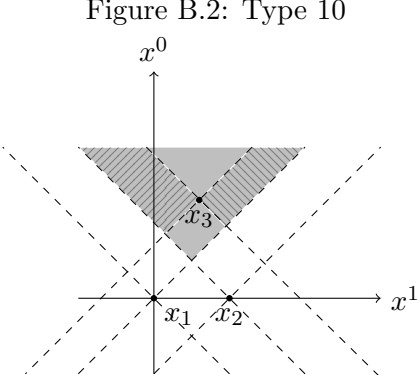

The remaining $x_3, x_4$ pair are in the intersection of the open forward light-cones of $x_1, x_2$, i.e. the grey region in figure B.2. Once $x_3$ is fixed, by the constraint that $x_3, x_4$ are space-like separated, $F^{\alpha}_{x_1,x_2,x_3}$ is given by the grey dashed region in figure B.2, which is obviously non-empty. This region is topologically the same as $\mathbb{R}^d$ minus the light-cones of $x_3$, thus connected when $d \geq 3$.

**Type 11.** By translations, Lorentz transformations and dilatations we fix the configurations to

$$x_1 = 0, \quad x_2 = (i, 0, \ldots, 0), \quad x_3 = (x^0, x^1, 0, \ldots, 0) \tag{169}$$

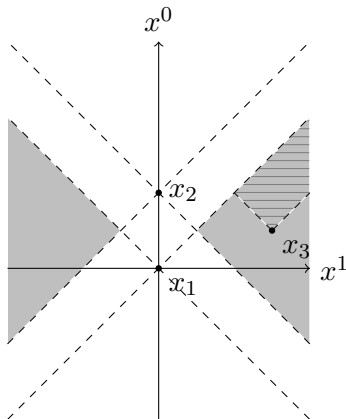

Figure B.3: Type 11

The remaining $x_3, x_4$ pair are in the grey region in figure B.3. One can see that in $d \geq 3$ the grey region is topologically the same as the triangle slice (see one of the grey triangle in figure B.3) times $S^{d-2}$, which is connected. Once $x_3$ is fixed, $F^\alpha_{x_1, x_2, x_3}$ is given by the forward light-cone of $x_3$ in the grey region (see the grey dashed region in figure B.3), which is connected.

## B.2 Step 2

By slightly improving the argument in step 1, we claim that for the representative set $\mathcal{D}^\alpha_L$ (which we chose in step 1) of each causal type in table 2, the map (163) is surjective. In other words, for each three-point configuration in $(\mathcal{D}^\alpha_L)_3$, its preimage in $\mathcal{D}^\alpha_L$ is non-empty.

This claim is true for the cases which satisfy one of the conditions at the beginning of section B.1 because for these cases we can always find an $x_4$ which is very far away from $x_1$, $x_2$ and $x_3$. This claim is also true for the exceptional cases because from the figure B.1, B.2 and B.3 we see that the remaining region for $x_4$ is always non-empty.

Now it remains to show that $(\mathcal{D}^\alpha_L)_3$, which is the set of all three-point configurations with a fixed causal ordering, is connected. For each causal type in (166), we choose

$$x_1 = b, \quad x_2 = c, \quad x_3 = a. \tag{170}$$

Analogously to the four-point case we define a projection

$$
\begin{aligned}
\pi : (\mathcal{D}^\alpha_L)_3 &\longrightarrow (\mathcal{D}^\alpha_L)_2, \\
(x_1, x_2, x_3) &\mapsto (x_1, x_2).
\end{aligned} \tag{171}
$$

Then we decompose $(\mathcal{D}^\alpha_L)_3$ into

$$(\mathcal{D}^\alpha_L)_3 = \bigcup_{(x_1, x_2) \in (\mathcal{D}^\alpha_L)_2} \left\{ x_3 \mid (x_1, x_2, x_3) \in (\mathcal{D}^\alpha_L)_3 \right\}. \tag{172}$$

By comparing (166) and (170), we find each $(\mathcal{D}^\alpha_L)_3$ satisfies one of the following conditions:

1. $x_i \to x_3$ for $i = 1, 2$.

2. $x_3 \to x_i$ for $i = 1, 2$.

3. $x_3$ is space-like separated from both of $x_1, x_2$.

This observation implies that for each $(\mathcal{D}^\alpha_L)_3$:

- For any fixed $(x_1, x_2) \in (\mathcal{D}^\alpha_L)_2$, the set $\left\{ x_3 \mid (x_1, x_2, x_3) \in (\mathcal{D}^\alpha_L)_3 \right\}$ is connected.

- $(\mathcal{D}^\alpha_L)_2 = \pi \left( (\mathcal{D}^\alpha_L)_3 \right)$ contains all two-point configurations with the corresponding causal ordering.

It remains to show that in $d \geq 3$, the set of two-point configurations with a given causal ordering is connected. This is trivial.

# C Tables of OPE convergence

In this appendix we will give 12 tables of the results about convergence properties of three OPE channels: one table for one causal type. For each causal type we will give a template graph with points $a, b, c, d$. Given a Lorentzian configuration $C_L = (x_1, x_2, x_3, x_4) \in \mathcal{D}_L$, the way to look up the tables is as follows.

1. Compute the causal ordering of $C_L$, draw the graph of this causal ordering. Find the corresponding type number (say type X) in table 2.

2. Go to the section of causal type X (which is appendix C.X). Compare the causal ordering of $C_L$ with the template causal ordering of causal type X at the beginning of appendix C.X. Match the points $i_1, i_2, i_3, i_4$ with ($abcd$). We will get a sequence ($i_1 i_2 i_3 i_4$).

3. Look up the convergence properties of ($i_1 i_2 i_3 i_4$) in the table of causal type X.

For example, consider the following template causal ordering

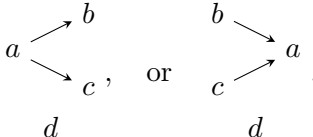

Then ($i_1 i_2 i_3 i_4$) means

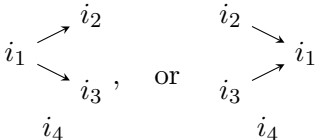

In appendix C.1 we will explain in detail how to make the table of OPE convergence for type 1 causal ordering. The procedure is similar for the other causal types, so we will only give the results for them. Before we start, we would like to introduce some tricks in appendix C.0.1, C.0.2 and C.0.3. They will be helpful in making the tables.

## C.0.1 $S_4$-action

There is a natural $S_4$-action on the space of four-point configurations. Let $\sigma \in S_4$ be a symmetry group element:

$$\sigma = \begin{pmatrix} 1 & 2 & 3 & 4 \\ \sigma(1) & \sigma(2) & \sigma(3) & \sigma(4) \end{pmatrix}. \tag{173}$$

Let $C = (x_1, x_2, x_3, x_4)$ be a four-point configuration such that $x_{ij}^2 \neq 0$ for all $x_i, x_j$ pairs. We define the action

$$\sigma \cdot C = (x_1', x_2', x_3', x_4'), \quad x_k' = x_{\sigma^{-1}(k)}. \tag{174}$$

By computing $z, \bar{z}$ of $\sigma \cdot C$ and comparing with $z, \bar{z}$ of $C$, we get a natural $S_4$-action on $z, \bar{z}$:

$$\begin{aligned} w_\sigma : \ \mathbb{C} \backslash \{0, 1\} &\longrightarrow \mathbb{C} \backslash \{0, 1\}, \\ z &\longmapsto w_\sigma(z), \\ \bar{z} &\longmapsto w_\sigma(\bar{z}), \end{aligned} \tag{175}$$

where $w_\sigma(z), w_\sigma(\bar{z})$ are the variables $z, \bar{z}$ computed from $\sigma \cdot C$. We have the following properties:

- $\{w_\sigma\}_{\sigma \in S_4}$ belong to a set of 6 fractional linear transformation forming a group which is isomorphic to $S_3$. The map $\sigma \mapsto w_\sigma$ is a group homomorphism from $S_4$ to $S_3$ (i.e. $w_{\sigma_1} \circ w_{\sigma_2} = w_{\sigma_1 \sigma_2}$).

- The $S_4$-action on $\mathcal{D}_L$ permutes classes S,T,U among themselves.

- The $S_4$-action on $\mathcal{D}_L$ permutes subclasses $E_{su}$,$E_{st}$,$E_{tu}$ among themselves.

- The $S_4$-action on $\mathcal{D}_L$ preserves the subclass $E_{stu}$.

Let us denote $\sigma$ by $[\sigma(1)\sigma(2)\sigma(3)\sigma(4)]$. We summarize the above properties in table 3.

Table 3: The list of $w_\sigma$ and the $S_4$ transformation between classes and subclasses.

| $\sigma$ | $w_\sigma(z)$ | S | T | U | $E_{su}$ | $E_{st}$ | $E_{tu}$ | $E_{stu}$ |
|---|---|---|---|---|---|---|---|---|
| [1234], [2143], [3412], [4321] | $z$ | S | T | U | $E_{su}$ | $E_{st}$ | $E_{tu}$ | $E_{stu}$ |
| [2134], [1243], [4312], [3421] | $\frac{z}{z-1}$ | S | U | T | $E_{st}$ | $E_{su}$ | $E_{tu}$ | $E_{stu}$ |
| [3214], [4123], [1432], [2341] | $1-z$ | T | S | U | $E_{tu}$ | $E_{st}$ | $E_{su}$ | $E_{stu}$ |
| [1324], [2413], [3142], [4231] | $\frac{1}{z}$ | U | T | S | $E_{su}$ | $E_{tu}$ | $E_{st}$ | $E_{stu}$ |
| [2314], [1423], [4132], [3241] | $\frac{1}{1-z}$ | T | U | S | $E_{st}$ | $E_{tu}$ | $E_{su}$ | $E_{stu}$ |
| [3124], [4213], [1342], [2431] | $1-\frac{1}{z}$ | U | S | T | $E_{tu}$ | $E_{su}$ | $E_{st}$ | $E_{stu}$ |

Suppose a configuration $C_L$ gives the template causal ordering of a causal type, which means that $C_L$ corresponds to the sequence (1234). For $\sigma = [i_1 i_2 i_3 i_4]$, we get a configuration $C'_L = \sigma \cdot C$ by eq. (174). The causal ordering of $C'_L$ is in the same causal type as $C_L$. By comparing the causal orderings of $C_L$ and $C'_L$, we see that the sequence of $C'_L$ is exactly $(i_1 i_2 i_3 i_4)$. Therefore, given a causal type, if we know the class/subclass of the template causal ordering, by looking up table 3 we decide the classes/subclasses of the other causal ordering in the same causal type. Then by looking up table 1, we immediately get a part of the OPE convergence properties for each causal ordering.

By using the above trick, the problem of determining the classes/subclasses of causal orderings belong is reduced to determining the class/subclass of the template causal ordering in each causal type. In appendix C.0.2, we will introduce a trick to determine the classes/subclasses of the template causal orderings.

### C.0.2  Lorentzian conformal frame

Our goal in this subsection is to give a systematic way to determine the class/subclass of $\mathcal{D}_L^\alpha$, where $\alpha$ is a fixed causal ordering.

Recalling lemma 5.1, all configurations in $\mathcal{D}_L^\alpha$ belong to the same class. We can choose a particular configuration $C_L \in \mathcal{D}_L^\alpha$ and compute $z, \bar{z}$ of $C_L$, then we immediately know the class (not subclass) of $\mathcal{D}_L^\alpha$.

If $\mathcal{D}_L^\alpha$ belongs to class S/T/U, then we are done. The rest of this subsection is for the case that $\mathcal{D}_L^\alpha$ belongs to class E. If $\mathcal{D}_L^\alpha$ belongs to class E, then according to theorem 5.2, we need to check the OPE convergence properties for the intersection of $\mathcal{D}_L^\alpha$ and each subclass of class E as long as the intersection is non-empty. We will find that only the type 1, 5, 6, 10, 11, 12 causal orderings in table 2 belong to class E.[35] In the tables of OPE convergence properties of causal type 5, 10 and 12 , we give the results of all

---

[35]This can be easily done by choosing one particular configuration and compute $z, \bar{z}$ for each template causal ordering, and by the fact that the $S_4$-action preserves class E (as discussed in appendix C.0.1).

subclasses for each causal ordering (see table 9, 14 and 16); while in the tables of causal type 1 ,6 and 11, we only give the results of one subclass for each causal ordering (see table 5, 10 and 15). We claim that our tables are complete, based on the following lemma.

**Lemma C.1.** Given a fixed causal ordering $\alpha$, if $\alpha$ is in causal type 1/6/11, then $\mathcal{D}_L^\alpha$ only belongs to one of the three subclasses $E_{st}, E_{su}, E_{tu}$.

The basic tool we use to prove the above lemma is the Lorentzian conformal frame. The Lorentzian conformal frame is similar to the Euclidean conformal frame (12). Given a Lorentzian configuration $C_L$, its conformal frame configuration $C_L'$ is a Lorentzian configuration which has one of the following forms

$$
\begin{aligned}
&1.\ x_1' = 0,\ x_2' = (ia, b, 0, \ldots, 0),\ x_3' = (i, 0, \ldots, 0),\ x_4' = \infty. \\
&2.\ x_1' = 0,\ x_2' = (ib, a, 0, \ldots, 0),\ x_3' = (0, 1, 0, \ldots, 0),\ x_4' = \infty. \\
&3.\ x_1' = 0,\ x_2' = (0, a, b, 0, \ldots, 0),\ x_3' = (0, 1, 0, \ldots, 0),\ x_4' = \infty.
\end{aligned}
\tag{176}
$$

$C_L'$ and $C_L$ are related by a Lorentzian conformal transformation. Computing the cross-ratios from (176), we see that: for the first and second cases $z = a + b$, $\bar{z} = a - b$; for the third case $z = a + ib$, $\bar{z} = a - ib$. Analogously to the Euclidean conformal frame, the Lorentzian conformal frame configuration is unique up to a reflection $b \mapsto -b$, which corresponds to interchanging $z$ and $\bar{z}$.

Let us describe how we map a four-point configuration to the conformal frame by conformal transformations. Let $C_L = (x_1, x_2, x_3, x_4)$ be a Lorentzian configuration. We will go from $C_L$ to $C_L'$ in a few steps, and each step is a conformal transformation. The configuration after the k-th step is denoted by $C_L^{(k)}$.
**Step 1.** We move $x_1$ to 0 by translation. The configuration $C_L^{(1)}$ after the first step is given by

$$
C_L^{(1)} = \left( x_1^{(1)}, x_2^{(1)}, x_3^{(1)}, x_4^{(1)} \right) = (0, x_2 - x_1, x_3 - x_1, x_4 - x_1).
\tag{177}
$$

This step preserves the causal ordering.
**Step 2.** We move $x_4$ to $\infty$ by special conformal transformation

$$
x'^\mu = \frac{x^\mu - x^2 b^\mu}{1 - 2x \cdot b + x^2 b^2}, \quad b^\mu = \frac{(x_4 - x_1)^\mu}{(x_4 - x_1)^2}.
\tag{178}
$$

$x_1 = 0$ is preserved by special conformal transformation. This step may change the causal ordering. Under general conformal transformations, $x_{ij}^2$ transforms as

$$
\begin{aligned}
(x_i' - x_j')^2 &= \Omega(x_i)\Omega(x_j)(x_i - x_j)^2, \\
(ds'^2 &= \Omega(x)^2 ds^2),
\end{aligned}
\tag{179}
$$

and for special conformal transformation (178), the scaling factor at $x_k^{(1)} = x_k - x_1$ is given by

$$
\Omega\left(x_k^{(1)}\right) = \frac{(x_4 - x_1)^2}{(x_4 - x_k)^2}, \quad k = 1, 2, 3, 4.
\tag{180}
$$

Let $C_L^{(2)} = \left( x_1^{(2)}, x_2^{(2)}, x_3^{(2)}, x_4^{(2)} \right)$ be the configuration after step 2. For any configuration $C_L$ in $\mathcal{D}_L$ (where the light-cone singularities are excluded), by (179) and (180) we have

$$
\left( x_i^{(2)} - x_j^{(2)} \right)^2 \neq 0, \quad i, j = 1, 2, 3,
\tag{181}
$$

Furthermore, the sign of $\left( x_i^{(2)} - x_j^{(2)} \right)^2$ is determined by the causal ordering of $C_L$. So we know if each $x_i^{(2)}, x_j^{(2)}$ pair of $C_L^{(2)}$ is space-like or time-like. The information we do not know a priori from (179) and

(180) is the causal orderings of time-like $x_i^{(2)}, x_j^{(2)}$ pairs (who is in the future of whom).[36]

**Step 3.** We move $x_3$ to its final position by some composition of Lorentz transformations, dilatations and time reversal $\theta_L$ (these conformal transformations preserve $x_1 = 0$ and $x_4 = \infty$). Lorentz transformations and dilatations preserve causal orderings, and time reversal only reverse causal orderings (i.e. $x_i, x_j$ pairs change from time-like to time-like, or from space-like to space-like). There are two possibilities after step 2: $x_3^{(2)}$ could be space-like or time-like to $x_1^{(2)}$. If $x_1^{(2)}, x_3^{(2)}$ are time-like, then $x_3^{(3)}$ is put at $(i, 0, \ldots, 0)$. If $x_1^{(2)}, x_3^{(2)}$ are space-like, then $x_3^{(3)}$ is put at $(0, 1, 0, \ldots, 0)$. Therefore, $C_L^{(3)}$ is in one of the following forms:

1. $x_1^{(3)} = 0$, $x_3^{(3)} = (i, 0, \ldots, 0)$, $x_4^{(3)} = \infty$.

2. $x_1^{(3)} = 0$, $x_3^{(3)} = (0, 1, 0, \ldots, 0)$, $x_4^{(3)} = \infty$.

**Step 4.** We move $x_2$ to somewhere in the (01)-plane or (12)-plane by Lorentz transformations in the little group of $x_3^{(3)}$. If $x_3^{(3)} = (i, 0, \ldots, 0)$, then we move $x_2$ to the (01)-plane by rotation, i.e. $x_2^{(4)} = (ia, b, 0, \ldots, 0)$. If $x_3^{(3)} = (0, 1, 0, \ldots, 0)$, then we move $x_2$ onto the (01)-plane or the (12)-plane, determined as follows:

- If $x_2^{(3)} = (i\beta_1, a, \beta_2, \ldots, \beta_{d-1})$ with $(\beta_1)^2 \geq (\beta_2)^2 + \ldots (\beta_{d-1})^2$, then $x_2$ is put in the (01)-plane, i.e. $x_2^{(4)} = (ib, a, 0, \ldots, 0)$ and $b^2 = (\beta_1)^2 - (\beta_2)^2 - \ldots - (\beta_{d-1})^2$.

- If $x_2^{(3)} = (i\beta_1, a, \beta_2, \ldots, \beta_{d-1})$ with $(\beta_1)^2 \leq (\beta_2)^2 + \ldots (\beta_{d-1})^2$, then $x_2$ is put in the (01)-plane, i.e. $x_2^{(4)} = (0, a, b, 0, \ldots, 0)$ and $b^2 = (\beta_2)^2 + \ldots + (\beta_{d-1})^2 - (\beta_1)^2$.

In the end, $C_L' = C_L^{(4)}$ has one of the forms in eq. (176). Moreover, the above discussion provides us with the following fact:

- the sign of $\left(x_{ij}'\right)^2$ of $C_L'$ is going to be the same for all configurations $C_L \in \mathcal{D}_L^\alpha$ in each causal ordering $\alpha$.

Now back to our question. Suppose $\mathcal{D}_L^\alpha$ is in class E. Let $C_L' = (x_1', x_2', x_3', x_4')$ be the conformal frame configuration of $C_L \in \mathcal{D}_L^\alpha$. Comparing the range of the $(a, b)$ pair in (176) with the range of the $(z, \bar{z})$ pair in class E (see section 5.1) and using the above fact, we see that there are only two possibilities for $C_L'$:

1. $\left(x_{13}'\right)^2, \left(x_{12}'\right)^2, \left(x_{23}'\right)^2 < 0$ for all $C_L \in \mathcal{D}_L^\alpha$. All possible $C_L'$ are given by the grey region in the first picture of figure C.1. In this case $z, \bar{z}$ are real, so we have

$$\mathcal{D}_L^\alpha = (\mathcal{D}_L^\alpha \cap \mathrm{E_{st}}) \sqcup (\mathcal{D}_L^\alpha \cap \mathrm{E_{su}}) \sqcup (\mathcal{D}_L^\alpha \cap \mathrm{E_{tu}}). \tag{182}$$

Because the $z, \bar{z}$ ranges corresponding to $\mathrm{E_{st}}, \mathrm{E_{su}}, \mathrm{E_{tu}}$ are disconnected from each other in figure C.1 and because $\mathcal{D}_L^\alpha$ is connected in $d \geq 3$, only one of the above intersections is non-empty. We conclude that such $\mathcal{D}_L^\alpha$ only belongs to one of the three subclasses $\mathrm{E_{st}}, \mathrm{E_{su}}, \mathrm{E_{tu}}$. This conclusion remains valid also in 2d, because 2d configurations can be embedded into higher d.

2. $\left(x_{13}'\right)^2, \left(x_{12}'\right)^2, \left(x_{23}'\right)^2 > 0$ for all $C_L \in \mathcal{D}_L^\alpha$. All possible conformal frame configurations are given by the grey region in the second and third pictures of figure C.1. In this case we have

$$\mathcal{D}_L^\alpha = (\mathcal{D}_L^\alpha \cap \mathrm{E_{st}}) \sqcup (\mathcal{D}_L^\alpha \cap \mathrm{E_{su}}) \sqcup (\mathcal{D}_L^\alpha \cap \mathrm{E_{tu}}) \sqcup (\mathcal{D}_L^\alpha \cap \mathrm{E_{stu}}), \tag{183}$$

which means that the configurations in $\mathcal{D}_L^\alpha$ may appear in all subclasses.

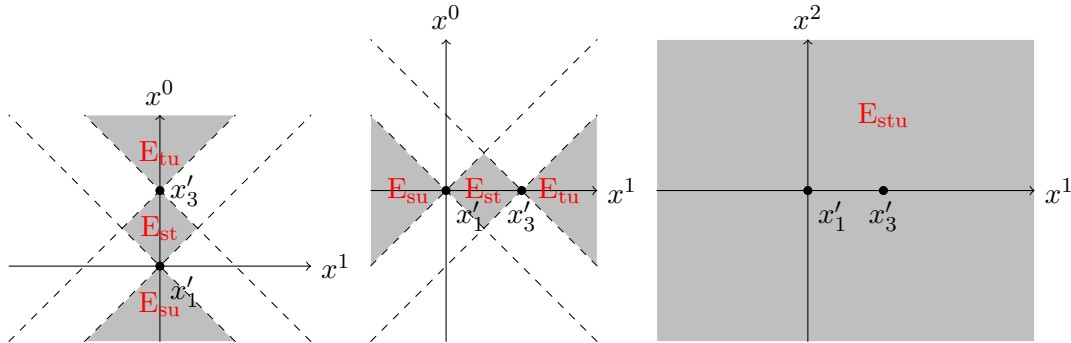

Figure C.1: The conformal frame configurations realized by configurations in class E.

To see which of these possibilities is realized, it is enough to know the sign of $(x'_{13})^2$. To finish the proof of lemma C.1, it remains to check that $(x'_{13})^2 < 0$ for all type 1, 6, 11 causal orderings (so that possibility 1 is realized for $C'_L$). Since all causal orderings of a fixed causal type can be realized by permuting the indices of the template causal ordering and such $S_4$-action permutes subclasses $\mathrm{E}_{st}, \mathrm{E}_{su}, \mathrm{E}_{tu}$ among themselves (see section C.0.1), it suffices to check that $(x'_{13})^2 < 0$ for the template causal orderings of causal type 1, 6, 11. Moreover, since we are only interested in the sign of $(x'_{13})^2$, we can use the following formula:

$$\mathrm{Sign}\left(\left(x'_{13}\right)^2\right) = \mathrm{Sign}\left(x^2_{13}x^2_{14}x^2_{34}\right). \tag{184}$$

(184) follows from (179), (180) and the fact that step 3,4 (of constructing the conformal frame) preserve the sign of $\left(x^{(k)}_{ij}\right)^2$. Now let us do the check.

**Type 1.** The template causal ordering is given by

$$1 \to 2 \to 3 \to 4, \tag{185}$$

which gives $x^2_{13}, x^2_{14}, x^2_{34} < 0$, hence $(x'_{13})^2 < 0$.

**Type 6.** The template causal ordering is given by

$$\begin{matrix} 1 \longrightarrow 2 \longrightarrow 3 \\ 4 \end{matrix}, \tag{186}$$

which gives $x^2_{13} < 0$ and $x^2_{14}, x^2_{34} > 0$, hence $(x'_{13})^2 < 0$.

**Type 11.** The template causal ordering is given by

$$\begin{matrix} 1 \longrightarrow 2 \\ 3 \longrightarrow 4 \end{matrix} \tag{187}$$

which gives $x^2_{13}, x^2_{14} > 0$ and $x^2_{34} < 0$, hence $(x'_{13})^2 < 0$.

So we finished the proof of lemma C.1. As a consistency check one can also compute the sign of $(x'_{13})^2$ for type 5,10,12 causal orderings and find that $(x'_{13})^2 > 0$ for these cases.

For type 1, 6, 11 causal orderings, to determine the subclasses, it remains to determine the subclass of one particular configuration of the template causal ordering in each type, then determine the subclasses of other causal orderings by looking up table 3.

We would like to remark that Lorentzian conformal frame is just a way to figure out the range of $z, \bar{z}$. There is no claim that correlation functions at $C_L, C'_L$ agree. As mentioned in section 6.3.3, the global conformal invariance does not hold in a general Lorentzian CFT.

---

[36]Of course for any particular configuration we can just compute $C^{(2)}_L$ and then determine its causal ordering.

### C.0.3 Symmetry of the graph

Usually, we go from one causal ordering to another by permuting the indices. For example, given the causal ordering $1 \to 2 \to 3 \to 4$, by permuting $1, 4$ we get another causal ordering $4 \to 2 \to 3 \to 1$.

However, a causal ordering may have a non-trivial little group.[37] For example, consider the following causal ordering

$$a \longrightarrow b \underset{d}{\overset{c}{\diagdown}} . \tag{188}$$

The causal ordering (188) does not change if we interchange $c$ and $d$, so it has a non-trivial little group $\mathbb{Z}_2$.

The little group is unique up to an isomorphism for all causal orderings in the same causal type. Let $G$ be the little group of one causal type and let $|G|$ be the order of $G$. Then the total number of causal orderings in this type is given by $24/|G|$. So in the table of each causal type, we will only list $24/|G|$ sequences $(i_1 i_2 i_3 i_4)$. Below the tables, we will point out the sequences which give the same causal ordering.

### C.1 Type 1

The type 1 causal ordering is given by

$$a \to b \to c \to d. \tag{189}$$

We let (189) be the template causal ordering, then the causal ordering $i_1 \to i_2 \to i_3 \to i_4$ is labelled by the sequence $(i_1 i_2 i_3 i_4)$. Any permutation of the indices in (189) will change the causal ordering, so the little group of the type 1 causal orderings is trivial. We have to list 24 causal orderings in the table.

Under time reversal $\theta_L$, $a \to b \to c \to d$ is mapped to $d \to c \to b \to a$, which is equivalent to the permutation $\theta_L : a \leftrightarrow d, b \leftrightarrow c$. This action is causal-type specific. In addition, we have $\theta_E$ action which is always given by $\theta_E : 1 \leftrightarrow 4, 2 \leftrightarrow 3$ (see eq. (98)). Using $\mathbb{Z}_2 \times \mathbb{Z}_2$ generated by these permutations, we divide 24 type 1 causal orderings into 8 orbits:

1. (1234), (4321).

2. (1243), (4312), (3421), (2134).

3. (1324), (4231).

4. (1423), (3241), (4132), (2314).

5. (1342), (2431), (4213), (3124).

6. (1432), (2341), (4123), (3214).

7. (2143), (3412).

8. (2413), (3142).

---

[37] By little group of a causal ordering, we mean the permutations of the indices which do not change this causal ordering.

As discussed in section 5.4, all the causal orderings in each orbit have the same convergent OPE channels.

Let us consider (1234), or equivalently the causal ordering $1 \to 2 \to 3 \to 4$. We first pick a particular configuration and compute $z, \bar{z}$. Here we choose

$$x_1 = 0, \ x_2 = (i, 0, \ldots, 0), \ x_3 = (2i, 0, \ldots, 0), \ x_4 = (3i, 0, \ldots, 0) \tag{190}$$

and get $z = \bar{z} = \frac{1}{4}$, which is in the range corresponding to subclass $E_{st}$. By lemma C.1, the whole (1234) is in subclass $E_{st}$.

All other $(i_1 i_2 i_3 i_4)$ causal orderings can be obtained by applying permutation $\sigma = [i_1 i_2 i_3 i_4]$ to the template ordering (1234). Using table 3, we can easily determine the subclasses of all other $(i_1 i_2 i_3 i_4)$ sequences (look at the column having $E_{st}$ on top). Then by looking up table 1, we get some OPE convergence properties of type 1 causal orderings, which are summarized in table 4.

| causal ordering | class/subclass | s-channel | t-channel | u-channel |
|---|---|---|---|---|
| (1234), (4321) | $E_{st}$ | ✓ | | ✗ |
| (1243), (3421), (4312), (2134) | $E_{su}$ | ✓ | ✗ | |
| (1324), (4231) | $E_{tu}$ | ✗ | | |
| (1423), (3241), (4132), (2314) | $E_{tu}$ | ✗ | | |
| (1342), (2431), (4213), (3124) | $E_{su}$ | ✓ | ✗ | |
| (1432), (2341), (4123), (3214) | $E_{st}$ | ✓ | | ✗ |
| (2143), (3412) | $E_{st}$ | ✓ | | ✗ |
| (2413), (3142) | $E_{tu}$ | ✗ | | |

Table 4:   The classes/subclasses of type 1 causal orderings

It remains to complete the rest of table 4. For this we choose a representative configuration for each orbit of causal orderings and choose a path to compute the curves of $z, \bar{z}$, then decide the convergence properties of t- and u-channel expansions. In practice this is done numerically, by plotting the curves of $z, \bar{z}$ and staring at the plots to determine $N_t, N_u$ (as in examples in section 5.6). The final results of OPE convergence properties in the three channels are shown in table 5, where we use red to indicate the new marks.

| causal ordering | class/subclass | s-channel | t-channel | u-channel |
|---|---|---|---|---|
| (1234), (4321) | $E_{st}$ | ✓ | ✓ | ✗ |
| (2143), (3412) | $E_{st}$ | ✓ | ✗ | ✗ |
| (1432), (2341), (4123), (3214) | $E_{st}$ | ✓ | ✓ | ✗ |
| (1324), (4231) | $E_{tu}$ | ✗ | ✓ | ✗ |
| (2413), (3142) | $E_{tu}$ | ✗ | ✗ | ✗ |
| (1423), (3241), (4132), (2314) | $E_{tu}$ | ✗ | ✓ | ✗ |
| (1342), (2431), (4213), (3124) | $E_{su}$ | ✓ | ✗ | ✗ |
| (1243), (3421), (4312), (2134) | $E_{su}$ | ✓ | ✗ | ✗ |

Table 5: OPE convergence properties of type 1 causal orderings

## C.2 Type 2

The type 2 causal ordering is given by

$$a \longrightarrow b \overset{\nearrow c}{\underset{\searrow d}{}} \quad , \quad \overset{c \searrow}{\underset{d \nearrow}{}} b \longrightarrow a \ . \tag{191}$$

We choose a particular configuration for (1234):

$$x_1 = 0, \ x_2 = (i, 0, \ldots, 0), \ x_3 = (2i, 0.5, 0, \ldots, 0), \ x_4 = (2i, -0.5, 0, \ldots, 0), \tag{192}$$

and get $z = -\frac{8}{5}$, $\bar{z} = \frac{4}{9}$, which is in the range corresponding to class S. So we conclude that (1234) is in class S. The remaining steps are the same as appendix C.1. The results of OPE convergence properties in the three channels are shown in table 6.

Table 6: OPE convergence properties of type 2 causal orderings

| causal ordering | class/subclass | s-channel | t-channel | u-channel |
|---|---|---|---|---|
| (1234), (4321) | S | ✓ | ✗ | ✗ |
| (1324), (4231) | U | ✗ | ✗ | ✗ |
| (1423), (4132) | T | ✗ | ✓ | ✗ |
| (2134), (3421) | S | ✓ | ✗ | ✗ |
| (3124), (2431) | U | ✗ | ✗ | ✗ |
| (2314), (3241) | T | ✗ | ✓ | ✗ |

There are only 12 causal orderings because $(i_1 i_2 i_3 i_4)$ and $(i_1 i_2 i_4 i_3)$ are the same causal ordering (the little group is $\mathbb{Z}_2$).

## C.3 Type 3

The type 3 causal ordering is given by

$$a \overset{\longrightarrow}{\underset{\searrow d}{}} b \longrightarrow c \quad , \quad c \longrightarrow b \overset{\longrightarrow a}{\underset{\nearrow d}{}} \ . \tag{193}$$

We choose a particular configuration for (1234):

$$x_1 = 0, \ x_2 = (i, 0, \ldots, 0), \ x_3 = (2i, 0, \ldots, 0), \ x_4 = (1.5i, 1, 0, \ldots, 0), \tag{194}$$

and get $z = \frac{1}{6}$, $\bar{z} = \frac{3}{2}$, which is in the range corresponding to class T. So we conclude that (1234) is in class T. The results of OPE convergence properties in the three channels are shown in table 7:

Table 7: OPE convergence properties of type 3 causal orderings

| causal ordering | class/subclass | s-channel | t-channel | u-channel |
|---|---|---|---|---|
| (1234), (4321) | T | ✗ | ✓ | ✗ |
| (1243), (4312) | U | ✗ | ✗ | ✓ |
| (1324), (4231) | T | ✗ | ✓ | ✗ |
| (1342), (4213) | S | ✓ | ✗ | ✗ |
| (1423), (4132) | U | ✗ | ✗ | ✓ |
| (1432), (4123) | S | ✓ | ✗ | ✗ |
| (2134), (3421) | U | ✗ | ✗ | ✗ |
| (2143), (3412) | T | ✗ | ✗ | ✗ |
| (2314), (3241) | U | ✗ | ✗ | ✗ |
| (2341), (3214) | S | ✓ | ✗ | ✗ |
| (2413), (3142) | T | ✗ | ✗ | ✗ |
| (2431), (3124) | S | ✓ | ✗ | ✗ |

## C.4 Type 4

The type 4 causal ordering is given by

$$
a \nearrow b \searrow d \searrow c \nearrow d .
\tag{195}
$$

We choose a particular configuration for (1234):

$$
x_1 = 0, \ x_2 = (i, 0.5, 0, \ldots, 0), \ x_3 = (i, -0.5, 0, \ldots, 0), \ x_4 = (2i, 0, \ldots, 0),
\tag{196}
$$

and get $z = \frac{1}{9}$, $\bar{z} = 9$, which is in the range corresponding to class T. So we conclude that (1234) is in class T. The results of OPE convergence properties in the three channels are shown in table 8:

Table 8: OPE convergence properties of type 4 causal orderings

| causal ordering | class/subclass | s-channel | t-channel | u-channel |
|---|---|---|---|---|
| (1234), (4321) | T | ✗ | ✓ | ✗ |
| (1243), (4312), (2134), (3421) | U | ✗ | ✗ | ✗ |
| (1342), (4213), (2341), (3214) | S | ✓ | ✗ | ✗ |
| (2143), (3412) | T | ✗ | ✗ | ✗ |

Here we use the fact that $(i_1 i_2 i_3 i_4)$ and $(i_1 i_3 i_2 i_4)$ are the same causal ordering (the little group is $\mathbb{Z}_2$).

## C.5 Type 5

The type 5 causal ordering is given by

$$
a \overset{\nearrow}{\underset{\searrow}{\rightrightarrows}} \begin{matrix} b \\ c \\ d \end{matrix} , \qquad \begin{matrix} b \\ c \\ d \end{matrix} \rightrightarrows a . \tag{197}
$$

We choose a particular configuration for (1234):

$$
x_1 = 0, \; x_2 = (i, 0.5, 0, \ldots, 0), \; x_3 = (i, 0, \ldots, 0), \; x_4 = (i, -0.5, \ldots, 0), \tag{198}
$$

and get $z = \frac{1}{4}$, $\bar{z} = \frac{3}{4}$, which is in the range corresponding to subclass $E_{st}$. So (1234) is in class E. We would like to find a particular configuration in each subclass of class E. The little group of this causal type is $S_3$, which corresponds to permutations among $b, c, d$ in (197). By looking up table 3, we see that permuting $x_2, x_3$ in (198) gives $E_{tu}$ and permuting $x_3, x_4$ gives $E_{su}$. To realize $E_{stu}$ we choose the following configuration in (1234):

$$
x_1 = 0, \; x_2 = (i, 0.5, 0, \ldots, 0), \; x_3 = (i, -0.5, 0, \ldots, 0), \; x_4 = (i, 0, 0.5, 0, \ldots, 0), \tag{199}
$$

and get $z = i$, $\bar{z} = -i$, which is indeed in the range corresponding to subclass $E_{stu}$. So we conclude that the configurations of (1234) do appear in all subclasses of class E in $d \geq 3$, while they only appear in subclasses $E_{st}, E_{su}, E_{tu}$ in 2d.[38]

The results of OPE convergence properties in the three channels are shown in table 9:

Table 9: OPE convergence properties of type 5 causal orderings

| causal ordering | class/subclass | s-channel | t-channel | u-channel |
|---|---|---|---|---|
| (1234), (4321) | $E_{st}$ | ✓ | ✓ | ✗ |
| | $E_{su}$ | ✓ | ✗ | ✓ |
| | $E_{tu}$ | ✗ | ✓ | ✓ |
| | $E_{stu}$ | ✓ | ✓ | ✓ |
| (2134), (3124) | $E_{st}$ | ✓ | ✗ | ✗ |
| | $E_{su}$ | ✓ | ✗ | ✗ |
| | $E_{tu}$ | ✗ | ✗ | ✗ |
| | $E_{stu}$ | ✓ | ✗ | ✗ |

Here we use the fact that for $(i_1 i_2 i_3 i_4)$, any permutation of $2, 3, 4$ does not change the causal ordering (the little group is $S_3$).

## C.6 Type 6

The type 6 causal ordering is given by

$$
\begin{matrix} a \longrightarrow b \longrightarrow c \\ d \end{matrix} . \tag{200}
$$

---

[38] We used two dimensions in (198) and three dimensions in (199). On the other hand, as mentioned at the end of section 5.3.2, subclass $E_{stu}$ does not exist in 2d because $z, \bar{z}$ can only be real.

We choose a particular configuration for (1234):

$$x_1 = 0, \ x_2 = (i, 0, \dots, 0), \ x_3 = (2i, 0, \dots, 0), \ x_4 = (i, 2, \dots, 0), \tag{201}$$

and get $z = \frac{1}{4}$, $\bar{z} = \frac{3}{4}$, which is in the range corresponding to subclass E$_{\text{st}}$. By lemma C.1, the whole (1234) is in subclass E$_{\text{st}}$. The results of OPE convergence properties in the three channels are shown in table 10.

Table 10: OPE convergence properties of type 6 causal orderings

| causal ordering | class/subclass | s-channel | t-channel | u-channel |
|---|---|---|---|---|
| (1234), (4321), (3214), (2341) | E$_{\text{st}}$ | ✓ | ✓ | ✗ |
| (1243), (4312), (4213), (1342) | E$_{\text{su}}$ | ✓ | ✗ | ✓ |
| (1324), (4231), (2314), (3241) | E$_{\text{tu}}$ | ✗ | ✓ | ✗ |
| (1423), (4132), (2413), (3142) | E$_{\text{tu}}$ | ✗ | ✗ | ✓ |
| (1432), (4123), (3412), (2143) | E$_{\text{st}}$ | ✓ | ✗ | ✗ |
| (2134), (3421), (3124), (2431) | E$_{\text{su}}$ | ✓ | ✗ | ✗ |

## C.7  Type 7

The type 7 causal ordering is given by

$$
\begin{array}{cc}
\ & b \qquad b \\
a \searrow \quad \searrow a \\
\quad c \ , \quad c \\
\ d \qquad d
\end{array}
\tag{202}
$$

We choose a particular configuration for (1234):

$$x_1 = 0, \ x_2 = (i, 0.5, 0, \dots, 0), \ x_3 = (i, -0.5, 0, \dots, 0), \ x_4 = (0, 2, 0, \dots, 0), \tag{203}$$

and get $z = \frac{7}{15}$, $\bar{z} = 9$, which is in the range corresponding to class T. So we conclude that (1234) is in class T. The results of OPE convergence properties in the three channels are shown in table 11.

Table 11: OPE convergence properties of type 7 causal orderings

| causal ordering | class/subclass | s-channel | t-channel | u-channel |
|---|---|---|---|---|
| (1234), (4321) | T | ✗ | ✓ | ✗ |
| (1243), (4312) | U | ✗ | ✗ | ✓ |
| (1342), (4213) | S | ✓ | ✗ | ✗ |
| (2134), (3421) | U | ✗ | ✗ | ✗ |
| (2143), (3412) | T | ✗ | ✗ | ✗ |
| (2341), (3124) | S | ✓ | ✗ | ✗ |

Here we use the fact that $(i_1 i_2 i_3 i_4)$ and $(i_1 i_3 i_2 i_4)$ are the same causal ordering (the little group is $\mathbb{Z}_2$).

## C.8 Type 8

The type 8 causal ordering is given by

$$a \begin{array}{c} \nearrow b \\ \searrow d \\ \nearrow \\ c \end{array} \quad . \tag{204}$$

We choose a particular configuration for (1234):

$$x_1 = 0, \ x_2 = (i, -0.5, 0, \ldots, 0), \ x_3 = (0, 1, 0, \ldots, 0), \ x_4 = (i, 0.5, 0, \ldots, 0), \tag{205}$$

and get $z = \frac{1}{4}$, $\bar{z} = \frac{9}{4}$, which is in the range corresponding to class T. So we conclude that (1234) is in class T. The results of OPE convergence properties in the three channels are shown in table 12:

Table 12: OPE convergence properties of type 8 causal orderings

| causal ordering | class/subclass | s-channel | t-channel | u-channel |
|---|---|---|---|---|
| (1234), (4321) | T | ✗ | ✗ | ✗ |
| (1243), (4312), (3421), (2134) | U | ✗ | ✗ | ✓ |
| (1342), (4213), (2431), (3124) | S | ✓ | ✗ | ✗ |
| (2143), (3412) | T | ✗ | ✗ | ✗ |
| (1324), (4231) | T | ✗ | ✗ | ✗ |
| (1432), (4123), (2341), (3214) | S | ✓ | ✗ | ✗ |
| (1423), (4132), (3241), (2314) | U | ✗ | ✗ | ✗ |
| (2413), (3142) | T | ✗ | ✗ | ✗ |

## C.9 Type 9

The type 9 causal ordering is given by

$$\begin{array}{c} a \longrightarrow b \\ c \\ d \end{array} \quad . \tag{206}$$

We choose a particular configuration for (1234):

$$x_1 = 0, \ x_2 = (i, 0, \ldots, 0), \ x_3 = (0, 2, 0, \ldots, 0), \ x_4 = (0, 3, 0, \ldots, 0), \tag{207}$$

and get $z = -\frac{1}{8}$, $\bar{z} = \frac{1}{4}$, which is in the range corresponding to class S. So we conclude that (1234) is in class S. The results of OPE convergence properties in the three channels are shown in table 13:

Table 13: OPE convergence properties of type 9 causal orderings

| causal ordering | class/subclass | s-channel | t-channel | u-channel |
|---|---|---|---|---|
| (1234), (4312), (2134), (3412) | S | ✓ | ✗ | ✗ |
| (1324), (4213), (3124), (2413) | U | ✗ | ✗ | ✓ |
| (1423), (4123) | T | ✗ | ✓ | ✗ |
| (2314), (3214) | T | ✗ | ✓ | ✗ |

Here we use the fact that $(i_1 i_2 i_3 i_4)$ and $(i_1 i_2 i_4 i_3)$ are the same causal ordering (the little group is $\mathbb{Z}_2$).

## C.10 Type 10

The type 10 causal ordering is given by

$$a \searrow \atop b \nearrow \quad c \to d \,. \tag{208}$$

We choose a particular configuration for (1234):

$$x_1 = 0, \;\; x_2 = (0,1,0,\ldots,0), \;\; x_3 = (2i,0,\ldots,0), \;\; x_4 = (2i,1,\ldots,0), \tag{209}$$

and get $z = \bar{z} = \frac{1}{4}$, which is in the range corresponding to subclass $\mathrm{E}_{st}$. So (1234) is in class E. We would like to find a particular configuration in each subclass of class E. The little group of this causal type is $\mathbb{Z}_2 \times \mathbb{Z}_2$, which is generated by $a \leftrightarrow b$ and $c \leftrightarrow d$ in (208). By looking up table 3, we see that by acting with the little group on configuration (209), we can get $\mathrm{E}_{su}$, but we cannot get $\mathrm{E}_{tu}$. The underlying fact is that the 2d configurations of (1234) do not appear in $\mathrm{E}_{tu}$ (it is obvious that 2d configurations do not appear in $\mathrm{E}_{stu}$.). Let us show this fact. In 2d Minkowski space we can use the light-cone coordinates:

$$z_k = t_k + \mathbf{x}_k, \quad \bar{z}_k = t_k - \mathbf{x}_k, \quad (x_k = (it_k, \mathbf{x}_k)) \,. \tag{210}$$

The causal ordering (208) implies

$$z_3, z_4 > z_1, z_2, \quad \bar{z}_3, \bar{z}_4 > \bar{z}_1, \bar{z}_2, \\ (z_1 - z_2)(\bar{z}_1 - \bar{z}_2) < 0, \quad (z_3 - z_4)(\bar{z}_3 - \bar{z}_4) < 0. \tag{211}$$

Since the little group $\mathbb{Z}_2 \times \mathbb{Z}_2$ of (1234) preserves $\mathrm{E}_{tu}$ (see table 3), by the $\mathbb{Z}_2 \times \mathbb{Z}_2$-action, it suffices to show that $\mathrm{E}_{tu}$ configurations do not exist when

$$z_3, z_4 > z_1, z_2, \quad \bar{z}_3, \bar{z}_4 > \bar{z}_1, \bar{z}_2, \\ z_1 - z_2 < 0, \quad \bar{z}_1 - \bar{z}_2 > 0, \\ z_3 - z_4 < 0, \quad \bar{z}_3 - \bar{z}_4 > 0. \tag{212}$$

In this case the computation is straightforward:

$$z = \frac{(z_2 - z_1)(z_4 - z_3)}{(z_3 - z_1)(z_4 - z_2)} < \frac{(z_3 - z_1)(z_4 - z_2)}{(z_3 - z_1)(z_4 - z_2)} = 1, \\ \bar{z} = \frac{(\bar{z}_1 - \bar{z}_2)(\bar{z}_3 - \bar{z}_4)}{(\bar{z}_3 - \bar{z}_1)(\bar{z}_4 - \bar{z}_2)} < \frac{(\bar{z}_4 - \bar{z}_2)(\bar{z}_3 - \bar{z}_1)}{(\bar{z}_3 - \bar{z}_1)(\bar{z}_4 - \bar{z}_2)} = 1. \tag{213}$$

So we conclude that the 2d configurations in (1234) have $z, \bar{z} < 1$, i.e. (1234) does not intersect with subclass $\mathrm{E}_{tu}$ in 2d. To find a $\mathrm{E}_{tu}$ configuration in (1234) we need to construct it in $d \geq 3$. We choose the 3d configuration (108) and get $z \approx 1.1$, $\bar{z} \approx 6.3$, which is in the range corresponding to subclass $\mathrm{E}_{tu}$.

To realize $\mathrm{E}_{stu}$ we choose the following configuration in (1234):

$$x_1 = 0, \;\; x_2 = (0,0.5,0,\ldots,0), \;\; x_3 = (2i,0,0.5,0,\ldots,0), \;\; x_4 = (i,0.5,0,\ldots,0), \tag{214}$$

and get $z \approx 0.33 + 0.24i$, $\bar{z} = 0.33 - 0.24i$, which is in the range corresponding to subclass $\mathrm{E}_{stu}$. So we conclude that the configurations of (1234) do appear in all subclasses of class E in $d \geq 3$, while they only appear in subclasses $\mathrm{E}_{st}, \mathrm{E}_{su}$ in 2d.

The results of OPE convergence properties in the three channels are shown in table 14:

Table 14: OPE convergence properties of type 10 causal orderings

| causal ordering | class/subclass | s-channel | t-channel | u-channel |
|---|---|---|---|---|
| (1234), (3412) | $E_{st}$ | ✓ | ✗ | ✗ |
| | $E_{su}$ | ✓ | ✗ | ✗ |
| | $E_{tu}$ | ✗ | ✗ | ✗ |
| | $E_{stu}$ | ✓ | ✗ | ✗ |
| (1324), (2413) | $E_{st}$ | ✓ | ✗ | ✗ |
| | $E_{su}$ | ✓ | ✗ | ✗ |
| | $E_{tu}$ | ✗ | ✗ | ✗ |
| | $E_{stu}$ | ✓ | ✗ | ✗ |
| (1423), (2314) | $E_{st}$ | ✓ | ✓ | ✗ |
| | $E_{su}$ | ✓ | ✗ | ✓ |
| | $E_{tu}$ | ✗ | ✓ | ✓ |
| | $E_{stu}$ | ✓ | ✓ | ✓ |

Here we use the fact $(i_1i_2i_3i_4)$, $(i_2i_1i_3i_4)$, $(i_1i_2i_4i_3)$ and $(i_2i_1i_4i_3)$ are the same causal ordering (the little group is $\mathbb{Z}_2 \times \mathbb{Z}_2$).

## C.11 Type 11

The type 11 causal ordering is given by

$$
\begin{aligned}
a &\longrightarrow b \\
c &\longrightarrow d
\end{aligned} . \tag{215}
$$

We choose a particular configuration for (1234):

$$
x_1 = 0, \ x_2 = (i, 0, \ldots, 0), \ x_3 = (0, 2, \ldots, 0), \ x_4 = (i, 2, \ldots, 0), \tag{216}
$$

and get $z = \bar{z} = \frac{1}{4}$, which is in the range corresponding to subclass $E_{st}$. By lemma C.1, the whole (1234) is in subclass $E_{st}$. The results of OPE convergence properties in the three channels are shown in table 15.

Table 15: OPE convergence properties of type 11 causal orderings

| causal ordering | class/subclass | s-channel | t-channel | u-channel |
|---|---|---|---|---|
| (1234), (2143) | $E_{st}$ | ✓ | ✗ | ✗ |
| (1243), (2134) | $E_{su}$ | ✓ | ✗ | ✓ |
| (1324), (3142) | $E_{tu}$ | ✗ | ✗ | ✓ |
| (1342), (3124) | $E_{su}$ | ✓ | ✗ | ✓ |
| (1423), (3241) | $E_{tu}$ | ✗ | ✓ | ✗ |
| (1432), (2341) | $E_{st}$ | ✓ | ✓ | ✗ |

Here we use the fact that $(i_1i_2i_3i_4)$ and $(i_3i_4i_1i_2)$ are the same causal ordering (the little group is $\mathbb{Z}_2$).

## C.12   Type 12

The type 12 causal ordering is given by

$$a \quad b \quad c \quad d. \tag{217}$$

One can show that this causal type belongs to class E. In fact this is the "Euclidean" case (that's why this class is called class E), and there is no need to check numerically for this type. Suppose we have a totally space-like separated configuration $(x_1, x_2, x_3, x_4)$, where $x_k = (it_k, \mathbf{x}_k)$. We can reach this configuration via the path

$$x_k(s) = \begin{cases} ((1 - 2s)\epsilon_k, 2s\mathbf{x}_k) & s \in [0, 1/2] \\ ((2s - 1)it_k, \mathbf{x}_k) & s \in [1/2, 1] \end{cases} \tag{218}$$

Along the path all the $x_i, x_j$ pairs are space-like separated. As a result, the totally space-like separated configurations always have $N_t = N_u = 0$ (as long as $N_t, N_u$ are well-defined). On the other hand, there is no doubt that all subclasses of class E can be realized in $d \geq 3$.[39] We summarize the OPE convergence properties of this type in table 16.

Table 16: OPE convergence properties of type 12 causal orderings

| causal ordering | class/subclass | s-channel | t-channel | u-channel |
|---|---|---|---|---|
| (1234) | $E_{st}$ | ✓ | ✓ | ✗ |
| | $E_{su}$ | ✓ | ✗ | ✓ |
| | $E_{tu}$ | ✗ | ✓ | ✓ |
| | $E_{stu}$ | ✓ | ✓ | ✓ |

Here we use the fact that there is only 1 causal ordering in this type (the little group is $S_4$).

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
