# Peer review of "Classification of Convergent OPE Channels for Lorentzian CFT Four-Point Functions"

_SciPost Physics_

## Round 4 · Referee Report · Sachin Jain (Referee 1) · 2022-4-21

Report

Three different sets of consistency conditions/axioms plays important role in the context of QFT/CFT. 1. Euclidean CFT Consistency conditions: This include unitarity which puts bound on scaling dimension of scalar primary operator, requirement of convergent OPE. 2. Osterwalder-Schrader (OS) theorem 
 3.Wightman axioms

Some of the important question to understand are, how these consistency condition/axioms are related, given Euclidean correlation function how to obtain consistent Minkowski CFT correlation function.

OS to Wightman relation is well studied and one requires some growth condition to be consistent with each other. Question is how Euclidean CFT corrector are consistent with OS or directly how Euclidean CFT corrector can be made consistent with out relying on OS theorem.

This paper is very important step toward these interesting issues.

As a step towards understanding these issues, authors study convergence properties of OPE for Lorentzian CFT Correlators and give classification of Lorentzian correlator based on these convergent properties. One interesting out come for example is, that there are regions in Minkowski space where no OPE channel is convergent.

This paper should be published. However I have few questions/suggestions

  1. Author analytically continue the Euclidean corrector to domain D. Can the analytic continuation not be done in much larger domain beyond D?

  2. In the context of analytic continuation directly from Eucledian CFT corrector to Whitman function: Some discussion on growth condition of OS theorem should be included. In particular how is the growth condition related to Euclidean CFT correaltor consistency conditions? Violation of Growth condition should come as a obstruction to analytic continuation from eucledean CFT correlator. Is this understood?

  3. Paper is rather technical and long. It would greatly help reading if more concrete examples are given.

  4. Author in section 5.2 mentions that special conformal transformation can take space like separation to time like separation and vice versa which messes up with causal ordering. This might seem very puzzling to whole discussion of the paper. Author should explain why this does not lead to any problem and possibly with concrete example.

  5. First sentences in summary section 2.2 in each case looks incomplete with full stop. Author may like to complete the sentence or use appropriate punctuation.

  • validity: high
  • significance: high
  • originality: high
  • clarity: high
  • formatting: excellent
  • grammar: excellent

Author:  Jiaxin Qiao  on 2022-04-24  [id 2413]

(in reply to Report 1 by Sachin Jain on 2022-04-21)
Category:
answer to question

I thank the referee for the patient reading and the helpful suggestions/questions. Here are my responses.

  1. The analytic continuation can indeed be done in the much larger domain. One example is the forward tube T, where all the variables are complex (in contrast, D only considers real spatial variables). The main goal of my paper is to study the analyticity domain of the Euclidean correlator in the Lorentzian regime. For this purpose, domain D is big enough because all the Lorentzian four-point configurations live on the boundary of D (i.e. a slight extension of D will include the Lorentzian regime).

  2. If needed, I will comment on this in the revised version as follows. It is not understood whether the growth condition of the OS theorem is a consistency condition or not. It is introduced as a technical condition for showing that the analytically continued Euclidean correlator is a tempered distribution in the Lorentzian regime. Without the growth condition, the Euclidean correlator still has analytic continuation to the domain D. However, since D does not contain any Lorentzian configurations, one has to take a limit towards the Lorentzian regime from D. To show such a limit exists in the sense of tempered distributions, one needs to derive some power-law upper bound of the correlator in D (or T), and this is where the growth condition is used.

  3. I have an example section (sec. 5.6). I'd be happy to provide more examples, but this suggestion is too general for me. It would be better if more specific suggestions are given, e.g. which arguments need examples?

  4. Indeed the special conformal transformation messes up with causal orderings, but it does not contradict the results in this paper. My classification of Lorentzian four-point configurations is according to causal orderings instead of conformal equivalence classes. Suppose we have two configurations c1 and c2 such that (a) they have different causal orderings; (b) they are related to each other by a special conformal transformation, then the cross-ratios at c1 are the same as at c2, but they may have different OPE convergence properties. I will give a concrete example for this point in the revised version.

  5. Thanks for pointing out this. I will correct them in the revised version.

---

## Round 4 · Referee Report · Anonymous (Referee 2) · 2022-8-22

Report

This paper considers CFT four-point correlators in Lorentzian configurations. For each such configuration it decides whether there is an absolutely convergent OPE channel, which would imply that the correlator is a function rather than merely a distribution. The final results of this analysis are captured in numerous tables in the appendices. A comparison with the results obtainable from Wightman axioms is also included.

The paper is well written. The mathematical style is appropriate for this type of result and makes it easier to track the logic at each stage. The results are worthwhile and generally convincing. Altogether the paper certainly deserves publication. It would, however, in my opinion benefit from a few clarifications listed below.

In subsection 3.4.2 it is stated that $(\rho \bar \rho)^\Delta$ is an analytic function on $\mathcal D$. The evidence provided relies (among other things) that "$u^\Delta$ is an analytic function on $\mathcal D$ because $(x_{ij}^2)^\Delta$ are non-zero analytic functions on $\mathcal D$." The latter statement indeed holds because the $x_{ij}^2$ are never non-positive real (as per footnote 5) and therefore no path in $\mathcal D$ intersects with the branch cut of $(x_{ij}^2)^\Delta$. But I think it does not immediately follow that $u^{\Delta}$ is also analytic, since $u^\Delta = \left(\frac{x_{12}^2 x_{34}^2}{x_{13}^2 x_{24}^2}\right)^\Delta$ rather than $\frac{(x_{12}^2)^\Delta (x_{34}^2)^\Delta}{(x_{13}^2)^\Delta (x_{24}^2)^\Delta}$.

Similarly I would object to the usage of $\log(\rho) + \log(\bar \rho) = \log(\rho \bar \rho)$ on $\mathcal D$ without further clarification, as is implicitly done in the proof of lemma 3.3. This is particularly important since this lemma underlies most of the results of section 4.

In addition I think that it would be useful to comment why, in equation (31), a path that avoids $\Gamma$ for $0 \leq s < 1$ always exists.

Requested changes

I will not insist on any of the changes I suggested.

  • validity: -
  • significance: -
  • originality: -
  • clarity: -
  • formatting: -
  • grammar: -

Author:  Jiaxin Qiao  on 2022-08-23  [id 2748]

(in reply to Report 2 on 2022-08-22)

I thank the referee for the careful reading and for pointing out the parts of my paper that were unclear.

For the analytic continuation of $(\rho\bar{\rho})^{\Delta}$ from Euclidean regime to $\mathcal{D}$, I will make the statement clearer in the following way. The analytic continuation of $u^{\Delta}=\left(\frac{x_{12}^2 x_{34}^2}{x_{13}^2 x_{24}^2}\right)^{\Delta}$ is globally well-defined on $\mathcal{D}$ because (a) $x_{ij}^2$'s are non-zero analytic functions on $\mathcal{D}$, (b) $\mathcal{D}$ is simply connected. Condition (a) guarantees that the path-dependent analytic continuation can be performed, and condition (b) implies that the analytic continuation is path-independent (i.e. no monodromy issue).

By the similar reason (together with the fact that $z,\bar{z}\notin[1,+\infty]$), $\log(\rho\bar{\rho})$ has analytic continuation to $\mathcal{D}$. My point there is that $\log(\rho)+\log(\bar{\rho})=\log(\rho\bar{\rho})$ always holds along any fixed analytic continuation path. Although the extension of $\log(\rho)$ and $\log(\bar{\rho})$ depends on the choice of the analytic continuation path, the sum $\log(\rho)+\log(\bar{\rho})$ is path-independent since the r.h.s. of the equation is globally well-defined on $\mathcal{D}$.

I will make a revision concerning the above two points. Concerning eq.(31), a path exists bassically because $\Gamma$ is a "(real-)codimension-2 subspace" in $\mathcal{D}$ (I use quotes because there might be bad points, but a generic point in $\Gamma$ is good so we can say codim=2). Removing $\Gamma$ from $\mathcal{D}$ does not affect the connectedness (like removing discrete points in 2d, removing lines in 3d, etc..). I will make a comment with more details on this.

---

## Editorial Decision

resubmitted